# Thematic vent opening probability maps and hazard assessment of small-scale pyroclastic density currents in the San Salvador Volcanic Complex (El Salvador) and Nejapa-Chiltepe Volcanic Complex (Nicaragua)

Andrea Bevilacqua[1], Alvaro Aravena[2,3], Augusto Neri[1], Eduardo Gutiérrez[4,†], Demetrio Escobar[4], Melida Schliz[5], Alessandro Aiuppa[6], Raffaello Cioni[2]

[1]Istituto Nazionale di Geofisica e Vulcanologia, Sezione di Pisa, Pisa, Italy.
[2]Dipartimento di Scienze della Terra, Università di Firenze, Firenze, Italy.
[3]Laboratoire Magmas et Volcans, Université Clermont Auvergne, CNRS, IRD, OPGC, Clermont-Ferrand, France.
[4]Dirección del Observatorio Ambiental, MARN, San Salvador, El Salvador.
[5]Instituto de Geología y Geofísica, UNAN, Managua, Nicaragua.
[6]Dipartimento di Scienze della Terra e del Mare, Università di Palermo, Palermo, Italy.
[†]Recently deceased.

*Correspondence to*: A. Aravena (alvaro.aravena@uca.fr).

**Abstract.** San Salvador Volcanic Complex (El Salvador) and Nejapa-Chiltepe Volcanic Complex (Nicaragua) have been characterized by a significant variability in eruption style and vent location. Densely inhabited cities are built on them and their surroundings, including the metropolitan areas of San Salvador (~2.4M people) and Managua (~1.4M people), respectively. In this study we present novel vent opening probability maps for these volcanic complexes, which are based on a multi-model approach that relies on kernel density estimators. In particular, we present *thematic* vent opening maps, i.e. we consider different hazardous phenomena separately, including lava emission, small-scale pyroclastic density currents, ejection of ballistic projectiles, and low-intensity pyroclastic fallout. Our volcanological dataset includes: (1) the location of past vents, (2) the mapping of the main fault structures, and (3) the eruption styles of past events, obtained from the critical analysis of literature and/or inferred from volcanic deposits and morphological features observed remotely and in the field. To illustrate the effects of considering the expected eruption style in the construction of vent opening maps, we focus on the analysis of small-scale pyroclastic density currents derived from phreatomagmatic activity or from low-intensity magmatic volcanism. For the numerical simulation of these phenomena we adopted the recently developed branching energy cone model by using the program ECMapProb. Our results show that the implementation of thematic maps of vent opening can produce significantly different hazard levels from those estimated with traditional, non-thematic, maps.

## 1 Introduction

Volcanic hazard assessment is typically influenced by eruption style and the associated eruptive phenomena (e.g., pyroclastic fallout, ballistic projectiles, pyroclastic density currents, lava flows). A widely adopted methodology for assessing volcanic

hazard is based on the determination and analysis of a number of expectable eruptive scenarios (e.g., Newhall and Hoblitt, 2002; Cioni et al., 2008; Neri et al., 2008; Martí et al., 2012; Ferrés et al., 2013; Newhall and Pallister, 2015; Wright et al., 2019). These scenarios are commonly based on the characteristics of past eruptions, while their potential effects in the surrounding area can be evaluated by coupling field observations, uncertainty quantification and numerical simulations (e.g., Bonadonna et al., 2005; Del Negro et al., 2013; Bayarri et al., 2015; Neri et al., 2015; Bevilacqua et al., 2017b; Mead and Magill, 2017; Strehlow et al., 2017; Rutarindwa et al., 2019). In addition to the expected magnitude and eruption style, other sources of uncertainty for volcanic hazard assessment may derive from the unknown position of the vent during future eruptions, especially for volcanic fields associated with large calderas or controlled by regional tectonics (e.g., Connor et al., 2019). Accordingly, vent opening probability maps have become fundamental tools for the assessment of hazard and risk in different volcanic systems (e.g. Alberico et al., 2002; Connor et al., 2012; Bevilacqua et al., 2015; 2017a; Thompson et al., 2015; Poland and Anderson, 2020). The procedures adopted to construct these maps typically consider the data associated with vent position of past eruptions and, sometimes, major fault structures. Thus, they require the availability of a detailed data set of geological and volcanological information (Marzocchi and Bebbington, 2012; Connor et al., 2015; Németh and Kereszturi, 2015).

Vent opening maps have been developed for a growing number of volcanoes, such as Campi Flegrei (Alberico et al., 2002; Selva et al., 2012; Bevilacqua et al., 2015; 2017b), Somma-Vesuvius (Tadini et al., 2017a; 2017b), Etna (Cappello et al., 2012; Del Negro et al., 2020), Yucca Mountain Region (Connor and Hill, 1995; Connor et al., 2000), Main Ethiopian Rift (Mazzarini et al., 2013; 2016), Auckland Volcanic Field (Bebbington, 2013; 2015; Ang et al., 2020), Long Valley Volcanic Region (Bevilacqua et al., 2017a; 2018), El Hierro Island (Becerril et al., 2013), Pacaya Volcanic Complex (Rose et al., 2013), Harrat-Rahat Volcanic Field (Runge et al., 2014), Snake River Plain (Gallant et al., 2018), Taupo Volcanic Zone (Kósik et al., 2020), and Aluto volcano (Clarke et al., 2020), among others. Furthermore, the expected eruption style and resulting volcanic phenomena are likely influenced by vent position. In other words, vent position is not only important for volcanic hazard assessment because it controls the final dispersal of volcanic products, but also because it can be correlated with eruption style and intensity. However, there are significant difficulties associated with interpreting and modelling the dependence between eruption style and vent position (e.g. Thompson et al., 2015; Clarke et al., 2020). A number of different volcanic processes may produce this correlation, such as the involvement of different portions of the plumbing system, the presence of groundwater or surface water in specific zones of the volcanic field, or variations in the mechanical characteristics of the country rocks (e.g., Andronico et al., 2005; Coppola et al., 2009; Aravena et al., 2020b). This is evident for example in partially submerged calderas, in which the style of activity of vents in the submerged zones is clearly influenced by their position either increasing the explosivity by magma-water interaction, or reducing explosivity in deep water conditions (Tonini et al., 2015; Sandri et al., 2018; Paris et al., 2019). Thus, the spatial distribution of the expected style of activity could be strongly relevant when vent opening maps are used to produce long-term hazard maps for different eruption scenarios.

In this context, in order to deal simultaneously with the uncertainty associated with vent position, eruptive style and their interdependence, here we present *thematic* vent opening maps that consider separately the occurrence of four selected volcanic

phenomena: lava emission, small-scale pyroclastic density currents, ejection of ballistic projectiles, and low-intensity pyroclastic fallout. The input geological information adopted for the construction of these maps includes the mapping of the main fault structures, the position of past eruptive vents, and various information on the eruption styles and processes. This is deduced from a critical analysis of literature, the study of volcanic deposits, and from remote analysis of their morphological
characteristics. The resulting thematic probability maps include uncertainty quantification both on past vents locations and on model parameters. To illustrate the effects of using thematic vent opening maps instead of ordinary non-thematic maps, we show a systematic application of the recently developed branching energy cone model (Aravena et al., 2020a), which is able to describe the invasion area of pyroclastic density currents considering channelization processes.

In this investigation, done in the ambience of the RIESCA project, funded by the Italian Development Cooperation and aimed
at promoting applied training in risk scenarios in Central America, we construct thematic vent opening maps for two volcanic complexes of the Central America Volcanic Arc: San Salvador Volcanic Complex (SSVC, El Salvador; Figure 1a) and Nejapa-Chiltepe Volcanic Complex (NCVC, Nicaragua; Figure 1b). SSVC and NCVC are both associated with the subduction of the Cocos Plate beneath the Caribbean Plate at an average rate of 70-90 mm/year (Barckhausen et al., 2001; DeMets, 2001) and pose particularly critical volcanic risks due to the presence of densely inhabited communities in the surrounding areas,
including the metropolitan areas of San Salvador (~2.4M people) and Managua (~1.4M people), respectively (Figure 1a-b). This implies that any future volcanic activity could threaten the life of hundreds of thousands of people. Thus, in addition to be an interesting application of our new methodology, the availability of vent opening maps becomes relevant for improving hazard assessment in these regions.

In this paper, after the description of the geological framework of SSVC and NCVC (Sect. 2), we present new thematic vent
opening maps for the two studied volcanic complexes (Sect. 3). Then, we present invasion probability maps of small-scale pyroclastic density currents (PDC), which are derived from the application of the branching energy cone model (program ECMapProb) and the adoption of the thematic vent opening maps described previously (Sect. 4). Finally, Sect. 5 and Sect. 6 present the discussion and conclusions of this study.

## 2 Geological framework and volcanological dataset

We performed a detailed literature analysis in order to collect volcanological data and structural information on SSVC and NCVC (Figure 1a-b).

First, we collected information about past vent locations and their uncertainty areas, which were defined using polygons (Figure 1a-b). Inside these polygons the probability is distributed uniformly. Small polygons are associated with well constrained vent locations (e.g., by the existence of a well-preserved crater), while large polygons are related to significant uncertainty in vent
position. Figure 1c-d presents the eruptive sequence of the two volcanic complexes addressed in this work, where yellow symbols represent their well-documented eruptions (see Tables 1 and 2) and black symbols represent regional stratigraphic markers. In this figure, eruption numbers refer to the source vents (see Fig. 1a-b and Tables 1 and 2) while the letters are used

to identify different eruptions from a single vent. In both the case studies there are many monogenetic vents and only one polygenetic vent, i.e. San Salvador - Boquerón volcano and Apoyeque volcano, respectively.

Then, we considered the major fault structures present in the studied volcanic areas (Figure 1a-b), which likely had a significant influence in the vent position of past events because of their effect in magma ascent dynamics (Gaffney and Damjanac, 2006; Valentine and Krogh, 2006; Ferrés et al., 2011; Avellán et al., 2012; Le Corvec et al., 2013).

Finally, we considered the range of eruptive phenomena associated with past events, mainly based on the nature of the different volcanic edifices and deposits, and on the available volcanological literature (see Tables 1 and 2). In this way, for each vent,

we defined one of four probability levels, i.e. Y = 1, PY = 0.75, PN = 0.25, and N = 0, associated with the occurrence of four different eruptive phenomena: (1) lava emission, (2) small-scale pyroclastic density currents associated with phreatomagmatic activity or low-intensity magmatic volcanism, (3) ejection of ballistic projectiles, and (4) low-intensity pyroclastic fallout due to Strombolian, violent Strombolian or Vulcanian activity (see Tables 1 and 2 and their footnotes). These probability levels are then used to weight the importance of the different vents in the construction of thematic vent opening maps, as described

below. We remark that, although sub-Plinian to Plinian eruptions able to produce large-scale pyroclastic flows and fallout deposits are also present in the geological record of SSVC and NCVC, this eruption type was only associated with the two main central edifices (i.e. San Salvador volcano and Apoyeque volcano, respectively). Thus, the probability of vent opening for these phenomena is largely concentrated on the apical part of these edifices and, consequently, the construction of vent opening maps for these events is not considered in this work.

## 2.1 San Salvador Volcanic Complex (SSVC)


The Pleistocene-Holocene San Salvador Volcanic Complex (SSVC, El Salvador; Figure 1a) sits in a E-W trending graben and is formed by a summit cone called Boquerón Volcano (BV) enclosed by the remnants of the ancient volcanic edifice of San Salvador Volcano (SSV), and by at least 25 monogenetic volcanic edifices located at the SE, NW, and N of the main edifice (Figure 1a). The monogenetic edifices include scoria cones, maars, tuff rings, tuff cones, and explosion craters. Their positions

are strongly influenced by two NW-trending normal faults (Figure 1a). In this work, following Ferrés et al. (2013) and Ferrés (2014), these structures are referred as faults A (i.e., the N40W-trending fault) and B (i.e., the N65W-trending fault; see Figure 1a).

Ferrés et al. (2013) and Ferrés (2014) identified three main periods in the eruptive history of SSVC:

(a) Stage I (>72 ka – 36 ka) corresponds to the construction and collapse of SSV. It is represented by pyroclastic deposits

intercalated with andesitic and basaltic andesitic lavas, and fallout sequences associated with the Coatepeque caldera (Kutterolf et al., 2008; Ferrés et al., 2011). El Picacho (~1960 m a.s.l.), El Jabalí (~1400 m a.s.l.) and the SW portion of the current volcanic edifice are the present remnants of the collapsed and deeply eroded SSV (Fig. 1a; Ferrés, 2014). No monogenetic centers have been identified during this stage (Ferrés, 2014). The collapse of the SSV was a consequence of a phreato-Plinian eruption responsible of the emplacement of the pyroclastic sequence G1 (event 25a in the nomenclature adopted in Fig. 1c;

Sofield, 1998).

**(b) Stage II (36 ka – 3 ka)** represents the construction of the active Boquerón Volcano (volume of circa 8.5 km$^3$; Ferrés, 2014). It includes both effusive and explosive volcanism associated with the emission of basaltic andesitic and andesitic magmas, intercalated with pyroclastic deposits from the Ilopango caldera (Kutterolf et al., 2008; Ferrés et al., 2011; Smith et al., 2020). During this period, lava flows were issued from the SSV (in particular, fourteen lava flows were recognized by Fairbrothers et al., 1978), while the explosive activity included at least six events after the pyroclastic sequence G1 (events 25b-g in the nomenclature of Fig. 1c, the last one marked the limit between Stages II and III; Ferrés et al., 2013).

**(c) Stage III (3 ka – present)** includes eruptive activity mainly concentrated in the flanks of the BV and in the plain nearby. Sofield (1998) suggests that this shift was related to the reaching of the critical height of the younger edifice (i.e., BV). Several monogenetic edifices were formed during this period as a consequence of both explosive and effusive volcanism, which frequently involved magma interaction with external water, such as Crater La Escondida and Loma Caldera (Table 1). In any case, no hydromagmatic explosions have occurred in the area in the last 1000 years (after Talpetate I, see Fig. 1c; Sofield, 1998), possibly as a consequence of a reduction in the level of groundwater in the flanks of the volcano.

Ferrés (2014) suggests that the enlargement of the volcanic field during Stage III is associated with a general decrease of activity in the Central Crater. It is important to note that, although flank activity started approximately 10 ka ago, it has been dominant only during the last 3 ka, exhibiting a strong structural control (Fig. 1a). On the other hand, during this period, BV volcanic activity has included at least three volcanic events: Talpetate I (San Andres Tuff, which includes fallout and surge deposits widely distributed toward SW, event 25h in the nomenclature of Fig. 1c), Talpetate II (event 25i in the nomenclature of Fig. 1c), and the last eruption (i.e., A.D. 1917 event, vent 29; Fig. 1c and Table 1), when minor explosive activity in the central crater, associated with the construction of the intra-crateric Boqueroncito tuff ring, coexisted with an important lava effusion from parasitic vents of the N flank (vents 27 and 28 in our nomenclature, Fig. 1c and Table 1).

Sofield (2004) proposed five different eruption scenarios: (1) hydromagmatic flank eruption of VEI 1-3, (2) monogenetic magmatic eruption of VEI 1-3, (3) small-scale eruption of VEI 1-3 within Boquerón crater, (4) sub-Plinian eruption from the central vent (VEI 4-5) and (5) Plinian eruption from the central vent (VEI 6). Major et al. (2001) also described plausible scenarios associated with lahar occurrence. A first hazard assessment exercise for BV was done by Ferrés et al. (2013), considering ash fall, ballistic projectiles, and pyroclastic density currents issued from the central vent. However, since a recurrence period of 85±50 years was proposed by Sofield (1998) and Sofield (2004) for flank eruptions during the Stage III, the most probable future event is associated with monogenetic volcanism, and thus the analysis of the hazard associated with flank eruptions is of paramount importance in SSVC in order to complement the recent literature.

In this study, we focused on the last 36 ka of volcanic activity at SSVC (i.e. our analysis starts with the end of Stage I, when the collapse of SSV occurred). In particular, following Sofield (1998), Ferrés et al. (2011), and Ferrés (2014), we considered 29 vents in the SSVC (Fig. 1a and Table 1), most of them with a monogenetic character (the only exception is vent 25, i.e., the central crater). Mainly considering the edifice type, the deposits characteristics, and the historical activity, for each volcanic vent we defined the probability of occurrence of the four volcanic phenomena studied here (i.e., lava emission, small-scale PDCs, emission of ballistic projectiles, and low-intensity pyroclastic fallout; Table 1). In general terms, small-scale PDCs were

not associated with the activity of scoria cones, while all the four volcanic phenomena considered here have been linked, although with different probability of occurrence, to maars, explosion craters, tuff cones, and tuff rings as well as to the central activity of BV. It is important to highlight that the products of Plinian and sub-Plinian volcanism, which have been produced at the central vent of SSVC, were not considered in the construction of our vent opening maps due to the small uncertainty in vent position for these events.

## 2.2 Nejapa-Chiltepe Volcanic Complex (NCVC)

The Pleistocene-Holocene Nejapa-Chiltepe Volcanic Complex (NCVC; Figure 1b) is located at the western edge of the active Managua graben, Nicaragua. It includes at least 31 volcanic vents mainly emplaced following the Nejapa-Miraflores lineament (NML in Fig. 1b; Espinoza, 2007; Avellán et al., 2012). At its northern limit, NCVC includes the polygenetic Apoyeque stratovolcano, and several monogenetic volcanoes including maars, tuff rings, tuff cones, and scoria cones (Table 2, Avellán

et al., 2012). One of these vents, named Tiscapa maar (vent 26 in the nomenclature presented in Fig. 1d and Table 2) and whose emplacement was controlled by the seismically active Tiscapa fault, is located inside the city of Managua (Ward et al., 1974; Freundt et al., 2010; Freundt and Kutterolf, 2019).

NCVC deposits are intercalated with pyroclastic deposits derived from Apoyo and Masaya calderas, which represent useful stratigraphic markers (Figure 1d; Kutterolf et al., 2008). Apoyeque products are mainly dacitic to rhyolitic in composition,

while the products of monogenetic vents range from basaltic to andesitic basaltic (Avellán et al., 2012). Most of the eruptions associated with NCVC involved hydromagmatic activity, possibly triggered by the interaction between rising magma and shallow aquifers (Avellán et al., 2012). The possible presence of sources of surface water is so of primary importance for volcanic hazard assessment in this area.

No clear trends are recognized in the temporal evolution of vent position and eruption style, even if important climate variations

have been documented in this zone during the Holocene (Freundt et al., 2010). In any case, it is important to highlight that phreatomagmatic activity in central and southern NCVC has been dominantly observed relatively near the Nejapa fault (e.g., El Plomo craters, Ticomo craters, Refineria crater), while scoria cones are preferentially observed in the peripheral zone (e.g., Motastepe, Altos de Ticomo, San Patricio). A detailed description of the eruptive history of NCVC, particularly of its central and southern parts, is presented in Avellán et al. (2012), while Pardo et al. (2008) and Freundt et al. (2010) focus on the

youngest eruptions of NCVC, which created the Asososca and Tiscapa maars. Details on Plinian eruptions of Apoyeque volcano are present in Kutterolf et al. (2011) and Avellán et al. (2014). A previous hazard assessment exercise for NCVC was done by Connor et al. (2019) by adopting an elliptical kernel density estimator. However, these maps were based on 28 past vent locations, did not consider uncertainty in vent locations, and did not incorporate fault structures neither the information on the past eruptive styles. The main differences between our approach and theirs are briefly discussed in Sect. 5.

Based on Pardo et al. (2008), Freundt et al. (2010), Avellán et al. (2012), and Avellán et al. (2014), we considered 31 vents in the NCVC (Fig. 1b and Table 2). Also in this case, for each volcanic vent we defined the probability of occurrence of the four volcanic phenomena studied in this work, mainly based on the edifice type, the characteristics of the documented deposits, and

the current presence of surface water, which may induce an eruption dynamics controlled by magma-water interaction processes (Table 2). In general terms, scoria cones were mainly related to the production of small-volume fallout deposits, ejection of ballistic projectiles and the emission of lavas. On the other hand, the hydromagmatic activity typical of maars, tuff cones, and tuff rings has been linked to the emission of ballistic projectiles and generation of relatively small PDCs and fallout deposits; while lava flow activity is generally absent at this type of edifices. Finally, in Apoyeque volcano (i.e., vent 31, Fig. 1d and Table 2), we consider that the typical volcanism is characterized by the emission of ballistic projectiles and the eventual generation of relatively small volumes of fallout deposits and small-scale PDCs. We remark that the analysis of large-scale explosive eruptions, which have characterized the activity of Apoyeque volcano, is beyond the objective of this work and thus they were not considered in the construction of our vent opening maps. We choose to consider the volcanic activity in Apoyeque volcano and in other zones of this volcanic system in a common framework to assess volcanic hazard. Although volcanoes of the Nejapa-Miraflores lineament and those of the Chiltepe peninsula have different style and magma composition, it is undoubtful that their activity was strongly intefingered in the recent past (Kutterolf et al., 2007), and that the tectonic structures controlling these volcanoes are strictly interrelated. It is worth noting that the influence of this assumption in the analysis of small-scale events is limited because of the restricted influence of a single vent within the entire volcanic system, and because the weight assigned to the different volcanic phenomena at Apoyeque caldera is null or small in most of the cases (for three of the four considered volcanic phenomena).

## 3 Probability maps of vent opening

### 3.1 Methods

In both the case studies of SSVC and NCVC, we adopt the kernel-based multi-model approach developed in Bevilacqua et al. (2015), Bevilacqua et al. (2017a) and Tadini et al. (2017b). These are long-term assessments based on the record of past eruptions and mapped faults, which lie on the assumption that a new vent will likely open close to previous vents and to geological structures that, in the past, have favoured the ascent of magma to the surface.

We linearly integrate two models named *Model 1* and *Model 2*, with weights affected by uncertainty (Fig. 2). *Model 1* considers the faults and the position of past vents following a Bayesian approach, and lies in the assumption that new vents will likely occur near structures that interacted with ascending magmas in the past (Martin et al., 2004; Jaquet et al., 2012). On the other hand, *Model 2* adopts a Gaussian kernel density estimator applied on a uniform distribution within the uncertainty areas enclosing the past vents (Tadini et al., 2017b). This model does not consider structures but only a measure of the expected distance between past and future vent positions. We remark that in both the models past vents do not comprise simple points, but areas of uncertainty of different extent. Each area can cover several cells of our 100-m resolution computational grid, some of them completely, others only partially. Therefore, for each cell, it was taken into account the fraction of each uncertain area that it contains (Bevilacqua et al., 2015; Tadini et al., 2017a). Both models are reviewed in the Appendix A. A key novelty of this study is the fact that we weight the past vents differently according to the chance that the new vents related to them will

be the source of a specified hazardous phenomenon, based on the past eruptions locally occurred. The thematic weights, which lead to the production of thematic maps, are reported in Tables 1 and 2.

    The location of the next eruptive vent is modelled as a random variable X with a continuous probability density function (PDF) in the spatial domain. The vent opening map is then displayed as the probability density of the variable X per km$^2$. We remark that the probabilities of vent opening that we obtain are conditioned on the occurrence of a new eruption and there is no

associated temporal window. We did not count the polygenetic vents multiple times in these models. In Sect. 5 we discuss this assumption and we test the opposite choice.

    We remark that our estimates are doubly stochastic (e.g., Cox and Isham, 1980; Daley and Vere-Jones, 2003; 2008; Jaquet et al., 2008; 2012; 2017), which means that the statistical distribution of the location of the next eruptive vent is represented using ill-constrained parameters that are treated as uniformly distributed random variables (e.g., Bevilacqua, 2016). These parameters

are two distance values $d_1$ and $d_2$ tuning the kernel bandwidth of the two models, and two probability values $p_1$ and $p_2$ tuning the probabilistic relevance of *Model 1* compared to *Model 2* and of the mapped faults compared to unknown structures, respectively. As a consequence of this approach, the PDF values will have their own confidence intervals.

    In particular, the two map layers associated with different conceptual models (*Model 1* and *Model 2*) are linearly combined with specific weights, $p_1$ and $(1 - p_1)$, for the development of a multi-model vent opening map (Fig. 2). A similar approach

is adopted inside *Model 1* to define the prior fault map as the combination of two layers, weighted $p_2$ and $(1 - p_2)$. One layer, $z_1$, is related to the mapped faults (i.e. fault outcrops), and the other layer, $z_2$, is a uniform PDF representing the unknown (i.e. buried) faults. Figure 2a illustrates the logic of this vent opening model combination. Figure 2b shows the two models and their differences when applied to the test example of a past vent near a fault.

    The input parameter $p_1 = \text{Unif}(0.75,1)$ was constrained through a straightforward expert judgment after the assumption that

*Model 1* should have a greater relevance than *Model 2*. This is because both SSVC and NCVC are characterized by robust geological information which suggests significant relationships between past vents, fault structures and volcanism, with several aligned vents along the main faults. Similarly, $p_2 = \text{Unif}(0.75,1)$ was chosen through expert judgment due to the greater relevance of mapped faults compared to unknown structures that are assumed to be uniformly distributed in the region. Thus, the vent opening maps produced here are dominated by *Model 1* and by the information of mapped faults, but the possibility

of a new vent not influenced by structural alignment or by an unknown fault is considered. Further research focused on a deeper understanding of the two case studies of SSVC and NCVC could improve these constraints, either through more structured expert judgment techniques (Bevilacqua et al., 2015; Tadini et al., 2017a), likelihood based techniques (Bevilacqua et al., 2017a; 2018) and/or through other geophysical models able to address regional volcanism (e.g. Martin et al., 2004; Jaquet et al., 2012; Runge et al., 2016; Deng et al., 2017).

On the other hand, our multi-model approach depends on two additional parameters: $d_1$ and $d_2$ (see Appendix A). The parameter $d_1$ in *Model 1* is the average distance between sub-parallel regional faults. Different distances are measured in different sub-regions, thus defining the uncertainty range of $d_1$. Specifically, $d_1 = \text{Unif}(5,10)$ km for both SSVC and NCVC. The parameter $d_2$ in *Model 2* is the average of the distance of the $k$-th nearest neighbor of each past vent. The number $k$ is

varied in [1,3] to consider the presence of spatial clusters, thus defining the uncertainty range of $d_2$. Specifically, $d_2 =$
Unif(1,2.5) km for SSVC and $d_2 =$ Unif(1,2) km for NCVC.

In summary, if $g(\mathbf{x})$ is the PDF of *Model 1* and $f(\mathbf{x})$ is the PDF of *Model 2*, then the multi-model "integrated" vent opening map is given by:

$$F(\mathbf{x}) = p_1 g(\mathbf{x}) + (1 - p_1)f(\mathbf{x}) , \qquad (1)$$

that is:

$$F(\mathbf{x}; d_1, d_2) = p_1[p_2 g(\mathbf{x}; z_1, d_1) + (1 - p_2)g(\mathbf{x}; z_2, d_1)] + (1 - p_1)f(\mathbf{x}; d_2) , \qquad (2)$$

where we also expressed the dependence of $g$ on the mapped faults $z_1$, the uniformly distributed map of unknown faults $z_2$, and the distance parameter $d_1$. We also included the dependence of $f$ on the distance parameter $d_2$.

### 3.2 Results

### 3.2.1 San Salvador Volcanic Complex

Figures 3 and 4 present the resulting vent opening map associated with SSVC for two of the four hazardous volcanic phenomena considered in this work: lava emission and small-scale PDCs, respectively. The results related to the emission of ballistic projectiles and the generation of small-scale fallout deposits are described in Figures S1 and S2 in the supplementary material. Each figure presents the mean value of the computed probability distribution and the results associated with the 5[th] percentile and the 95[th] percentile, expressed in terms of probability density per km$^2$, in percentage. For comparison purposes,
Figure 5a presents the results associated with the mean value of a non-thematic vent opening map constructed using the same procedure but with all the vents adopting the same weight. Please note that, in practice, this map is equivalent to the vent opening map of ballistic projectiles (Fig. S1). In general, we remark that if almost all past eruptions are characterized by a common volcanic hazardous phenomenon (e.g. the small scale fallout), the corresponding thematic map will be very similar to the non-thematic map (Fig. S2).

Results exhibit significant differences between the various thematic maps. While the highest vent opening probability associated with small-scale PDCs follows the N65W-trending fault B (Figure 4), in the other thematic maps the highest vent opening probabilities tend to locate along the northern portion of the N40W-trending fault A (Figs. 3, S1 and S2).

In particular, in the case of events able to produce lava flows (Fig. 3), the maximum probability density is located on the NW flank of San Salvador volcano along the fault A, reaching probabilities of up to 1.0% per km$^2$ in the mean value map, with
90% confidence interval [0.7%, 1.6%] per km$^2$. Considering the mean value map, the total probability of vent opening in a 4 km wide belt across the northern portion of fault A is 39.3%, excluding the pixels that are closer to fault B than to fault A. On the other hand, the total probability of vent opening close to fault B is 13.9%. This value was also computed by considering a 4 km wide belt across the fault under examination. Please note that this criterion is also used in the analysis of the other thematic maps.

Instead, as mentioned above, the region of maximum probability of vent opening conditioned on the occurrence of an eruption able to produce small-scale PDCs is located along the N65W-trending fault B (Fig. 4), with a peak probability in the mean value map of 1.1% per km$^2$, with 90% confidence interval [0.8%, 1.6%] per km$^2$. In this case, in fact, the total probability of vent opening near the fault B (i.e. 23.5%) is significantly higher than the results observed in the thematic map of lava emission. Instead, the probabilities of vent opening along the northern and southern portions of fault A are 26.8% and 6.1%, respectively.

The thematic maps associated with ballistic projectiles (Fig. S1) are quite similar to the case of lava emission, with the maximum values observed along fault A. These maps show a peak probability density of 0.9% per km$^2$ in the mean value map, with 90% confidence interval [0.7%, 1.3%] per km$^2$. The probabilities of vent opening near the northern portion of fault A, the southern portion of fault A and fault B are 35.0%, 7.1% and 16.7%, respectively.

Finally, the vent opening maps related to eruptions able to produce small-scale fallout deposits (Fig. S2) are similar to those
presented for lava emission and ballistic projectiles. The maximum probability of vent opening, located at the NW flank of the volcano, is 0.9% per km$^2$, with 90% confidence interval [0.7%, 1.2%] per km$^2$. In this case, the probabilities of vent opening close to the northern portion of fault A, the southern part of fault A and fault B are 35.1%, 6.5% and 16.3%, respectively. Results are summarized in Table 3.

### 3.2.2 Nejapa-Chiltepe Volcanic Complex

The thematic vent opening maps associated with NCVC are presented in Figures 6 (lava flows), 7 (small-scale PDCs), S3 (ballistic projectiles), and S4 (small-scale fallout pyroclastic deposits). Again, each figure presents the mean value of probability per km$^2$, in percentage, including the results of the 5[th] percentile and the 95[th] percentile as well. Instead, Figure 5b shows the mean value of a non-thematic vent opening map, i.e. with no differences in the weight of the different vents. Also in this case, this map is equivalent to the vent opening map of ballistic projectiles (Fig. S3).

All the maps show the maximum probability near Asososca maar (vent number 13 in Fig. 1b,d and Table 2), and the only significant differences are related to the N-S extent of the high-probability zone, which tends to be longer in the maps associated with the occurrence of small-scale PDCs (Fig. 7). A secondary peak is observed near Miraflores scoria cone.

In particular, the peak value of vent opening probability density in the mean value map related to the emission of lavas is 1.6% per km$^2$, with 90% confidence interval [1.0%, 2.3%] per km$^2$ (Fig. 6). The maximum values in the other maps are: 1.7% per
km$^2$ for events able to produce small-scale PDCs, with 90% confidence interval [1.2%, 2.5%] per km$^2$ (Fig. 7); 1.6% per km$^2$ for ballistic projectiles, with 90% confidence interval [1.1%, 2.2%] per km$^2$ (Fig. S3); and 1.6% per km$^2$ for eruptions that produce small-scale fallout deposits, with 90% confidence interval [1.1%, 2.3%] per km$^2$ (Fig. S4). The resulting probabilities of vent opening inside the limits of Managua are 28.9%, 35.1%, 31.2% and 32.0% for events able to produce lava flows, small-scale PDCs, ballistic projectiles and small-scale fallout deposits, respectively. Also these results are summarized in Table 3.

The previous hazard assessment exercise in Connor et al., (2019) produced spatial density maps which differ from our results. As said above, they did not associate any uncertainty area to the past vent locations, they considered slightly less events, and they did not use the map of the main fault structures. Moreover, their method is not doubly stochastic, thus they did not quantify

the uncertainty affecting their results. In other words, their spatial map is not a PDF but a spatial density. However, once divided by the number of past vents considered, it becomes a vent opening map. Their maximum probability values are ~2.8-

3.2% per km$^2$ and shifted of ~1 km towards west compared to ours. Their region above 1% probability per km$^2$ has a similar areal extent to ours, but is significantly stretched in the N-S direction, due to the use of an elliptical kernel. Further comparison would be needed to fully evaluate the strengths and weaknesses of both approaches.

## 4 Numerical simulation of small-scale PDC invasion hazard maps

For both SSVC and NCVC we extracted 100 sets of 1024 samples of vent position by using the thematic vent opening maps

associated with small-scale PDCs. Each set of 1024 vent positions is associated with one of 100 different arrays of the model parameters $(p_1, p_2, d_1, d_2)$ adopted in the construction of our thematic vent opening maps. The sampling method is based on a Latin Hypercube Sampling scheme, generalized with Orthogonal Arrays to increase space-filling properties (e.g., Bevilacqua et al., 2019b; Patra et al., 2020). We modified the method to work under the assumption of a non-uniform PDF, as detailed in Appendix A. In particular, our set of PDC initiation points changes for each of the 100 LHS designs explored (Rutarindwa et

al., 2019), and we did not rely on a number of fixed grid cells as initiation points (e.g. Sandri et al., 2018; Clarke et al., 2020). The samples are denser where vent opening probability is higher, and their total number is great enough not to leave any region free of testing.

Through the use of the branching energy cone model, implemented in the program ECMapProb and described in Aravena et al. (2020a), we performed these 100 x 1024 simulations (i.e., 102,400 simulations) for each volcanic complex. Despite the

branching energy cone model (Aravena et al., 2020a) is able to consider channelization processes in the construction of PDC invasion maps, the topographies of both volcanic systems do not present significant channelization zones. Under these conditions, the branching energy cone model tends to present similar results to the traditional formulation, although not equal. The reason for running different sets of simulations for each volcanic system is to investigate the propagation of the uncertainty associated with our vent opening maps in the resulting probability maps of PDC inundation. We fixed the other PDC initial

conditions. In particular, we assumed $H_c = 500$ m and $\tan(\varphi) = 0.25$ (i.e. ~14°). These input conditions are consistent with the runout distances observed in the small phreatomagmatic deposits recognized in the field (i.e. of the order of 2 kilometres), e.g., Loma Caldera in SSVC, and are assumed to be representative of the studied PDCs. In this study we focus on gravity driven PDCs, but we remark that small scale phreatomagmatic eruptions may also produce dominantly inertial dilute fully-turbulent density currents, whose dynamics is better replicated by the so-called box-model approach (Huppert and Simpson

1980). We decided not to use variable input conditions for initial PDC characteristics, so our hazard assessment is only valid in this specific scenario of PDC size and friction angle (for examples of friction angle variability in studies based on the energy cone model, see Hayashi and Self, 1992; Sheridan and Macias, 1995; Tierz et al., 2016b; Sandri et al., 2018). A more complete PDC hazard assessment considering variable size and friction properties would require additional information to properly calibrate these input parameters (e.g., Cioni et al., 2020) and/or complementary data coming from analogue volcanoes (e.g.

Tierz et al., 2016a; Clarke et al., 2020). Further analysis could investigate the sensibility of numerical results on these parameters, and increase the number of inputs that are sampled in the LHS scheme, also considering correlations between PDC size and friction properties, but this is beyond the purpose of this study (e.g., Spiller et al., 2014; Ogburn et al., 2016; Ogburn and Calder, 2017; Tierz et al., 2018; Rutarindwa et al., 2019; Patra et al., 2020).

Figures 8 and 9 present the invasion probability of small-scale PDCs at SSVC and NCVC, derived from the systematic

application of the branching energy cone model (Aravena et al., 2020a) and the adoption of the thematic vent opening maps shown in Figures 4 and 7. Numerical simulations were performed using a DEM at 30 m resolution. PDC invasion probabilities are described in percentage, including the mean value at each pixel of the map, the 5$^{th}$ percentile and the 95$^{th}$ percentile. These results derive from the coupled effect of the vent opening probability distribution, controlled by structures and past vents, and the transport dynamics of PDCs, controlled by volcano topography and the characteristics of the branching energy cone model.

In the case of SSVC, results show a zone of high PDC invasion probability located at NW of San Salvador volcano, with a maximum value of 23%, with 90% confidence interval [19%, 29%] and high invasion probabilities near the cities of Nuevo Sitio del Niño and Lourdes (Fig. 8). Modelled invasion probability at the SE flank of San Salvador volcano and in the city of San Salvador tends to be low, where a significant effect is exerted by the topographic barrier of Cerro El Picacho, as well as of the entire volcanic edifice of SSVC (Fig. 8). This is because vent-opening probabilities are high on the NW flank of the

volcano and on the surrounding plain to the N-NW of the edifice, while significantly lower probabilities of vent opening were computed on the E flank of the volcano (Fig. 8). The highest PDC invasion probability calculated at the metropolitan area of San Salvador, ~2-3%, is located in its western sector, i.e. between the western sector of San Salvador and Santa Tecla. These values might be explained by the peak in vent-opening probability shown in Figure 4 and/or by the high slope of the main edifice and the absence of significant topographic barriers on the southern sector of the SSVC edifice.

Instead, in the case of NCVC, numerical results indicate the highest invasion probabilities along the Nejapa fault, near the western border of Managua city and inside it. The maximum values of invasion probability, i.e. 22% and 23% in the mean value map, are associated with the low-elevation areas of Asososca maar and Nejapa maar, respectively (Fig. 9). Non-negligible invasion probabilities have been calculated at the eastern portion of Ciudad Sandino as well, while the invasion probability of small-scale PDCs at the northern part of the studied zone, i.e. near Apoyeque volcano, tends to be low (Fig. 9).

Finally, for the two volcanic complexes addressed here, Figure 10 presents the difference between the mean PDC invasion probability percentages obtained by adopting thematic vent opening maps (i.e. Figs. 8a and 9a) and the equivalent results that would be derived from the application of non-thematic vent opening maps, i.e. by using the maps displayed in Fig. 5. These results show clearly the relevance of using thematic vent opening maps in the assessment of hazard at volcanoes where eruptive style may significantly change with vent location. Positive values (i.e. red zones) represent the portions of the map where the

application of thematic vent opening maps implies an increase in the calculated PDC invasion probability, while negative values (i.e. blue zones) are the sectors where the use of thematic vent opening maps instead of the traditional ones is manifested in a reduction of the computed PDC invasion probability. SSVC exhibits differences ranging from -7.2%, on the N-NNE flank of Boquerón Volcano (BV), south of Quezaltepeque, to +5.8% on the NW flank of BV, near Lourdes (Fig. 10a). These results

involve dramatic modifications in the computed PDC invasion probability. In fact, at the N-NNE flank of BV, the PDC invasion probability is halved when thematic vent opening maps are considered, while probability increases by 70% in some portions of the NW flank of BV. On the other hand, the computed differences for NCVC are moderate and vary from -2.9% in the northern portion of NCVC to +3.2% in the southern portion of this volcanic field, including a significant area that belongs to the city of Managua. In any case, in relative terms the differences between the results derived from the application of thematic and non-thematic maps are smaller both in Managua and in its surroundings.

## 5 Discussion

Several probabilistic assessments of vent opening at different volcanoes have been presented during the last decade (e.g., Marzocchi and Bebbington, 2012; Connor et al., 2015; Poland and Anderson, 2020), improving the methodologies implemented to construct these maps and sensibly increasing the number of volcanic systems for which a probabilistic assessment of vent opening is available. It is remarkable, for example, the inclusion of structured expert judgment procedures to constrain the input parameters of the models adopted to construct these maps (Chapman et al., 2012; Bevilacqua et al., 2015; Tadini et al., 2017a; Bebbington et al., 2018), and their coupling with models aimed at describing the dispersal of volcanic products (e.g., Del Negro et al., 2013; Neri et al., 2015; Thompson et al., 2015; Tierz et al., 2018; Gallant et al., 2018; Hyman et al., 2019; Rutarindwa et al., 2019). A growing effort is aimed at the production of short-term vent opening maps which can modify the long-term estimates after plugging-in the monitoring information that progressively evolves during volcanic unrest (e.g., Sandri et al., 2012; Chaussard and Amelung, 2012; Selva et al., 2014; Bevilacqua et al., 2019c; 2020a; 2020b; Patra et al., 2019; Sandri et al., 2020).

In this context, in this work we adopted a novel approach where we have considered separately different volcanic hazardous phenomena (i.e. emission of lava flows, small-scale PDCs, ballistic projectiles, and small-scale pyroclastic fallout). This led to the construction of thematic vent opening maps which allow to assess the hazard that different volcanic phenomena involve separately. We remark that previous studies already produced vent opening maps devoted to specific types of eruptions (e.g. Plinian and sub-Plinian eruptions in Tadini et al., 2017b; or pumice-cone-forming eruptions in Clarke et al., 2020), to eruptions inside selected sub-regions (Bevilacqua et al., 2017b) or to a suite of pre-imposed eruptive scenarios (Ang et al., 2020). It is worth highlighting that the approach adopted here, where the resulting vent opening maps are conditioned on the occurrence of a specific volcanic phenomenon, is opposite from the strategy adopted in a series of studies of hazard assessment (e.g. Neri et al., 2015; Thompson et al., 2015; Bevilacqua et al., 2017a; Tierz et al., 2020), where the probability of having a specific hazardous phenomenon or a specific eruption size is conditioned on vent location. Although both alternatives can be useful for hazard assessment, their suitability is controlled by the expected application, and the two approaches are linked through the application of the Bayes Theorem. Our approach seems to be more appropriate for the construction of thematic hazard maps for a specific volcanic phenomenon and eruption size, while the opposite strategy, possibly coupled with event trees, seems to suit better in the formulation of hazard maps that combine multiple phenomena and/or multiple eruption sizes.

In order to apply our approach, in addition to the mapped faults and the record of past eruptions, we considered the eruption style and the likely eruptive phenomena that the past eruptions could have produced, on the basis of deposits and morphological features of volcanic edifices. This is justified by the suggested relation of vent position and eruption style and intensity, which has been observed at different volcanic fields (e.g., Andronico et al., 2005; Coppola et al., 2009; Sigmundsson et al., 2010). In particular for the studied volcanic fields, at SSVC, small-scale PDCs have been preferentially produced by vents located along the N65W-trending fault B (Fig. 1a and Table 1), while lava flows have been commonly observed in vents located along the N40W-trending fault A (Fig. 1a and Table 1). Apparently, no small-scale PDCs have been issued in the past from the central vent of the present Boquerón volcano. On the other hand, at NCVC, maars and tuff rings have been mainly produced in events whose vent was located near the Nejapa fault, while scoria cones are predominant in the peripheral zones. Additional evidence is reported also for the possible presence of hydromagmatic eruptive centers located in the south-western corner of Xolotlán Lake.

As expected, the integration of this information in the construction of thematic vent opening maps is manifested in their results (Figures 3, 4, 6, 7 and S1-S4), which in some cases present significant differences as a function of the considered hazardous phenomenon. We also estimated how these changes are sensible to some of the main sources of uncertainty. In fact, the 90% confidence interval of the vent opening probability per $km^2$ can strongly enhance the differences between thematic maps as can be observed, for example, in Figures 3c and 4c.

The design and implementation of mitigation strategies of volcanic risk are deeply associated with the characteristics of the volcanic process under consideration and vent position. In this sense, the importance of using thematic maps when the studied volcanic complex shows a vent position-controlled eruption style has been illustrated by modeling the propagation dynamics of small-scale PDCs. Our results, which were obtained by adopting the branching energy cone model (Aravena et al., 2020a), show the coupled effect of vent opening maps and volcano topography in determining the PDC invasion hazard. Results stress that the adoption of thematic vent opening maps is able to produce significant effects in the construction of hazard maps, which are particularly relevant for SSVC.

Finally, an important aspect in our analysis is that we did not count multiple times the polygenetic vents (San Salvador - Boquerón volcano in SSVC, and Apoyeque volcano in NCVC), i.e. every vent was weighted only in consideration of its past eruptive phenomena and not the number of eruptions. Similarly, in Connor et al. (2019) the Apoyeque volcano counted one in the production of their spatial density maps. In contrast, a straightforward event-counting approach could lead to different results, and this fact deserves some additional comments.

In the case of SSVC we counted nine explosive events during the last 36 ka, including the collapse of SSV at the end of Stage I: six eruptions during Stage II, and three during Stage III (Figure 1c). Moreover, in Stage II at least fourteen lava flows were issued from the SSV (Fairbrothers et al., 1978). In the NCVC we counted three explosive eruptions from Apoyeque, and no lava flows during the last 30 ka (Fig. 1d). In our results we did not follow an event counting approach because of several reasons explained below.

Firstly, an event counting approach would rely on the assumption that the volcanic system is stationary over the time period considered in the statistics. In contrast, for example, SSVC apparently shifted from a central volcanism to the development of the monogenetic field on the flanks of the BV and in the plain nearby 3 ka ago. Most of the twenty-eight known monogenetic events occurred after 3 ka BP, while only three of the BV eruptions are associated with this period. In addition, how much the Central Crater should count depends on the time interval considered. In a similar situation, Bevilacqua et al. (2017a) treated the uncertainty related to a shift of volcanic activity from Long Valley Caldera to Mono-Inyo chain by weighting the events in the older site from 0% to 50% of the more recent events. Bevilacqua et al. (2015) weighted differently the vents in the three past epochs of Campi Flegrei post-collapse activity, after an expert elicitation. Again, the older vents tended to weigh less than the more recent vents.

Secondly, in the two case studies of SSVC and NCVC the specific knowledge of many eruptive centers is variable and sometimes poor, under-recording is likely, and a robust event counting is not possible. However, the possible errors in event counting are not the same in the various thematic maps. For example, in the case of small-scale PDCs at SSVC the capability of producing low energy events in the central edifice should not be substantially different from monogenetic vents. In contrast, BV is a stratovolcano built after a large number of lava flows, while monogenetic vents typically produce a single flow.

Figure 11 shows two illustrative examples of modified vent opening maps of SSVC after following an event counting-based approach. We only present the results of SSVC because the potential effect of this modification would be minor in NCVC, due to the relatively small number of known polygenetic events. In particular, Figure 11a shows a vent opening map for lava flow emission that counts the central crater 16.25 times, i.e. 14 times for the past lava flows in Stage II, plus 9 x 0.25 times for the past explosive eruptions during the last 36 ka. Figure 11b shows a vent opening map for ballistics that counts the central crater 9 times, i.e. the past explosive eruptions. Figure S5 and S6 in the Supporting Information show the $5^{th}$ and the $95^{th}$ percentile values of these maps. The significant concentration of vent opening probability towards the polygenetic center is evident in these figures. Conversely, in terms of hazard assessment, our choice (i.e. with no consideration of a higher weight for polygenetic vents) spreads the vent opening probability further from the central volcano and potentially closer to highly inhabited areas. In summary, to improve this aspect of the analysis, separate maps for the central volcano and the surrounding volcanic field could be produced, and their differential weight better constrained after additional research.

**6 Conclusions**

In this study, we have presented vent opening probability maps for two Central American volcanic systems: San Salvador Volcanic Complex and Nejapa-Chiltepe Volcanic Complex. Like many other volcanoes in the world, these volcanic fields present some features that make critical the availability of this tool for the design and implementation of volcanic risk mitigation procedures: (1) they are next to highly inhabited cities, i.e. San Salvador (2.4M people) and Managua (1.4M people), respectively; (2) they present a significant variability in vent position; and (3) they produced a number of different types of eruption style.

We implemented quantitative vent opening probability estimates which are based on main fault structures and past vent positions, and include uncertainty quantification and the type of activity of the documented eruptions. We produced different thematic maps related to emission of lava, small-scale pyroclastic density currents, ballistic projectiles and small-scale fallout deposits.

The main findings derived from the new vent opening maps are:

(a) At SSVC, the maps show their maximum values on the NW flank of San Salvador volcano and in the northern region of this volcanic complex. In particular, results indicate that the zone of high probability of vent opening (>0.5% mean probability per km$^2$) of events able to produce small-scale PDCs follows the N65W-trending fault B, while the high-probability zone related to lava emission, ballistic projectiles and the production of small-volume fallout deposits follows the N40W-trending

fault A. This shows that the separate consideration of different hazardous phenomena is able to produce relevant differences in vent opening probability maps.

(b) At NCVC, results show that the maximum vent opening probability is located near Asososca maar. The only differences between the different thematic maps is the N-S extent of the high-probability zone, which tends to be larger in the maps associated with small-scale PDCs than in the other thematic maps. Importantly, a significant portion of the vent opening

probability distribution is located inside the limits of Managua City, which implies major challenges in managing the volcanic risk associated with NCVC.

We remark that we did not consider the occurrence of large volume pyroclastic deposits derived from sub-Plinian or Plinian eruptions, e.g., large-volume fallout or PDC deposits. Indeed, in the past they were only emitted from the polygenetic central vents of both volcanic complexes, i.e. San Salvador and Apoyeque, respectively. Thus in these cases the construction of vent

opening probability maps is not meaningful at the scale of the other thematic maps, i.e. tens of kilometers.

Finally, we have adopted the branching energy cone model described in Aravena et al. (2020a) to illustrate the use of the thematic vent opening maps presented here. In particular, we performed a Monte Carlo simulation of ~10$^5$ small-scale PDCs, representative of those derived in the past from phreatomagmatic activity or low-intensity magmatic volcanism. We varied the vent location following a doubly stochastic approach that enabled us to estimate the effects of the uncertainty affecting the

515 vent opening PDF. We did not change the size and the friction properties of the flow to better analyse the effects of the different vent opening maps.

Results of this new hazard assessment show that:

(a) At SSVC, a significant effect is exerted by volcanic topography. The highest values of invasion probability are observed at NW of San Salvador volcano, with high invasion probabilities (5-10% in the mean value map) near the cities of Nuevo Sitio

del Niño and Lourdes. Invasion probability in the city of San Salvador tends to be relatively small, with the highest values (2-3% in the mean value map) observed at the western sector of the urbanized zone (i.e., western zone of San Salvador and Santa Tecla). The use of the thematic map increased the small-PDC invasion hazard estimates on the western side of BV, while decreased the same hazard in the northern side of the volcano.

(b) At NCVC, the high vent opening probability inside and near the limits of Managua produces high small-PDC invasion probabilities inside this city (~10-20% in the mean value map). High invasion probabilities (~5-10% in the mean value map) are computed at the eastern portion of Ciudad Sandino as well. The use of the thematic map increased the small-PDC invasion hazard estimates in the southern part of NML, while decreased the same hazard in the northern part.

(c) In both case studies, the examination of the 5[th] and 95[th] percentile maps show that hazard values can change of ca. $\pm30$-50% because of the uncertainty sources affecting the vent opening PDF. The effects of using a thematic map in the small-scale PDC hazard estimates are evident in both SSVC and NCVC, but significantly stronger in the first case. This is enhanced by considering the uncertainty range.

The findings presented in this paper represent relevant information for the management of volcanic risk in these volcanic complexes, and complement previous works focused on the hazard assessment of volcanic activity in the central vents (e.g., Avellán et al., 2012; Ferrés et al., 2013). Coupling of thematic vent opening probability maps with numerical models able to describe other eruptive phenomena (e.g. lava flows and ballistic projectiles) would provide further relevant data for the assessment of volcanic hazard, as shown here for PDCs.

**Appendix A**

In the following we review the details *Model 1* and *Model 2*, and the non-uniform Latin Hypercube Sampling scheme adopted in this work.

**A.1 Model 1: Kernel-based Bayesian update of fault map**

*Model 1* merges the faults and past vents locations following a Bayesian approach. This model is also detailed in Bevilacqua et al. (2017a), but differs in the method to select the distance parameter $d_1$. We assume that the mapped faults provide information in the forecast of future vent locations (e.g., Felpeto et al., 2007; Cappello et al., 2012; 2015). In fact, we expect the new vents to more likely occur near mapped faults that previously interacted with rising magma (Martin et al., 2004; Jaquet et al., 2012). The key idea in *Model 1* is that the vent opening map depends on an additional spatial parameter $\zeta = (\zeta_1, \zeta_2)$ representing the surface projection of a fault interacting with a future potential rising dike. We call $z$ the prior PDF of $\zeta$, i.e. the prior fault map. Namely, $z = z_1$ is the distribution of mapped faults, and $z = z_2$ is the distribution of unknown faults. In particular, $z_1$ is uniform inside a 100 m buffer along the fault outcrop, which is not sensibly affecting the results because it is an order of magnitude lower than $d_1$. Considering that linear averaging is not commutative with the Bayes theorem, we average the posterior probability maps because, in this way, the linear weights directly impact the final results, and they are not altered by the multiplicative step.

For each past vent location $(x_i, y_i)$, we use Bayes theorem to calculate the posterior probability density values $z(\zeta_1, \zeta_2 \,|\, x_i, y_i)$, up to a multiplicative normalization constant $C$ detailed below. This represents the fault locations that may have interacted

with the rise of magma that led to the past vent $(x_i, y_i)$. We integrate past vent locations $(i = 1, ..., N)$ on their uncertainty areas $(D_i)$, according to the expression:

$$z(\zeta|D_i) = C \cdot \int_{D_i} z(\zeta) g(\xi|\zeta) d\xi \, , \tag{3}$$

where $g(\cdot \mid \cdot)$ is a Gaussian likelihood defined below (Eq. 4). We calculate different posterior probabilities, one for each past vent considered $(i = 1, ..., N)$, and we define the global posterior of $\zeta$ as the weighted average of them, with weights $C = w_i$, where $w_i$ is related to the specified hazardous phenomenon (see Tables 1 and 2). Different hazards will lead to different posterior maps.

Then, conditioned on the value of parameter $\zeta$, we define the likelihood for a vent location x as a symmetrical, two dimensional Gaussian function of mean $\zeta$ and covariance matrix $\sigma^2$:

$$g(\mathrm{x}|\zeta) = \frac{1}{2\pi\sigma^2} \exp\left(-\frac{1}{2\sigma^2} \|x - \zeta\|^2\right) \tag{4}$$

This kernel function creates a link between the past vents and the portions of the faults next to them, i.e. $\zeta$. The value of the standard deviation is $\sigma = d_1/2$, where $d_1$ is the average distance between sub-parallel regional faults. In other words, we assume that, on average, about 95% of the vents that are related to a fault will open closer than the next sub-parallel fault. This is a different strategy from that adopted by Bevilacqua et al. (2017a), where the distance bound was obtained through a geometric argument on the maximum depth of the fault-dike interaction and the dip angle of the faults.

Finally, we calculate the vent opening map applying again the kernel function $g(\cdot \mid \cdot)$ to the global posterior probability distribution of $\zeta$. So, the kernel function creates a link between $\zeta$ and the new vents. Thus, the vent opening probability density $g$ according to *Model 1* is defined by:

$$g(x) = \int_{\mathbb{R}^2} g(\mathrm{x}|\zeta) \sum_{j=1}^{2} \frac{m_j}{N} \sum_{i=1}^{N} z_j(\zeta|D_i) d\zeta \tag{5}$$

where $z_j$ with $j = 1,2$ are the two posterior probability density functions of $\zeta$, $m_1 = p_2$, and $m_2 = 1 - p_2$.

**A.2 Model 2: Kernel density estimator**

*Model 2* is a Gaussian kernel density estimation applied on a uniform distribution within the uncertainty areas enclosing the past vents (Bebbington and Cronin, 2011; Connor et al., 2012; Bevilacqua et al., 2015). This method is detailed in Tadini et al. (2017b), but differs in the selection of the distance parameter $d_2$. The kernel function describes the expected distance between past and future eruptive vents. The kernel function can be any positive function $K$ that integrates to one (Weller et al., 2006), and in general, given a finite sample $\mathbf{x}_i = (x_i, y_i)$, a kernel density estimator can be defined as follows:

$$f_h(\mathrm{x}) = \frac{1}{N} \sum_{i=1}^{N} w_i K\left[\frac{\mathrm{x} - \mathbf{x}_i}{h}\right] \tag{6}$$

where $h$ is the bandwidth, and $w_i$ are the vent weights related to the specified hazardous phenomenon. We assume $K$ equal to a two-dimensional radially symmetric Gaussian function, and $h$ is the standard deviation parameter. In this study we consider $h = 2d_2/\pi$, where $d_2$ is the average of the distance of the $k$-th nearest neighbor of each past vent, formally represented by a Rayleigh distribution. This is a different procedure from that adopted by Tadini et al. (2017b), where $k = 1$ and $h = d_2$.

## A.3 Non-uniform Latin Hypercube Sampling

Our sampling technique of the input variables in the PDC simulations is based on the Latin Hypercube Sampling (LHS) idea, and in particular, on the improved space-filling properties of the orthogonal array-based Latin Hypercubes.

The LHS is a well-established procedure for defining pseudo-random designs of samples with good properties with respect to the uniform probability distribution on an hypercube $[0, 1]^d$. In particular, if compared to a random sampling, LHS: (1) enhances the capability to fill the space; (2) avoids the overlapping of point locations projections; and (3) reduces the dependence on dimensionality (Stein, 1987; McKay et al., 2000; Ranjan and Spencer, 2014).

Once the desired number of samples $N$ is selected, [0,1] is divided in $N$ equal bins, then each bin will contain one and only one projection of the samples over every coordinate. There is a large number of possible designs, i.e. the number of permutations of the $N$ bins in the $d$-projections. If these permutations are randomly sampled, clusters of points or regions of void space may be observed. For this reason, we base our design on the orthogonal arrays, which constrain the samples to fill the space with respect to a regular grid at a coarser scale than the LHS bins (Owen, 1992; Tang, 1993; Ai et al., 2016; Patra et al., 2018; 2020; Bevilacqua et al., 2019b).

In this study we further modified a two-dimensional LHS to work under the assumption of a non-uniform PDF, i.e. the vent opening probability map $f(\cdot)$. This approach was also implemented in Bevilacqua et al. (2019a). In particular, for any given y we defined the conditional PDF along x-direction:

$$\varphi(x \mid y) := f(x, y) / C \tag{7}$$

where $C$ is the appropriate normalizing constant.

Then we calculated the marginal PDF along y-direction:

$$f_X(y) = \int_{a(y)}^{b(y)} f(x, y) dx \tag{8}$$

where $a(y)$ and $b(y)$ are the limits of the mapping region along the x-direction for any given $y$.

For any value $(i, j)$ in $[0, 1]^2$ we define the transformation:

$$H(i, j) = (\, \Phi^{-1}(\, i \mid F_X^{-1}(j)\,),\, F_X^{-1}(j)\,) \tag{9}$$

where $\Phi(\cdot \mid y)$ and $F_X(\cdot)$ are the cumulative functions of $\varphi(\cdot \mid y)$ and $f_X(\cdot)$, respectively.

Thus, once sampled a uniformly distributed LHS, we apply the transformation $H(\cdot)$ to obtain a set of input variables that are distributed according to the vent opening map $f(\cdot)$, but preserves the good space-filling properties of the original LHS.

## Author contribution

All co-authors contributed in the compilation and critical revision of volcanological data of SSVC and NCVC. AB, RC and AN contributed in the choice of the different strategies and assumptions adopted to construct vent opening probability maps. AB constructed the presented vent opening probability maps. AA performed the branching energy cone simulations and then processed the numerical results. AB, AA and RC prepared the manuscript with contributions from all co-authors.

## Competing interests

The co-authors do not have any competing interests.

## Acknowledgements

The work is dedicated to the memory of Eduardo Gutierrez, who suddenly passed away before having the opportunity to receive the final version of the manuscript. We are all deeply indebted to him: without his enthusiasm and continuous stimulus to initiate and enforce our collaboration, we would never reach these results. We lost a precious teammate and a good friend. Rest in peace, Eduardo!

This research was supported by the RIESCA project, funded by the Italian Development Cooperation and aimed at promoting applied training in risk scenarios in Central America. The constant coordination and support to the project by Prof. Giuseppe Giunta is greatly acknowledged.

Alvaro Aravena was also financed by the French government IDEX-ISITE initiative 16-IDEX-0001 (CAP 20-25).

We thank an anonymous referee and Dr. Pablo Tierz for their useful comments to improve this work.

## Code/Data availability

The program ECMapProb used to apply the branching energy cone model is available in https://github.com/AlvaroAravena/ECMapProb. The vent opening maps and derived data are also available upon request.

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

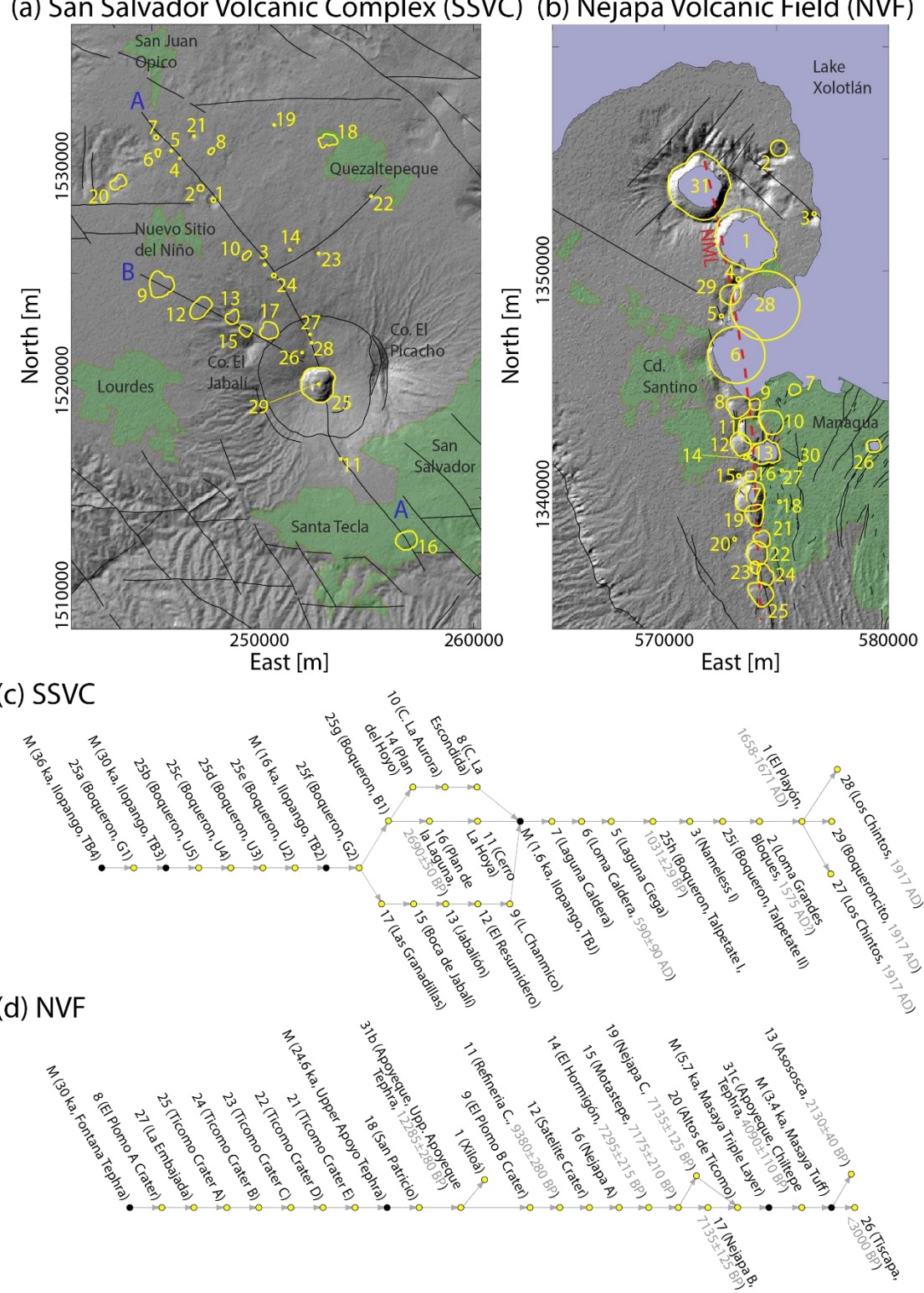

(a) San Salvador Volcanic Complex (SSVC)  (b) Nejapa Volcanic Field (NVF)

(c) SSVC

(d) NVF

**Figure 1:** (a-b) Topographic maps of San Salvador Volcanic Complex (a), mainly based on Ferrés et al. (2011), and Nejapa-Chiltepe Volcanic Complex (b), mainly based on Avellán et al. (2012). Data associated with the vents of past eruptions and major structures are also presented, which were employed for the construction of thematic vent opening probability maps. In panel (a), the letter A refers to a N40W-trending fault and the letter B refers to a N65W-trending fault. Eruptions 25a (~36 ka BP) and 25g (~3 ka BP) mark the limits between the three evolution stages defined for SSVC. In panel (b), NML refers to the Nejapa-Miraflores lineament. The green zones present indicative limits of the urbanized areas. (c-d) Schemes of the eruptive sequences associated with San Salvador Volcanic Complex (c) and Nejapa-Chiltepe Volcanic Complex (d). Yellow symbols represent their eruptions (see Tables 1 and 2) and black symbols represent regional stratigraphic markers. Eruption numbers refer to the vents (see panels a and b) and letters are used to identify different eruptions from a single vent. Bifurcations are used to separate sets of eruptions whose chronostratigraphic relationship is not known.

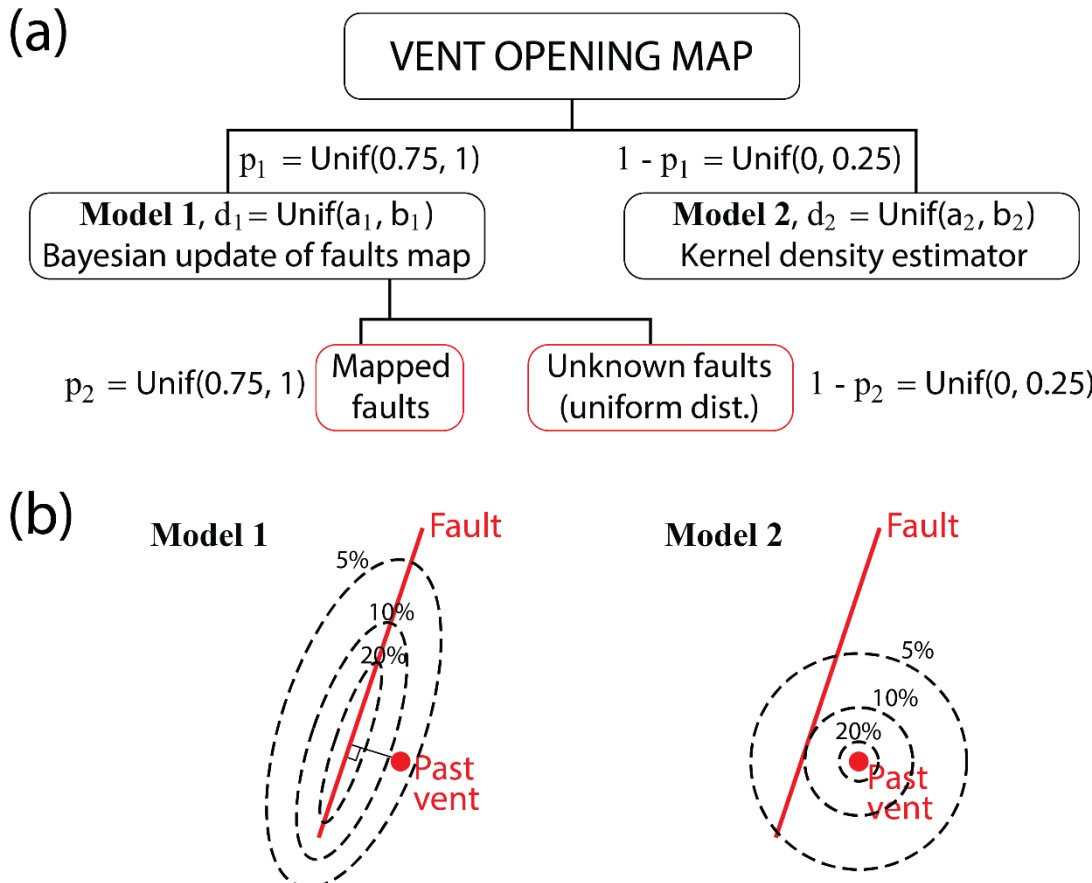

**Figure 2:** (a) Logic tree of the multi-model scheme, presenting epistemic uncertainty sources. Random variables modeling epistemic uncertainty sources are displayed. Red boxes show the two maps of prior probability of faults locations. (b) Schematic example of the application of the two models used in this work in the test case of a past vent near a fault.

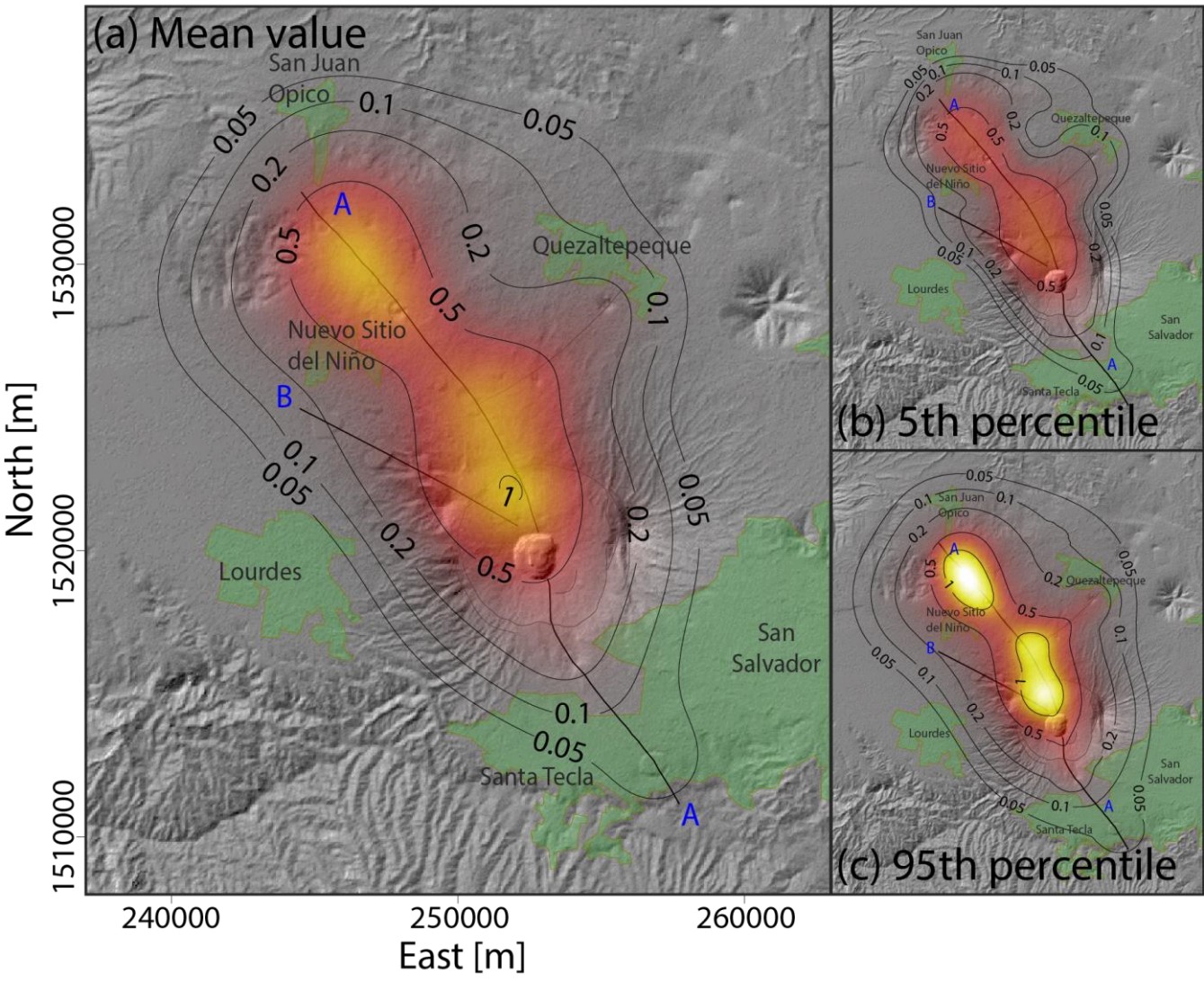

**Figure 3:** Density distribution of the probability of vent opening at San Salvador Volcanic Complex, associated with the occurrence of volcanic activity able to produce lava flows. (a) Mean value. (b) 5th percentile. (c) 95th percentile. Results are expressed in percentage per km².

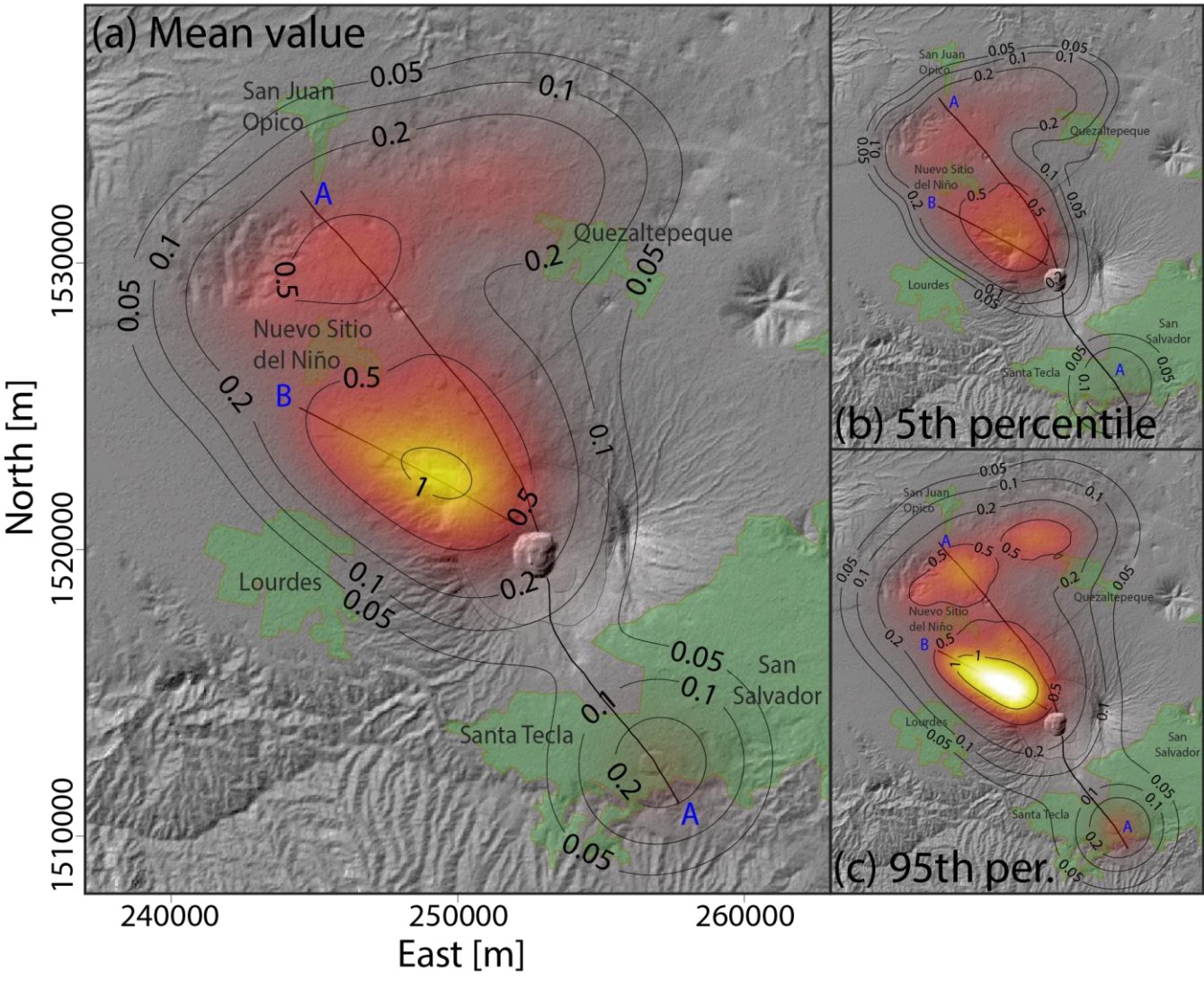

**Figure 4:** Density distribution of the probability of vent opening at San Salvador Volcanic Complex, associated with the occurrence of volcanic activity able to produce small-scale pyroclastic density currents. (a) Mean value. (b) 5th percentile. (c) 95th percentile. Results are expressed in percentage per km².


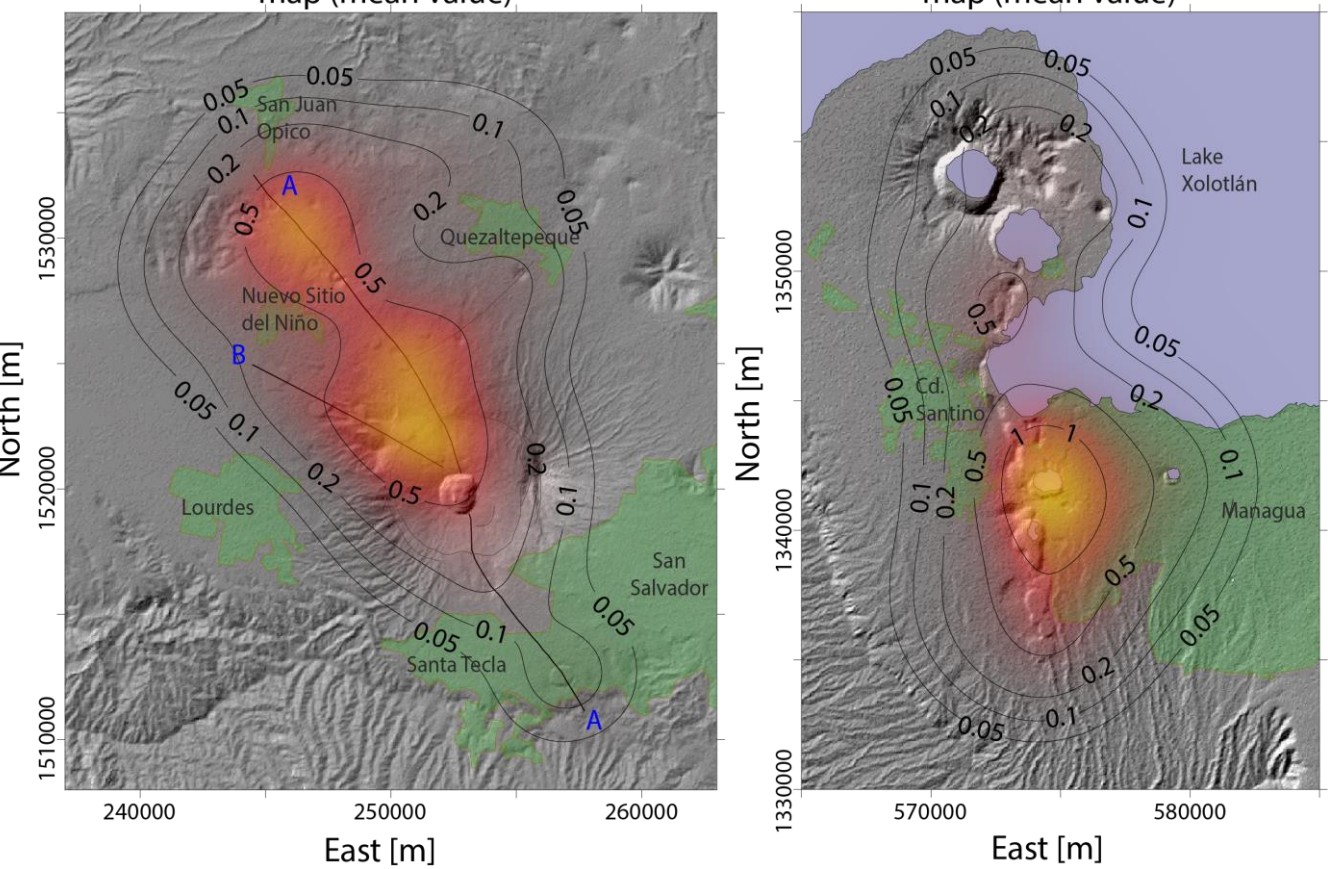

**Figure 5:** Density distribution of the probability of vent opening (mean value) at San Salvador Volcanic Complex (a) and Nejapa-Chiltepe Volcanic Complex (b). Results, which are not thematic, are expressed in percentage per km².


# Nejapa-Chiltepe Volcanic Complex - Lava emission

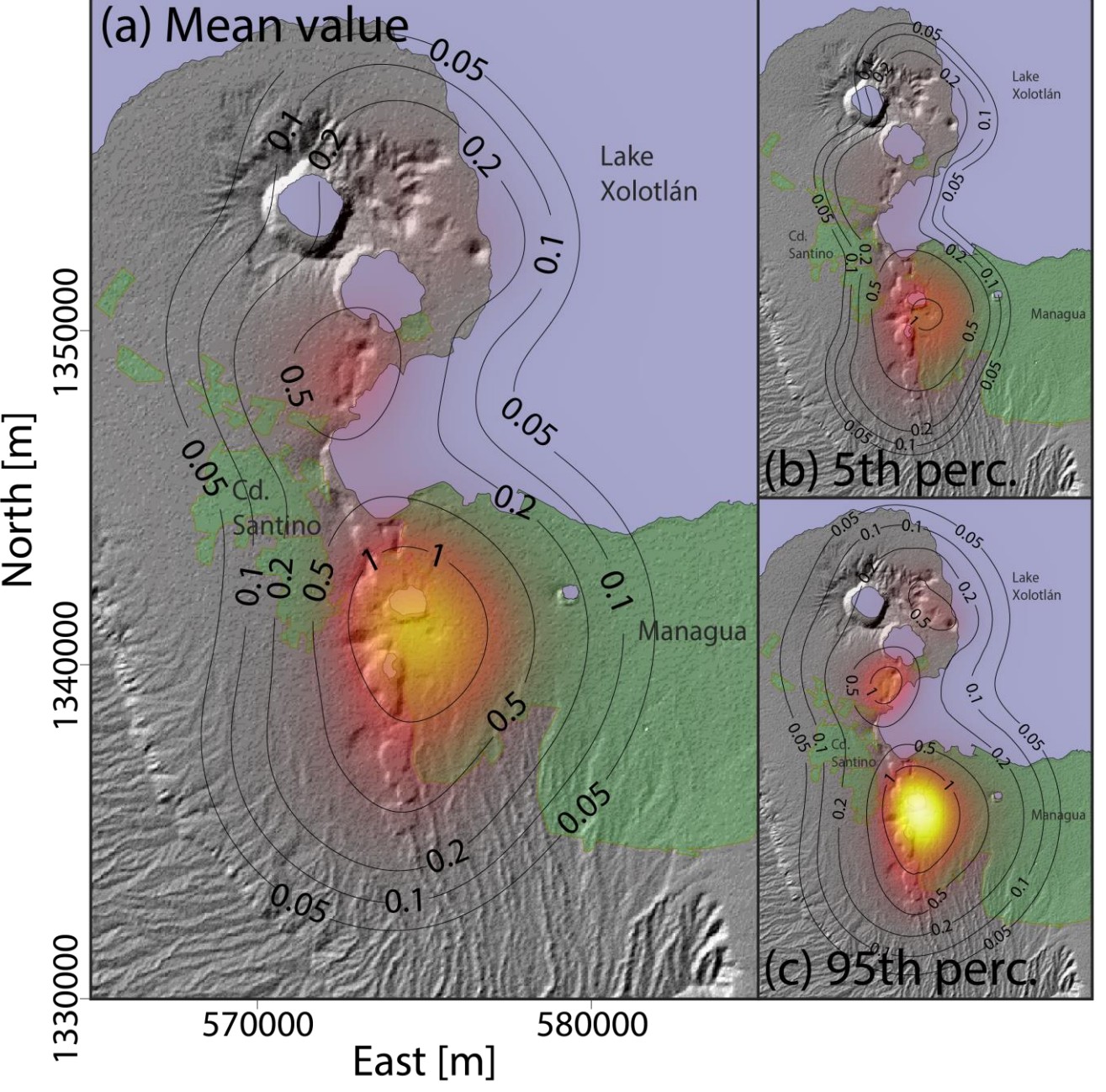

**Figure 6:** Density distribution of the probability of vent opening at Nejapa-Chiltepe Volcanic Complex, associated with the occurrence of volcanic activity able to produce lava flows. (a) Mean value. (b) 5th percentile. (c) 95th percentile. Results are expressed in percentage per km².


# Nejapa-Chiltepe Volcanic Complex - Small-scale PDCs

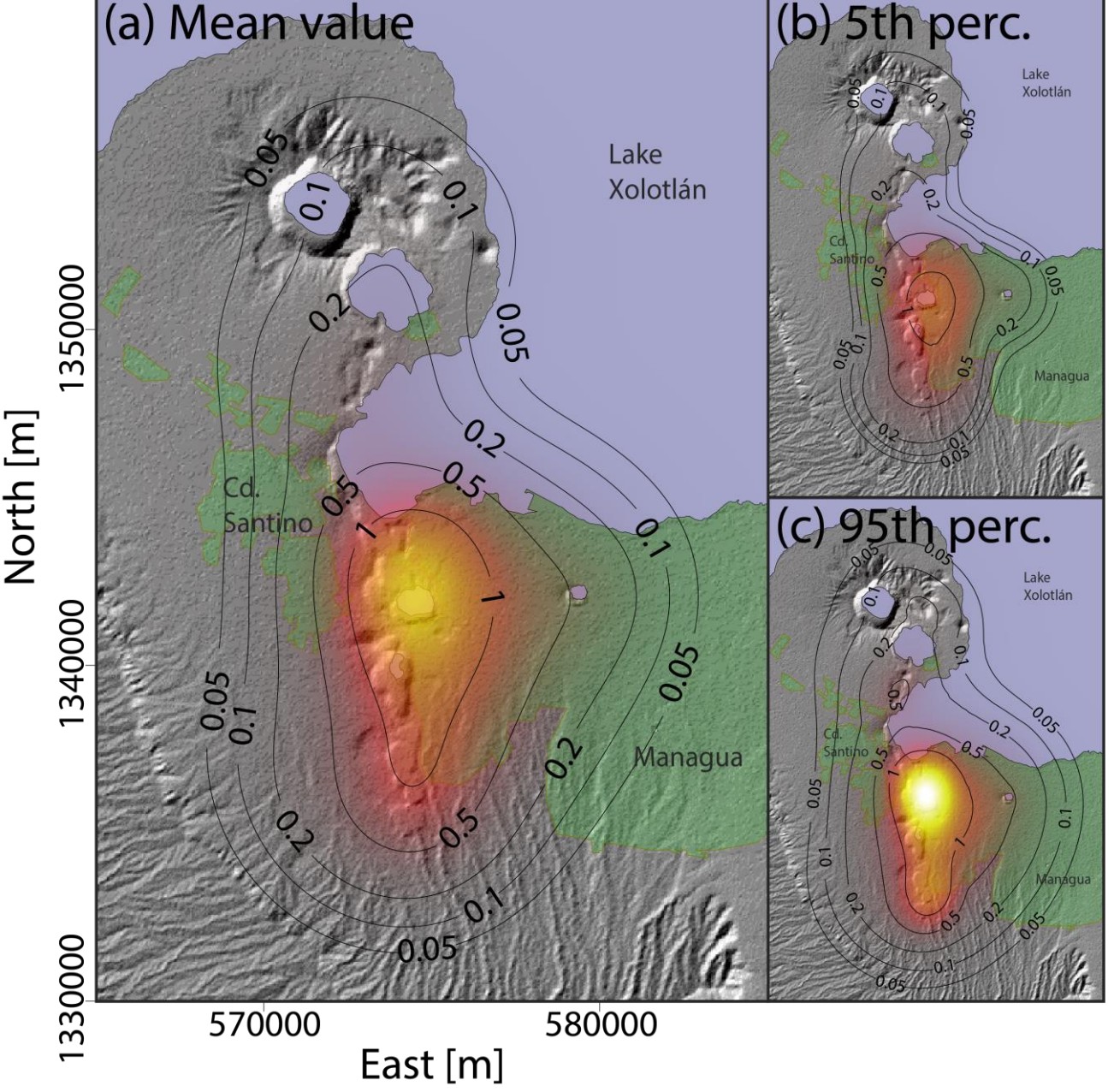

**Figure 7:** Density distribution of the probability of vent opening at Nejapa-Chiltepe Volcanic Complex, associated with the occurrence of volcanic activity able to produce small-scale pyroclastic density currents. (a) Mean value. (b) 5th percentile. (c) 95th percentile. Results are expressed in percentage per km$^2$.


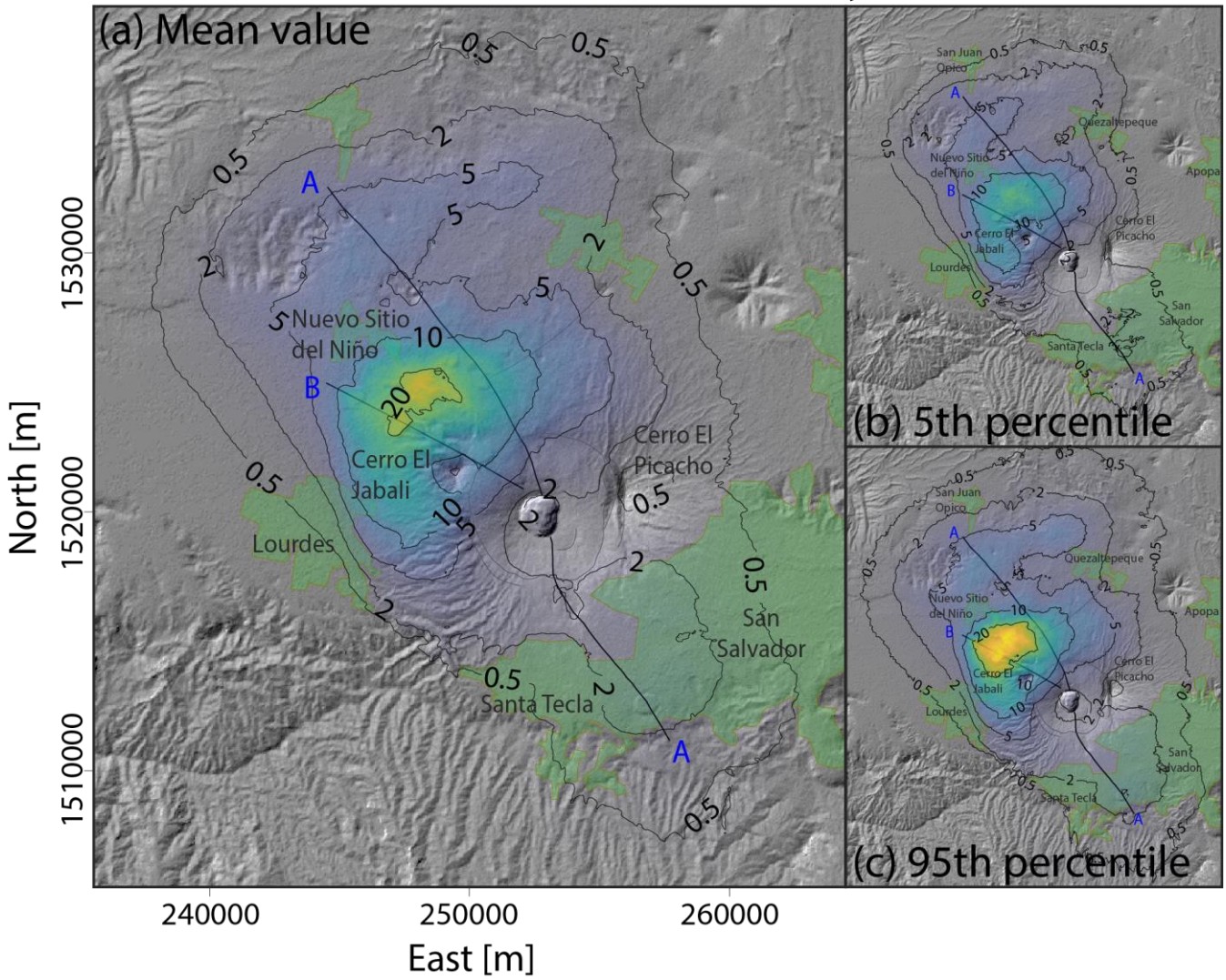

**Figure 8:** Inundation probability of small-scale PDC at San Salvador Volcanic Complex, derived from the systematic application of the branching energy cone model (Aravena et al., 2020a) and the vent opening probability map shown in Figure 4. (a) Mean value. (b) 5th percentile. (c) 95th percentile. Inundation probability is expressed in percentage. These probability maps are conditioned on the occurrence of an eruption able to produce small-scale PDCs.

# Nejapa-Chiltepe Volcanic Complex - Small-scale PDCs
## (Inundation probability)

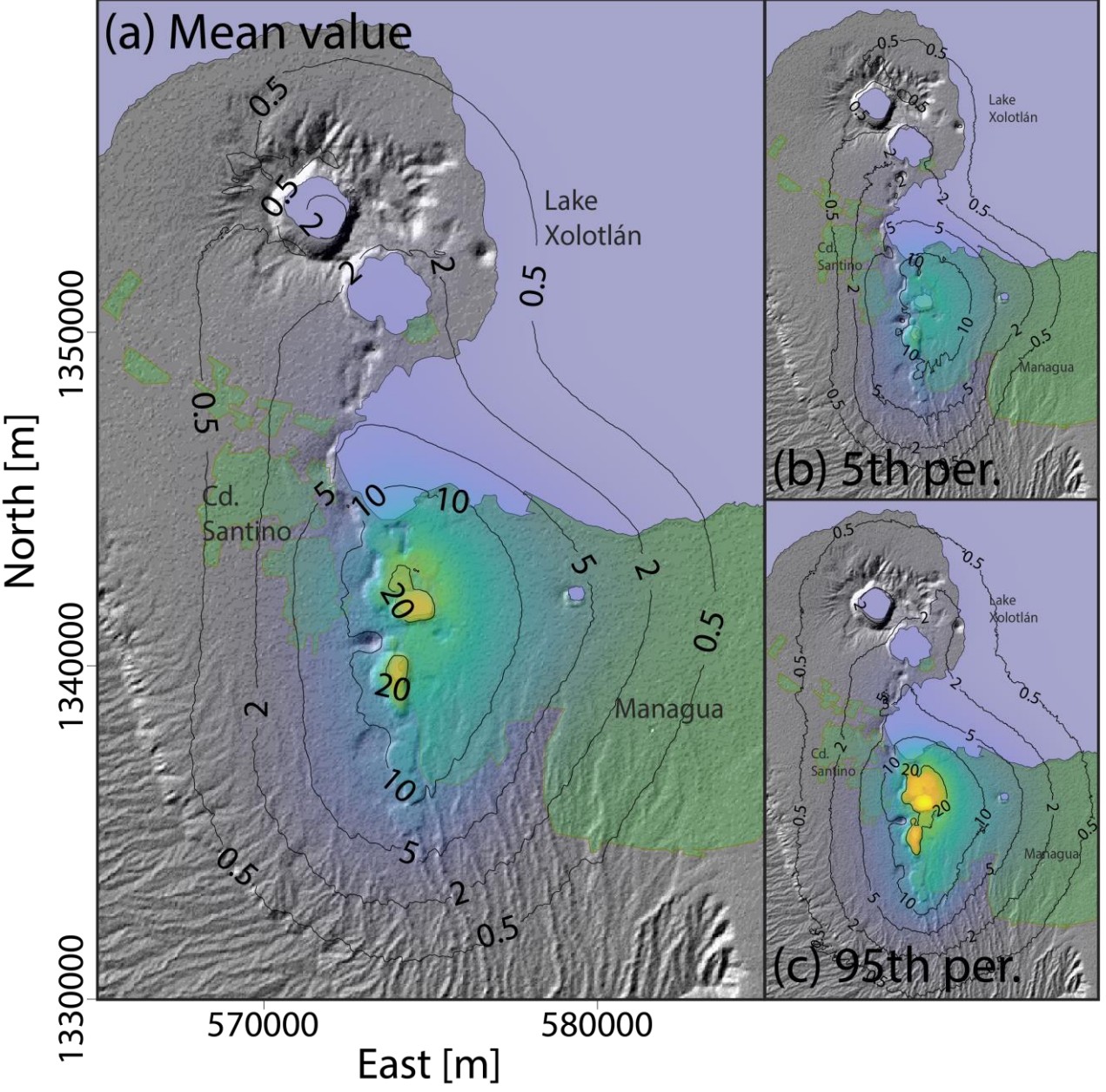

**Figure 9:** Inundation probability of small-scale PDC at Nejapa-Chiltepe Volcanic Complex, derived from the systematic application of the branching energy cone model (Aravena et al., 2020a) and the vent opening probability map shown in Figure 7. (a) Mean value. (b) 5th percentile. (c) 95th percentile. Inundation probability is expressed in percentage. These probability maps are conditioned on the occurrence of an eruption able to produce small-scale PDCs.

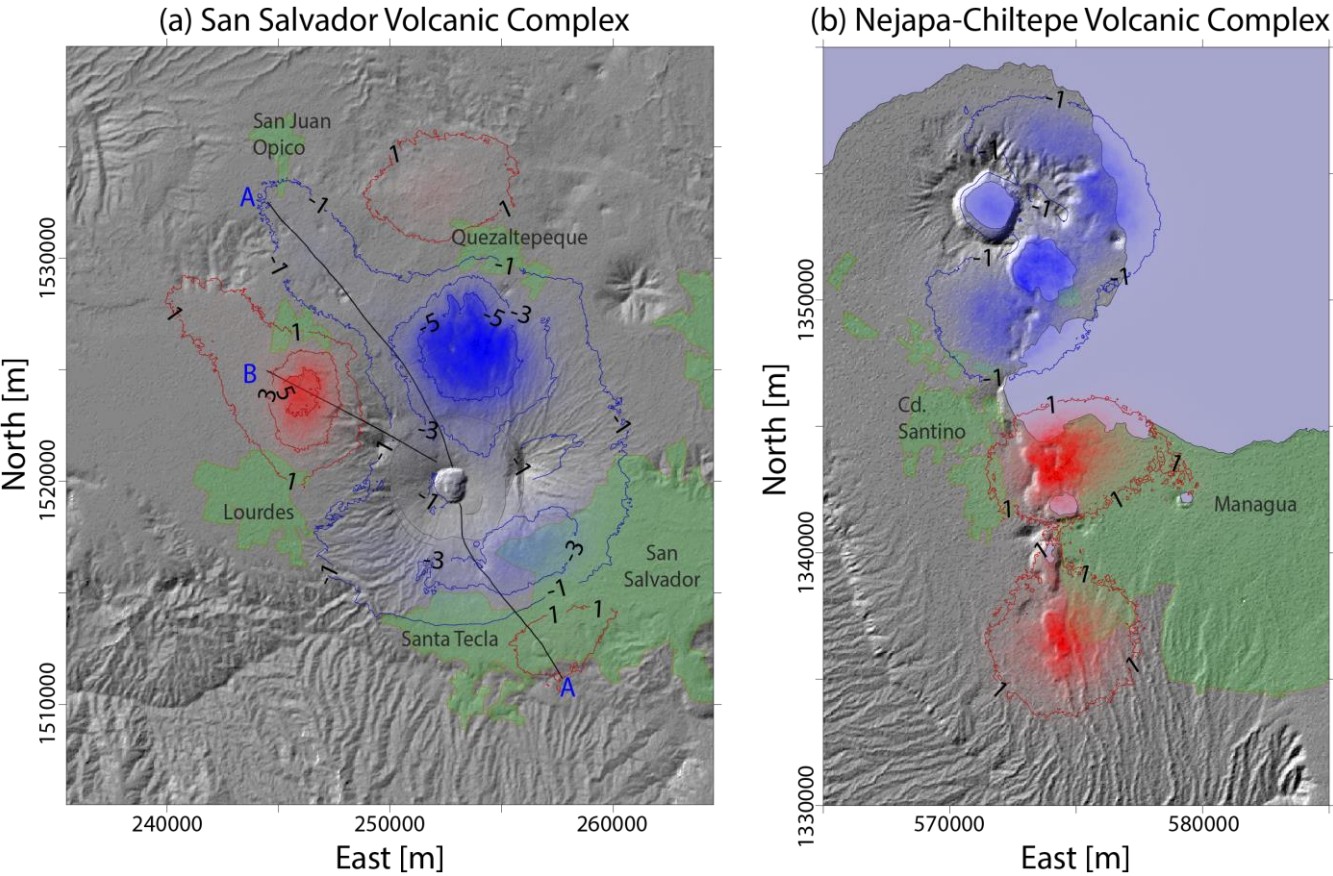

**Figure 10:** Difference between the probability maps of small-scale PDC inundation presented in Figs. 8 and 9 (mean values), which are derived from the use of thematic vent opening maps, and the equivalent results that would be derived from the application of non-thematic vent opening maps. The difference of inundation probability is expressed in percentage. These maps highlight the relevance of using thematic vent opening maps in the assessment of volcanic hazard at volcanoes where eruptive style may significantly change with vent location. (a) San Salvador Volcanic Complex. (b) Nejapa-Chiltepe Volcanic Complex.

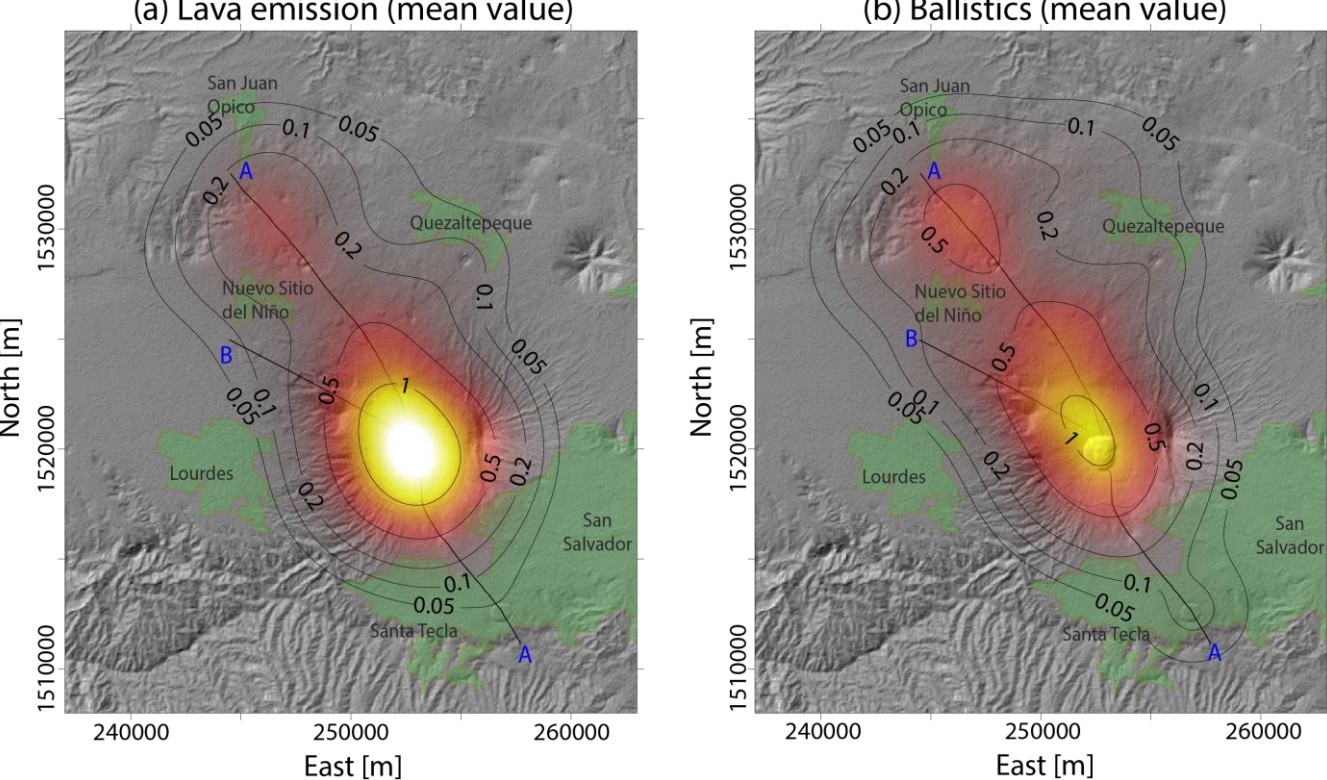


**Figure 11:** Density distributions of the probability of vent opening at San Salvador Volcanic Complex (mean value) computed using an event counting-based approach (i.e. the polygenetic central vent presents a weight higher than monogenetic vents). These maps, which are thematic, are associated with the occurrence of volcanic activity able to produce lavas flows (a) and ballistic fragments (b). Results are expressed in percentage per km$^2$.


**Table 1:** Vents considered in the SSVC (Sofield, 1998; Ferrés et al., 2011; Ferrés et al., 2013; Ferrés, 2014).

| N | Name | Edifice | Lava emission | Minor PDCs | Ballistic fragments | Minor fallout |
|---|------|---------|---------------|------------|---------------------|---------------|
| 1 | El Playón | SC | Y | N | Y | Y |
| 2 | Loma de Grandes Bloques | SC | Y | N | Y | Y |
| 3 | Nameless I | SC | PY | N | Y | Y |
| 4 | Boca Tronadora | LF | Y | N | Y | PY |
| 5 | Laguna Ciega | SC | PY | N | Y | Y |
| 6 | Loma Caldera | TC/M | PN | Y | Y | Y |
| 7 | Laguna Caldera | SC | PY | N | Y | Y |
| 8 | Crater La Escondida | EC | PN | Y | Y | Y |
| 9 | Laguna de Chanmico | M | PN | Y | Y | Y |
| 10 | Crater La Aurora | TC | PN | Y | Y | Y |
| 11 | Cerro La Hoya | SC | PY | N | Y | Y |
| 12 | El Resumidero | TR | PN | Y | Y | Y |
| 13 | Jabalión | EC | PN | Y | Y | Y |
| 14 | Plan del Hoyo | SC | PY | N | Y | Y |
| 15 | Boca de Jabalí | EC | PN | Y | Y | Y |
| 16 | Plan de La Laguna | M | PN | Y | Y | Y |
| 17 | Las Granadillas | M | PN | Y | Y | Y |
| 18 | Crater Quezaltepeque | TR | PN | Y | Y | Y |
| 19 | Crater Lavas El Playón | EC | PN | Y | Y | Y |
| 20 | Plan de La Hoya | TC | PN | Y | Y | Y |
| 21 | Montaña Las Viboras | SC | PY | N | Y | Y |
| 22 | El Cerrito | SC | PY | N | Y | Y |
| 23 | Cerro 14 de Marzo | SC | PY | N | Y | Y |
| 24 | Nameless II | SC | PY | N | Y | Y |
| 25 | Boquerón | CC | PN | PN | Y | PN |
| 26 | Nameless III | SC | PY | N | Y | Y |
| 27 | Los Chintos Vents I | LF | Y | N | Y | PN |
| 28 | Los Chintos Vents II | LF | Y | N | Y | PN |
| 29 | Boqueroncito | SC | Y | N | Y | Y |

SC: Scoria cone. LF: Lava flow (emitted from a fissure vent). TC: Tuff cone. M: Maar. EC: Explosion crater. TR: Tuff ring.
CC: Central crater.

Y: Yes. PY: Probably yes. PN: Probably no. N: No.

A scheme of the eruptive sequence is presented in Figure 1c.

**Table 2:** Vents considered in the NCVC (Pardo et al., 2008; Freundt et al., 2010; Avellán et al., 2012).

| N | Name | Edifice | Lava emission | Minor PDCs | Ballistic fragments | Minor fallout |
|---|------|---------|---------------|------------|---------------------|---------------|
| 1 | Xiloá | M | N | Y | Y | Y |
| 2 | El Tamagas | SC | PY | N | Y | Y |
| 3 | Chiltepe Scoria Cone | SC | PY | N | Y | Y |
| 4 | Xiloá Scoria Cone | SC | PY | N | Y | Y |
| 5 | Miraflores Scoria Cone | SC | PY | N | Y | Y |
| 6 | Miraflores Maar | M? | N | Y | Y | Y |
| 7 | Acahualinca Crater | M? | N | Y | Y | Y |
| 8 | El Plomo A Crater | TC | PN | Y | Y | Y |
| 9 | El Plomo B Crater | TC | PN | Y | Y | Y |
| 10 | Los Arcos Crater | M? | PN | Y | Y | Y |
| 11 | Refineria Crater | TR | PN | Y | Y | Y |
| 12 | Satelite Crater | TR | PN | Y | Y | Y |
| 13 | Asososca | M | N | Y | Y | Y |
| 14 | El Hormigon | SC | PY | N | Y | Y |
| 15 | Motastepe | SC | PY | N | Y | Y |
| 16 | Nejapa A | M | PN | Y | Y | Y |
| 17 | Nejapa B | M | N | Y | Y | Y |
| 18 | San Patricio | SC | PY | N | Y | Y |
| 19 | Nejapa C | M | PN | Y | Y | Y |
| 20 | Altos de Ticomo | SC | PY | N | Y | Y |
| 21 | Ticomo Crater E | M | PN | Y | Y | Y |
| 22 | Ticomo Crater D | M | PN | Y | Y | Y |
| 23 | Ticomo Crater C | M | PN | Y | Y | Y |
| 24 | Ticomo Crater B | M | PN | Y | Y | Y |
| 25 | Ticomo Crater A | M | PN | Y | Y | Y |
| 26 | Tiscapa | M | N | Y | Y | Y |
| 27 | La Embajada | SC | PY | N | Y | Y |
| 28 | Xolotlán Lake | M? | N | Y | Y | Y |
| 29 | Nameless I | SC? | PY | N | Y | Y |
| 30 | Nameless II | SC? | PY | N | Y | Y |
| 31 | Apoyeque | StC | N | PN | Y | PN |

SC: Scoria cone. TC: Tuff cone. M: Maar. TR: Tuff ring. StC: Stratocone.

Y: Yes. PY: Probably yes. PN: Probably no. N: No.

A scheme of the eruptive sequence is presented in Figure 1d.

**Table 3:** Summary of main features of the vent opening probability maps presented in this work.

| San Salvador Volcanic Complex (SSVC) | | | | |
|---|---|---|---|---|
| **Thematic map** | **Lava emission** | **Small-scale PDCs** | **Ballistics** | **Low-intensity fallout** |
| Figure | 3 | 4 | S1 | S2 |
| Zone of maximum probability density | N40W-trending fault A | N65W-trending fault B | N40W-trending fault A | N40W-trending fault A |
| Maximum probability density (mean value map) | 1.0% per km$^2$ | 1.1% per km$^2$ | 0.9% per km$^2$ | 0.9% per km$^2$ |
| 90% confidence of maximum probability density | [0.7%, 1.6%] per km$^2$ | [0.8%, 1.6%] per km$^2$ | [0.7%, 1.3%] per km$^2$ | [0.7%, 1.2%] per km$^2$ |
| Total vent opening probability near the northern portion of fault A | 39.3% | 26.8% | 35.0% | 35.1% |
| Total vent opening probability near the southern portion of fault A | 7.0% | 6.1% | 7.1% | 6.5% |
| Total vent opening probability near fault B | 13.9% | 23.5% | 16.7% | 16.3% |
| Nejapa-Chiltepe Volcanic Complex (NCVC) | | | | |
| **Thematic map** | **Lava emission** | **Small-scale PDCs** | **Ballistics** | **Low-intensity fallout** |
| Figure | 6 | 7 | S3 | S4 |
| Zone of maximum probability density | Near Asasosca maar | Near Asasosca maar | Near Asasosca maar | Near Asasosca maar |
| Maximum probability density (mean value map) | 1.6% per km$^2$ | 1.7% per km$^2$ | 1.6% per km$^2$ | 1.6% per km$^2$ |
| 90% confidence of maximum probability density | [1.0%, 2.3%] per km$^2$ | [1.2%, 2.5%] per km$^2$ | [1.1%, 2.2%] per km$^2$ | [1.1%, 2.3%] per km$^2$ |
| Total vent opening probability inside the limits of Managua city | 28.9% | 35.1% | 31.2% | 32.0% |