# Peer review of "Thematic vent opening probability maps and hazard assessment of small-scale pyroclastic density currents in the San Salvador Volcanic Complex (El Salvador) and Nejapa-Chiltepe Volcanic Complex (Nicaragua)"

_Natural Hazards and Earth System Sciences, 2020_

## Referee Comment (RC1) · Anonymous Referee #1 · 2 Feb 2021

This paper is well prepared, clear, and present a good structure. I think could be of interest to a broader audience interested in volcanic hazards and their impact in Central America. In this context there are some elements to considered, in order to make this study easier to fallow. In Results section there are a lot numbers indicating probability and interval confidence. Maybe if uses a table, the information will be more clear and easier to analyze and compare.

[Figure]

Technical corrections - Figure 3 and 4: consider to change some characteristic of trending faults lines, in order to contrast. Keep consistency in identification of A and B (Blue color letters) - Figure 5a: Keep consistency in identification of A and B - Table 1: line 15 – Jabalí, line 25 – Boquerón

———————————————

---

## Referee Comment (RC2) · Pablo Tierz (Referee) · 17 Feb 2021

**General comments**

The manuscript presents a collection of 'thematic' maps for the probability of vent open-ing, given eruption, at San Salvador (El Salvador) and Nejapa-Chiltepe (Nicaragua) volcanic complexes. By 'thematic', the authors interpret that vent-opening-probability maps are built using data points (or areas, including epistemic uncertainty) of past

vent locations, which are partitioned according to the occurrence of different hazardous phenomena during the associated eruptions. Additionally, the authors present maps of probability of invasion from pyroclastic density currents (PDCs), computed adopting the thematic description of the vent-opening probabilities, and compare them with those computed using 'non-thematic' (i.e. independent of hazardous phenomena) vent-opening models. They use the novel tree-branching energy cone model, recently developed and presented by some of the authors, to simulate PDC invasion.

I honestly think that the manuscript is a valuable contribution, as it proposes an interesting approach to explore spatial patterns in eruptive style, and could be complementary to previous and future studies tackling this complex problem, which is highly relevant to volcanic hazard assessment. Moreover, the initial probabilistic volcanic hazard assessment (PVHA), carried out at two volcanic systems with such high density of population on and around them, should represent vital information to manage volcanic risk in the area.

My main reservations concerning the content and presentation of the manuscript are the following:

1. The main methodology of the manuscript, currently on Appendices A.1 and A.2 should be moved to the main text of the manuscript, and be better explained (perhaps, step by step), including an improved visual description of the methods in Figure 2 (or another figure). In relation to the methodology, further clarification should also be provided as regards: (1) whether the ballistics thematic maps are mathematically equivalent to the non-thematic maps; (2) why the thematic maps for lava emission appear to be quite similar to the ballistics/non-thematic maps, considering that the datasets for the two phenomena seem to be relatively different (Tables 1 and 2).

2. The justification of joining the Nejapa-Miraflores fissure-vents complex with the Apoyeque caldera, under a common, underlying data-generation process in terms of vent locations (as well as eruptive phenomena) should be, at least, expanded and

made more clear, considering the significant differences in terms of geochemistry, eruption size and style at the two volcanic systems.

3. Discussion could be enriched around the differences between the approach of calculating P(vent location | hazardous phenomena), presented here; and that of calculating P(hazardous phenomena | vent location), which is more common in volcanic hazard assessments using event tree models (e.g. Tonini et al., 2014; Thompson et al., 2015; Sandri et al., 2018; Tierz et al., 2020).

4. The justification of using the tree-branching energy cone model should be also expanded, given the expected radial dispersal of PDCs related to phreatomagmatic activity, and the apparent limited channelization that the flows may be subject to (judging, very preliminarily, by the topography of the two volcanic systems). If possible, comparison of results obtained with the 'classical' (non-branching) energy cone could be an interesting addition to the manuscript.

The rest of my comments mostly relate to more specific points and suggestions, which I believe could help improve the manuscript. In summary, I would support the acceptance of this contribution to Natural Hazards and Earth System Sciences, after major revisions have been made.

Receive my best regards,

Pablo Tierz

**Specific comments**

L16 – Change to "Densely inhabited cities are built on them and their surroundings"?

L18-19 – Perhaps explain here that the novelty is on the 'thematic' representation of the vent-opening maps.

L26-27 – Considering that PDC channelization does not appear to be crucial at the two volcanic systems presented (judging, preliminarily, by the topography in the DEMs

shown in Figure 1), the manuscript should better justify the use of the branching-version of the energy cone, instead of the 'classical' one. If possible, performing an analysis with the classical energy cone at San Salvador Volcanic Complex, to compare the results from the two versions, would be an interesting addition to the manuscript, even if this analysis had to be included in the Supplementary Information (SI). Many thanks.

L26-28 – OK, but one could also argue that the aleatory uncertainty in PDC generation (i.e. source conditions, including vent location) could be explored using a non-thematic vent-opening map coupled with spatially-varying probability density functions (PDFs) for the energy cone model parameters. This could describe how 'external factors' (e.g. groundwater) might influence the eruptive style (e.g. phreatomagmatism) and, therefore, the types of PDCs generated (modeled via different model parameter values, cf. Tierz et al., 2016a). While I understand that this is beyond the scope of this manuscript, I think a richer discussion on these aspects should be incorporated. In the end, the authors are proposing to calculate the probability of vent opening conditional on eruptive style (i.e. P(vent | style) while most of the previous approaches in the literature (e.g. event-tree models) have devised the opposite conditional probabilities: that is, probability of eruptive style given vent location: P(style | vent). In my opinion, this aspect merits further discussion in the manuscript. Many thanks.

L30-31 – Please rephrase. Volcanic hazard assessment also includes the assessment of the probability of eruption or the probability of vent location. Many thanks.

L32-33 – Please add Newhall and Hoblitt (2002); Newhall and Pallister (2015); Sandri et al. (2014). Many thanks.

L35-36 – I would diversify the references here to include other research groups and/or volcanic hazardous phenomena. You could include, for example: Bonadonna et al. (2005); Costa et al. (2009); Del Negro et al. (2013); Mead and Magill (2017); Strehlow et al. (2017); Gallant et al. (2018); Tierz et al. (2018). Many thanks.

L40 – Similarly to the previous comment, please expand/diversify the list of references.

You could consider removing Alberico et al. (2002), as Bevilacqua et al. (2015) also analyze vent opening at Campi Flegrei. Additionally, you could include the following references: Connor et al. (2012); Thompson et al. (2015); Clarke et al. (2020). Many thanks.

L47 – Please add Clarke et al. (2020) for Aluto volcano, in the Main Ethiopian Rift. Many thanks.

L50-51 – I agree with this statement, but I think it would be important to briefly comment here, and add a proper argumentation in the Discussion section, about the significant difficulties associated with interpreting and modeling the dependence of eruptive style on vent location (see e.g. Thompson et al., 2015; Tonini et al., 2015; Sandri et al., 2018; Tierz et al., 2020). Many thanks.

L52 – Please use "hazard" instead of "risk" here. Many thanks.

L57 – "is clearly influenced by their position **(e.g. Tonini et al., 2015; Sandri et al., 2018)**"

L66 – Please see my previous comments about the 'dichotomy' between quantifying P(vent | style) vs P(style | vent) (e.g. in event trees), and add some more details in the Discussion section. Many thanks.

L70-71 – Is the sentence about the funding needed here within the body text of the manuscript? The statement is already included in the Acknowledgements section.

L88 – Maybe a brief description of how exactly the polygons were defined or calculated would be a good addition to the manuscript, even if it appears in the SI. Many thanks.

L90-91 – What is the source of this uncertainty? Poorly-preserved deposits? Buried crater structures? Other? Please briefly complement. Many thanks.

L91-94 – Whether here and/or in the caption of Figure 1c-d, I think it would be important to add some explanation about the 'bifurcations' and the arrows in the 'stratigraphical/temporal trees' presented. Do the bifurcations represent events (units) identified on different stratigraphical sections? If they do: can (some of) the units be correlated across sections, beyond their stratigraphical position with respect to the regional markers? In the context of Stage III of volcanic activity at San Salvador, I think it could be beneficial to indicate which unit(s) is/are interpreted to represent the beginning of this most recent stage of activity. Many thanks.

L100-101 – Are these 'probability levels' used as the 'weighting' to count a particular vent location as belonging to the datasets used to compute the different thematic maps? Many thanks for the clarification.

L111 – "E-W"

L112 – Change to something like: "is formed by **a summit cone** called Boqueron Volcano (BV), **which is enclosed by the remnants of the** volcanic edifice of San Salvador Volcano (SSV), **which experienced a vertical collapse around 36 ka (Sofield, 1998)**. At least 25 monogenetic vents [...]"? Many thanks.

L135-137 – Are you implying that this type of activity is unlikely to occur in the future?

L150-151 – What is the recurrence period estimated for summit eruptions? Please report it, if it is available. Many thanks.

L154-155 – I think the merging of eruptive vent locations from the three stages of activity, considering the possibility of significant changes in the magmatic system after the end of each stage (as observed in changes in the eruptive dynamics and spatial location of vents), deserves an expanded justification. This choice implies the assumption of a common, underlying data-generation process for vent locations at SSVC over the last 70 ka, and it represents the basis for the statistical modeling adopted henceforth. In my opinion, it would be beneficial to present two different hypotheses/assumptions: one that considers all vent locations; another that considers only vent locations during the last stage of volcanic activity. You should clearly describe, for any future reader

of the manuscript, which implications each of the hypotheses/assumptions has (e.g. stationarity of vent locations in time). Many thanks.

L156-158 – Please clarify how these probabilities of occurrence are exactly used in the thematic vent-location datasets, to run Model 2 shown in Figure 2. As specified in my general comments, moving Appendices A.1 and A.2 to the main text of the manuscript may be necessary. Many thanks.

L173-174 – Given the substantial differences in magma geochemistry, eruption sizes and styles, and even spatial distribution of eruptive vents between the Nejapa-Miraflores fissure-vents system and the Apoyeque caldera, some extra explanation would be needed about why they are treated as having a common, underlying data-generation process in terms of vent locations. This should be carefully explained and justified in the manuscript. Many thanks.

L186-187 – Please rephrase slightly: the difference between 28 past vent locations and 31 does not seem critical. Could you also expand on which vents were not included by Connor et al. (2019) and why you included them? Many thanks.

L187 – Uncertainty in vent locations: a few extra details on how you defined the polygons that define this source of uncertainty would be appreciated. Many thanks.

L192 – It is not clear how the current presence of water relates to the eruptive phenomenology at the time of each eruption. Please expand. Many thanks.

L195-199 – Please see my previous comment on how the justification of mixing the Nejapa-Miraflores and Apoyeque volcanic systems under a common underlying data-generation process deserves substantial further explanation and justification. Many thanks.

L202-205 – Please add a few references for previous similar approaches, from which the ones that are cited built upon: e.g. Connor and Hill (1995); Martin et al. (2004); Connor and Connor (2009); Bebbington and Cronin (2011); Connor et al. (2012);

[Figure]

Cappello et al. (2012). I saw that many of this references are included in Appendices A.1 and A.2. This could be another reason to move these Appendices to the main text of the manuscript. Many thanks.

L206-207 – If Model 1 considers the past vent locations: are these vent locations used twice (Model 1 and 2) in generating the final vent-opening-probability maps? Please clarify. Many thanks.

L210 – Please report the spatial resolution of the vent-opening computational grid. Many thanks.

L212-214 – As a key novelty, it would be very beneficial if Figure 2 could incorporate some explanation about the new methodology you present here for the thematic maps. Many thanks.

L214 – Change "leads" to "lead"

L215-216 – Specify that the continuous PDF is bivariate (Easting-Northing)?

L216-218 – What is the area of each grid point in the computational grid? I wonder if providing the values of vent-opening probability by grid point (instead of per $km^2$) could be easier to interpret by the readers. Also, could you specify whether the thematic vent-opening probabilities integrate to 1, by hazardous phenomenon? That is: given, say, a lava flow (emission) to occur in one of the volcanic systems: is the (mean) probability of a vent opening, summed across the computational grid, considered equal to 1? I interpret from the Appendices that this is the case, but some clarification could be welcome around these lines. Many thanks.

L218-219 – I think this is a very important point and perhaps it deserves a little extra explanation or justification here. Many thanks.

L221 – Redundant text: "Jaquet et al."?

L233 – How much the results could be expected to change if, say, p1 = Unif(0.6, 1)?

Many thanks.

L236 – Would it be possible that the "unknown" fault structures in the region were not distributed uniformly, for instance, due to regional tectonics? Perhaps adding a brief comment about this aspect could be beneficial. Many thanks.

L238-240 – "[. . .] or likelihood based techniques (Bevilacqua et al., 2017a; Bevilacqua et al., 2018), **and/or by carrying out further geophysical surveys in the area/region (e.g. Martin et al., 2004; Jaquet et al., 2012; Runge et al., 2016; Deng et al., 2017)**".

L244 – Could you provide some more details on how many sub-regions and/or which ones? It is also interesting to see that the two volcanic systems, despite expected differences in terms of their local structural and tectonic conditions, have the same distribution for $d_1$. Could you expand on this a bit? Many thanks.

L253 – But $f(x)$ is independent on fault structures, right? Please clarify. In addition, the fact that the past vent locations seem to be used in Model 1 should be made more evident. Many thanks.

L258-264 – Maybe it is better to present the non-thematic vent-opening maps first (as Figure 3) and then all the thematic maps for the two volcanic systems under study. If the ballistics thematic vent-opening map is equivalent to the non-thematic map, I would suggest that the 5*th*- and 95*th*-percentile maps are included in the current Figure 5 and Figure S1 was not included in the manuscript. Many thanks.

L266-267 – "N65W" and "N40W", right?

L270-273 – It would be convenient to explain these calculations better: why 4 km? Where exactly is the "northern portion of fault A"? Which pixels are excluded (i.e. what "closer" means in this context)? Etc. Many thanks.

L273 – "Sequel"?

L275 – "N65W"?

L279-280 – Why these two maps are quite similar if the datasets for ballistics and lava flows appear to be relatively different in the two volcanic systems (Tables 1, 2)? Is it because the fault structures (known or unknown) have a very strong influence on the final vent-opening probabilities? If that is the case, the discourse about the thematic vent-opening maps, based on past vent locations, loses a bit of strength, in my opinion.

L283-284 – At some point, one may start to wonder if the stress on the thematic maps in the manuscript should be put almost exclusively on the 'small-PDCs-conditioned' vent-opening maps, given that the ballistics thematic map is mathematically equal to the non-thematic map, and both the lava-emission and tephra-fallout thematic maps seem similar to the former. In other words, if the data-generation process for the vent locations of (small-)PDC-forming eruptions is the only one that seems to be significantly different from the non-thematic vent-location variability (the former perhaps linked to local conditions, such as groundwater availability or fluid pathways, influencing the eruptive style?): why not to focus on these aspects more clearly in the manuscript? Many thanks.

L288-289 – Figure S3 may not be needed, if it is equivalent to Figure 5b. Many thanks.

L292 – Please see my previous comment.

L295 – Is the Miraflores scoria cone identified in Figure 1b?

L303-310 – This part could be moved to the Discussion section. If the visual comparison of the results from Connor et al. (2019) and those presented in this manuscript is important, perhaps a figure could be added to the SI, where the two vent-opening probability maps are displayed. Many thanks.

L306-307 – Please rephrase. A vent-opening probability map is a PDF, even if it only captures the aleatory uncertainty (i.e. single value of probability at each grid point) in the eruptive vent location. Many thanks.

L312-320 – If I understood correctly, you extracted 100 realizations from the array of

vent-opening model parameters ($p_1$, $p_2$, $d_1$, $d_2$), which generated 100 different values of vent-opening probability at each of the vent locations (1024). Is this correct? Then, you used these sets of vent-opening probabilities to weight the hazard footprints generated from running the branching-energy-cone model (cf. Sandri et al., 2018; Clarke et al., 2020, for the 'classical' energy cone), with fixed parameters independently of vent location. Is this correct? If it is, I do not understand why the total number of simulations is: 100 x no. of vents (1024), and not just 1 x 1024, as I picture vent-opening probabilities to be independent from PDC propagation. If any of the above is not correct, please clarify here and in the main text of the manuscript as well. Many thanks.

L319 – Please state the Digital Elevation Model (DEM) resolution over which the branching energy cone was run. Many thanks.

L320 – Please express $\phi$ in degrees, additionally. Many thanks.

L321 – Please expand on the links between the model parameters and the volcanological observations. What are the runout distances mentioned? From how many eruptions were they calculated? Were the associated eruptions large or small? Many thanks.

L322-323 – "We decided not to use variable input conditions for initial PDC characteristics **(e.g. Tierz et al., 2016a, b; Sandri et al., 2018; Aravena et al., 2020a; Clarke et al., 2020)**"

L325 - "would require additional information to properly calibrate these input parameters (e.g. Cioni et al., 2020) **and/or complementary data coming from analogue volcanoes (e.g. Sandri et al., 2012; Tierz et al., 2016a; Clarke et al., 2020)**"

L327-328 - "(e.g., Spiller et al., 2014; Ogburn et al., 2016; Ogburn and Calder, 2017; **Tierz et al., 2016b, 2018**; Rutarindwa et al., 2019; Patra et al., 2020)."

L331-332 – What is the spatial resolution of the hazard domain? Many thanks.

L335 – Consider removing "small-scale". Many thanks.

L337 – "Ciudad" or "Sitio"? Many thanks.

L336-338 – In my view, it is not only the topographic barrier of Cerro El Picacho which is hindering PDC inundation over (the western areas of) San Salvador city. Very importantly, given that the highest thematic vent-opening probabilities (for small-scale PDCs) are located on the NW flank of the stratocone (outside the caldera rim of the SSV edifice) and beyond, onto the surrounding plain (Figure 4); PDCs generated from some of these vents would need to overcome the whole 14km-basal-width, 1.2-1.4km-height volcanic edifice! (NB. Morphology data from Grosse et al., 2014).

L339-341 – I beg to disagree with this interpretation. In my opinion, the peak in thematic vent-opening probability for small-scale PDCs located 'between' Santa Tecla and San Salvador (and associated with vent 16, right? Figures 1 and 4) should be a more important factor conditioning the probabilities of PDC invasion in this area. I wonder how the map displayed in Figure 8 would change if you conditioned the vent opening, spatially, to vent locations with a (mean) vent-opening probability of (say) 0.5%/km$^2$ or greater. How much the probability of PDC invasion would decrease around the area of Santa Tecla, in this scenario? Perhaps, including this figure in the SI could be a very valuable addition to the manuscript. Many thanks.

L339 – Consider removing "small-scale". Many thanks.

L349 – Please change to "would be derived". Many thanks.

L349-351 – I would rephrase the sentence as follows: "These results show clearly the relevance of using thematic vent opening maps in the assessment of hazard **at** volcanoes **where eruptive style may significantly change with vent location**". Many thanks.

L356 – "at the **NNE flank**"?

L357 – Change "parameter" to "probability"?

L361 – Change "small" to "smaller"? Many thanks.

L364 – Please remove "Poland and Anderson (2020)" from this list and use examples previously cited instead (e.g. L44-50), and/or some of the additional references suggested below. Many thanks.

L365 – Please change "fields" by "systems". Many thanks.

L369 – This list of references should be more generic in terms of volcanic hazardous phenomena included, and wider in terms of research groups represented. Please add references like: Del Negro et al. (2013), Thompson et al. (2015), Sandri et al. (2014, 2018). Many thanks.

L370 - "plugging-in"?

L371-372 – Please add some references to previous relevant research in the topic: e.g. Marzocchi et al. (2008); Sandri et al. (2012); Selva et al. (2014); Tonini et al. (2016). Many thanks.

L376-378 - "We remark that previous studies already produced vent opening maps devoted to specific types of eruptions (e.g. Plinian and sub-Plinian eruptions in Tadini et al., 2017b; **or pumice-cone-forming eruptions in Clarke et al., 2020)**, to eruptions inside selected sub-regions (Bevilacqua et al., 2017b) or to a suite of pre-imposed eruptive scenarios (Ang et al., 2020)."

L382 – "which has been **reported for** different volcanic systems (e.g. Andronico et al., 2005; Coppola et al., 2009; **Sigmundsson et al., 2010; Sandri et al., 2012; Tierz et al., 2020)**"

L383 – "**N65W**-trending fault B" instead?

L384 – "**N40W**-trending fault A" instead?

L385-386 – This is interesting. Could you briefly expand on the possible reasons behind this lack of small-scale PDC activity from the central crater of Boqueron? Many thanks.

L388 – I think the exact location of Managua Lake may not be indicated in Figure 1?

L389-391 – This is mostly true for small-scale PDCs but it is more difficult to justify for any of the other three phenomena presented. Please try to expand your argumentation around this point here, and maybe in other parts of the manuscript. Many thanks.

L394 – Consider changing as follows: "The design and implementation of mitigation strategies **for** volcanic risk **are profoundly conditioned by** the characteristics of the volcanic **processes** under consideration and **the** vent position."

L409 - "sequel"?

L410-414 – If I am not wrong, you are adopting this very same assumption by using the past vent locations across different eruptive stages, for instance, at SSVC. Is this correct?

L414-416 – Please explain why did you not implemented a similar approach in the presented study. And/or excluded pre-3ka vents from your analysis. Many thanks.

L419-420 – One could also argue, then, that the spatial dataset used in the presented analysis may suffer from the same or very similar issues. How is the analysis influenced by 'buried vents', for instance? Many thanks.

L420-423 – I am not sure if I follow the reasoning presented here. If the stratocone has been the source of **many** eruptions, those may have produced **many** lava flows but also **many** PDC events, potentially. Moreover, monogenetic vents are interpreted to be the product of a single eruption, but that does not necessarily mean that they generated 1 lava flow or 1 PDC. In the end, all these considerations also depend on the degree of (temporal) detail on which the eruptions, and their associated hazardous phenomena, are analyzed (e.g. Jenkins et al., 2007; Wolpert et al., 2018; Bebbington Jenkins, 2019).

L426-434 – I wonder why not including the thematic vent-opening maps for small-scale PDCs in Figure 11, as the former are one of the main focuses of the manuscript. Many

thanks.

L431-434 – These sentences feel slightly disconnected and/or vague as they are now. Please improve the argumentation here. Many thanks.

L435-436 – Change "fields" to "systems"?

L436-440 – Please note that what are presented as unique features of the presented volcanic systems could be said of many other volcanoes in the world. Consider re-wording the sentence slightly. Many thanks.

L443 – Change "lavas" to "lava"

L448 – "**N65W**-trending fault B" instead?

L449 – "**N40W**-trending fault A" instead?

L455-456 - "which implies major challenges **in managing the volcanic risk** associated with NCVC" may be better. Many thanks.

L461-462 – You should introduce, and justify, more clearly early in the manuscript, why the use of the branching-energy-cone model is key in the context of the two volcanic systems analyzed, where PDC channelization does not appear to be a critical conditioning factor to PDC propagation (preliminarily judging by the DEMs of the areas). Many thanks.

L461 – Please see my previous comment on clarifying where these $10^5$ simulations exactly come from. Many thanks.

L470 – "Ciudad" or "Sitio"?

L477 – "decreased"

L482 – Change "hazards" to "risk"?

L491 – Therefore, the past vent-location data are used twice in the final vent opening map explained in Figure 2: once in Model 1 and another in Model 2. Is this correct?

Interactive
comment

Could you briefly expand on the reasoning behind this choice? Many thanks.

L496 – Please remove: "the outcrop of", and/or change it by "the surface projection of". Many thanks.

L496-497 – How are these prior PDFs calculated? $z(x)$ in formula (3) is the prior probability of $\zeta$ (or $\xi$? Symbols in the text and in formula (3) seem different. Are they referring to two different variables? Many thanks) at the vent location $x = (x_i, y_i)$. Is this correct? If it is, please explain better how this is calculated. Also: is $z_2$ a Uniform PDF over the whole vent-opening spatial domain, right? Many thanks.

L500, 520 – Perhaps it would be useful to report the formula for the posterior PDFs: $z_j(\zeta|D_i)$. It is not clear to me that it is shown. Many thanks.

L505 – Given the key importance of the weights $w_i$ in the paper, I suggest that (parts of) Appendix A.1 are actually moved into the main text of the manuscript. Many thanks.

L514 - "assume that, on average, about 95% of the vents [...]"?

L525-526 – Please rephrase slightly: e.g. "the expected distance **between past and future eruptive vents**". Many thanks.

L530 – Similarly to my previous comment, the critical relevance of the weights $w_i$ in the results shown in the manuscript could justify moving (parts of) Appendix A.2 to the main text of the manuscript. Many thanks.

L531-533 – Could you please expand a bit on the meaning, reasoning or implications of your choice in the presented manuscript? Many thanks.

**Figures and Tables**

Figure 1:

- It may be good to show where approximately within the 'timeline' of SSVC does the time 3ka sit. Many thanks.

- I would consider changing, at least one of, the symbol colors (dark blue and black) as, at the moment, they are very difficult to differentiate, visually. Many thanks.

- Please see my previous comments on further justifying the use of a common dataset to model vent-opening at the Nejapa-Miraflores fissure-vents system and the Apoyeque caldera. Many thanks.

Figure 2:

- I would recommend the inclusion here of some details on the approach to compute the 'thematic' vent-opening maps, not just the 'non-thematic' maps. Many thanks.

- If I understood correctly, the presence of the past vent in Figure 2b-Model 1 should modify the final vent-opening probability calculated using that model. Is this correct? If it is: would it be worthy to try show this more clearly with the 'example isolines' of vent-opening probability displayed in the figure? Many thanks.

Figures 8-9:

- Given the apparent low degree of PDC channelization that the topography of the area(s) suggests, combined with where the sectors of high vent-opening probability are located (e.g. on plain areas), I would suggest a brief justification for the use of the branching energy cone, instead of the 'classical' version. If it was feasible, running the analysis using this 'classical' version (also available in the code presented by Aravena et al., 2020a, right?), and compare the results with those of Figure 8 could be an interesting addition to the manuscript (even if it was included in the SI only). Many thanks.

Figure 10:

- L850 – "that would be derived"?

- L850 – I would change the last sentence of the caption to something like this: "in the assessment of volcanic hazard at volcanoes where eruptive style may significantly

change with vent location". Many thanks.

Table 2:

- I think there is an erratum in the title of the table (it should be "Table 2", right?). Many thanks.

Figure S1:

- If this figure is mathematically equivalent to Figure 5a, I would move the 5th- and 95th-percentile maps to the main text of the manuscript and I would then not include Figure S1 in the manuscript. Many thanks.

Figure S3:

- If this figure is mathematically equivalent to Figure 5b, I would move the 5th- and 95th-percentile maps to the main text of the manuscript and I would then not include Figure S3 in the manuscript. Many thanks.

**Suggested additional references**

Bebbington, M. S., Jenkins, S. F. (2019). Intra‐eruption forecasting. *Bulletin of Volcanology*, 81(6), 34.

Bonadonna, C., Connor, C. B., Houghton, B. F., Connor, L., Byrne, M., Laing, A., Hincks, T. K. (2005). Probabilistic modeling of tephra dispersal: Hazard assessment of a multiphase rhyolitic eruption at Tarawera, New Zealand. *Journal of Geophysical Research: Solid Earth*, 110(B3).

Clarke, B., Tierz, P., Calder, E., Yirgu, G. (2020). Probabilistic Volcanic Hazard Assessment for Pyroclastic Density Currents From Pumice Cone Eruptions at Aluto Volcano, Ethiopia. *Frontiers in Earth Science*, 8(348). http://dx.doi.org/10.3389/feart.2020.00348

Connor CB, Connor LJ (2009). Estimating spatial density with kernel methods. In:

Connor CB, Chapman NA, Connor LJ (eds) *Volcanic and tectonic hazard assessment for nuclear facilities*. Cambridge University Press, Cambridge, UK, pp 346–368.

Costa, A., Dell'Erba, F., Di Vito, M. A., Isaia, R., Macedonio, G., Orsi, G., Pfeiffer, T. (2009). Tephra fallout hazard assessment at the Campi Flegrei caldera (Italy). *Bulletin of Volcanology*, 71(3), 259-273.

Del Negro, C., Cappello, A., Neri, M., Bilotta, G., Hérault, A., and Ganci, G. (2013). Lava flow hazards at Mount Etna: constraints imposed by eruptive history and numerical simulations. *Sci. Rep.* 3:3493.

Deng, F., Connor, C. B., Malservisi, R., Connor, L. J., White, J. T., Germa, A., et al. (2017). A geophysical model for the origin of volcano vent clusters in a colorado plateau volcanic field. *J. Geophys. Res. Solid Earth* 122, 8910–8924. doi: 10.1002/2017jb014434

Grosse, P., Euillades, P. A., Euillades, L. D., de Vries, B. V. W. (2014). A global database of composite volcano morphometry. *Bulletin of Volcanology*, 76(1), 1-16.

Jenkins, S. F., Magill, C. R., McAneney, K. J. (2007). Multi‐stage volcanic events: A statistical investigation. *Journal of Volcanology and Geothermal Research*, 161(4), 275–288.

Marzocchi, W., Sandri, L., Selva, J. (2008). $BET_{EF} : a probabilistic tool for long- and short-term eruption forecasting.$ *Bulletin of Volcanology*, $70(5), 623-632.$

Mead, S. R., and Magill, C. R. (2017). Probabilistic hazard modelling of rain-triggered lahars. *J. Appl. Volcanol.* 6:8.

Newhall, C., and Hoblitt, R. (2002). Constructing event trees for volcanic crises. *Bull. Volcanol.* 64, 3–20. doi: 10.1007/s004450100173

Newhall, C. G., and Pallister, J. S. (2015). Using multiple data sets to populate probabilistic volcanic event trees, in *Volcanic Hazards, Risks, Disasters*, eds P. Papale, J. C.

Eichelberger, S. Nakada, S. Loughlin, and H. Yepes (Amsterdam: Elsevier), 203–232. doi: 10.1016/b978-0-12-396453-3.00008-3

Runge, M. G., Bebbington, M. S., Cronin, S. J., Lindsay, J. M., and Moufti, M. R. (2016). Integrating geological and geophysical data to improve probabilistic hazard forecasting of Arabian Shield volcanism. *J. Volcanol. Geotherm. Res.* 311, 41–59. doi: 10.1016/j.jvolgeores.2016.01.007

Sandri, L., Jolly, G., Lindsay, J., Howe, T., and Marzocchi, W. (2012). Combining long- and short-term probabilistic volcanic hazard assessment with cost-benefit analysis to support decision making in a volcanic crisis from the Auckland Volcanic Field, New Zealand. *Bull. Volcanol.* 74, 705–723. doi: 10.1007/s00445- 011-0556-y

Sandri, L., Thouret, J.-C., Constantinescu, R., Biass, S., and Tonini, R. (2014). Long-term multi-hazard assessment for El Misti volcano (Peru). *Bull. Volcanol.* 76, 1–26.

Sandri, L., Tierz, P., Costa, A., and Marzocchi, W. (2018). Probabilistic hazard from pyroclastic density currents in the neapolitan Area (Southern Italy). *J. Geophys. Res. Solid Earth* 123:890. doi: 10.1002/2017JB014890

Selva, J., Costa, A., Sandri, L., Macedonio, G., Marzocchi, W. (2014). Probabilistic short‐term volcanic hazard in phases of unrest: A case study for tephra fallout. *Journal of Geophysical Research: Solid Earth* 119(12), 8805-8826.

Sigmundsson, F., Hreinsdóttir, S., Hooper, A., Arnadóttir, T., Pedersen, R., Roberts, M. J., ... Feigl, K. L. (2010). Intrusion triggering of the 2010 Eyjafjallajökull explosive eruption. *Nature*, 468(7322), 426-430.

Strehlow, K., Sandri, L., Gottsmann, J. H., Kilgour, G., Rust, A. C., Tonini, R. (2017). Phreatic eruptions at crater lakes: occurrence statistics and probabilistic hazard forecast. *Journal of Applied Volcanology*, 6(1), 1-21.

Thompson, M. A., Lindsay, J. M., Sandri, L., Biass, S., Bonadonna, C., Jolly, G., et al. (2015). Exploring the influence of vent location and eruption style on tephra fall hazard

from the Okataina Volcanic Centre. New Zealand. *Bull. Volcanol.* 77:38.

Tierz, P., Sandri, L., Costa, A., Zaccarelli, L., Di Vito, M. A., Sulpizio, R., et al. (2016a). Suitability of energy cone for probabilistic volcanic hazard assessment: validation tests at Somma-Vesuvius and campi flegrei (Italy). *Bull. Volcanol.* 78:1073. doi: 10.1007/s00445-016-1073-9

Tierz, P., Sandri, L., Costa, A., Sulpizio, R., Zaccarelli, L., Vito, M. A. D., et al. (2016b). Uncertainty assessment of pyroclastic density currents at mount vesuvius (italy) simulated through the energy cone model, in *Natural Hazard Uncertainty Assessment: Modeling and Decision Support*, eds P. Webley, K. Riley, and M. P. Thompson (Hoboken: John Wiley  Sons), 125–145. doi: 10.1002/9781119028116.ch9

Tierz, P., Stefanescu, E. R., Sandri, L., Sulpizio, R., Valentine, G. A., Marzocchi, W., et al. (2018). Towards quantitative volcanic risk of pyroclastic density currents: probabilistic hazard curves and maps around Somma-Vesuvius (Italy). *J. Geophys. Res. Solid Earth* 123:383. doi: 10.1029/2017JB015383

Tierz, P., Clarke, B., Calder, E. S., Dessalegn, F., Lewi, E., Yirgu, G., ...  Loughlin, S. C. (2020). Event trees and epistemic uncertainty in long‐term volcanic hazard assessment of rift volcanoes: The example of Aluto (Central Ethiopia). *Geochemistry, Geophysics, Geosystems* 21(10), e2020GC009219.

Tonini, R., Sandri, L., Costa, A.,  Selva, J. (2015). Brief Communication: The effect of submerged vents on probabilistic hazard assessment for tephra fallout. *Natural Hazards and Earth System Sciences*, 15(3), 409-415.

Tonini, R., Sandri, L., Rouwet, D., Caudron, C., Marzocchi, W., and Suparjan, (2016), A new Bayesian Event Tree tool to track and quantify volcanic unrest and its application to Kawah Ijen volcano, *Geochem. Geophys. Geosyst.* 17, 2539– 2555, doi:10.1002/2016GC006327

Wolpert, R. L., Spiller, E. T.,  Calder, E. S. (2018). Dynamic statistical models for

pyroclastic density current generation at Soufrière Hills volcano. *Frontiers in Earth Science* 6, 55. https://doi.org/10.3389/feart.2018.00055

---

## Author Comment (AC1) · 24 Feb 2021

Thank you for your useful comments. We included all the suggestions. We included a new table (Table 3) for comparing the results described in Section Results. We also changed the contrast and label color of faults in Figs. 3, 4, 5, 8, 10, 11, S1, S2, S5 and S6. We also performed all the suggested modifications in Table 1 and in the main text.

---

## Author Comment (AC2) · 25 Feb 2021

Thank for your useful comments and suggestions. We followed most of your comments, as described in detail in the following pages (blue texts). We preferred not to follow two of your suggestions because we do not agree with them. Nevertheless, we thoroughly explained the reason behind that. In particular, we prefer to maintain the current structure of the manuscript with Appendixes rather than moving Appendixes to

the main text, as the main novelties of this work are not there. We also think that presenting a comparison of the results we obtained with the branching energy cone model respect to those that could be derived from the traditional formulation is completely beyond the scope of this paper, where this model is used just to illustrate the importance of adopting thematic maps. A comparison of these results has been already presented in detail in Aravena et al. 2020, clearly referenced in the text.

Moreover, you suggested to include thirty-one additional papers in the Bibliography, many of them repeated multiple times in the body text. We appreciate the effort to improve a literature review on PVHA, that however was not the purpose of this study. We included those that we found to be really significant additions in the context of the present paper.

We remark that all the modifications described here are already implemented in the text and figures (we did not include the file because it is not asked by the journal at this stage).

General comments

The manuscript presents a collection of 'thematic' maps for the probability of vent opening, given eruption, at San Salvador (El Salvador) and Nejapa-Chiltepe (Nicaragua) volcanic complexes. By 'thematic', the authors interpret that vent-opening-probability maps are built using data points (or areas, including epistemic uncertainty) of past vent locations, which are partitioned according to the occurrence of different hazardous phenomena during the associated eruptions. Additionally, the authors present maps of probability of invasion from pyroclastic density currents (PDCs), computed adopting the thematic description of the vent-opening probabilities, and compare them with those computed using 'non-thematic' (i.e. independent of hazardous phenomena) vent-opening models. They use the novel tree-branching energy cone model, recently developed and presented by some of the authors, to simulate PDC invasion.

I honestly think that the manuscript is a valuable contribution, as it proposes an interesting approach to explore spatial patterns in eruptive style, and could be complementary to previous and future studies tackling this complex problem, which is highly relevant to volcanic hazard assessment. Moreover, the initial probabilistic volcanic hazard assessment (PVHA), carried out at two volcanic systems with such high density of population on and around them, should represent vital information to manage volcanic risk in the area.

Thank you. We appreciate that.

My main reservations concerning the content and presentation of the manuscript are the following:

1. The main methodology of the manuscript, currently on Appendices A.1 and A.2 should be moved to the main text of the manuscript, and be better explained (perhaps, step by step), including an improved visual description of the methods in Figure 2 (or another figure).

We believe that moving the Appendix into the main text would decrease the paper readability. In fact, the other reviewer appreciates the current organization of the paper, that we decided after a thoughtful discussion between all the coauthors. Furthermore, despite the methods described in the Appendix are important to the description of our approach, the main novelties of this work are not related to that part of the paper. The methods are already fully described in several previous papers and the Appendix is mostly a summary, and highlights a few minor changes in some technicalities. However, we slightly expanded the explanation of the methods in the main text, also including more citations.

In relation to the methodology, further clarification should also be provided as regards: (1) whether the ballistics thematic maps are mathematically equivalent to the non-thematic maps; (2) why the thematic maps for lava emission appear to be quite similar to the ballistics/non-thematic maps, considering that the datasets for the two phenomena seem to be relatively different (Tables 1 and 2).

Yes, ballistics thematic maps are completely equivalent to the non-thematic maps because all the eruption observed at both volcanic systems are able to produce this phenomenon, and thus their weights in the thematic maps are the same (and equal to 1.0).

Regarding to the similarity of lava emission maps and ballistics/non-thematic maps, we have that the peaks of vent opening probability are located in the same zones, but the maps are not completely equivalent. The maximum vent opening probability is naturally located in the zones of high density of past vents (northern portion of fault A and the southern portion of NML, respectively). In the specific case of SSVC, this zone coincides with the zones where lava flows are concentrated, so the peaks coincide but the lava flows thematic maps present higher values in this peak. In the case of NCVC, a similar situation is observed, with most of the lava flows along NML, at south of Xolotlan Lake.

Accordingly, peaks are located in the same zone but maximum values differ. In other words, we agree, they are similar, but this is a consequence of the specific characteristics of these volcanic systems and it is not related to methodological, extrapolable considerations. We also want to repeat that the main message of the paper is related to the possibility and the importance to build "thematic maps of vent openinig probability" aimed at improving our capability of mapping any specific volcanic hazard in complex volcanic fields. Keeping this in mind, it is clear that the examples given in the manuscript are also used to show the potential of the method, despite the specific results of a single.

2. The justification of joining the Nejapa-Miraflores fissure-vents complex with the Apoyeque caldera, under a common, underlying data-generation process in terms of vent locations (as well as eruptive phenomena) should be, at least, expanded and made more clear, considering the significant differences in terms of geochemistry, eruption size and style at the two volcanic systems.

Thank you. This is an interesting point, similar to what we already discussed in the section about how to weight the sequences of lava flows that constructed the Boqueron volcano in SSVC. We believe that this should not be considered a methodological limitation in our approach, and we better clarified that. However, since we do not use an event-counting based approach, the effects of including the Apoyeque volcano are extremely restricted, as observed in the resulting vent opening maps (i.e. Apoyeque volcano does not present high vent opening probabilities in the different thematic maps). In fact, this is only 1 of 31 vents and the weight assigned is low (or zero) in 3 of 4 thematic maps. We added the following text: "We choose to consider the volcanic activity in Apoyeque volcano and in other zones of this volcanic system in a common framework to assess volcanic hazard. It is worth noting that the influence of this assumption in the analysis of small-scale events is limited because of the restricted influence of a single vent within the entire volcanic system, and because the weight assigned to the different volcanic phenomena at Apoyeque caldera is null or small in most of the cases (for three of the four considered volcanic phenomena).". From a volcanological point of view, although volcanoes of the Nejapa-Miraflores lineament and those of the Chiltepe peninsula have different style amd magma composition, it us undoubtful that their activity was strongly intefingered in the recent past (Kutterolf et al., JVGR 2007), and that the tectonic structures controlling these volcanoes are strictly interrelated (the Xiloa maar being placed just at the tip of the two structures). Dealing with a map aimed at defining probability of vent opening for volcanoes in the area of Managua, to be used for volcanic hazard assessment in that area, we are confident that the approach used is the most efficient. This is particularly true in the light of our suggestion of tracing thematic maps of vent opening probability for different hazards and/or styles of activity.

3. Discussion could be enriched around the differences between the approach of calculating P(vent location | hazardous phenomena), presented here; and that of calculating P(hazardous phenomena | vent location), which is more common in volcanic hazard assessments using event tree models (e.g. Tonini et al., 2014; Thompson et al., 2015; Sandri et al., 2018; Tierz et al., 2020).

Thank you. This is a very interesting suggestion. We added this comment in the discussion including some of the citations suggested: "It is worth highlighting that the approach adopted here, where the resulting vent opening maps are conditioned on the occurrence of a specific volcanic phenomenon, is opposite from the strategy adopted in a series of studies of hazard assessment (e.g. Neri et al., 2015; Thompson et al., 2015; Bevilacqua et al., 2017a; Tierz et al., 2020), where the probability of having a specific hazardous phenomenon or a specific eruption size is conditioned on vent location. Although both alternatives can be useful for hazard assessment, their suitability is controlled by the expected application, and the two approaches are linked through the application of the Bayes Theorem. Our approach seems to be more appropriate for the construction of thematic hazard maps for a specific volcanic phenomenon and eruption size, while the opposite strategy, possibly coupled with event trees, seems to suit better in the formulation of hazard maps that combine multiple phenomena and/or multiple eruption sizes.".

4. The justification of using the tree-branching energy cone model should be also expanded, given the expected radial dispersal of PDCs related to phreatomagmatic activity, and the apparent limited channelization that the flows may be subject to (judging, very preliminarily, by the topography of the two volcanic systems). If possible, comparison of results obtained with the 'classical' (non-branching) energy cone could be an interesting addition to the manuscript.

Although the topography may imply limited channelization processes, that is not a point against the use of the branching energy cone. In fact, in absence of channelization processes, the branching formulations tends to coincide with the traditional formulation and thus this is not a relevant element to discuss its applicability. Moreover, the focus of this study is not on the analysis of the properties of the branching energy cone model versus those of the traditional formulation. This is widely discussed already in Aravena et al. (2020) and another comparison would be a repetition of those results. The main target of this study is the illustration of the use of thematic maps, which is the reason for
*using a model to describe an example of hazard maps. To address this point we clarify that channelization is not expected to be large and that under these conditions, the branching formulation tends to be similar to the traditional model, but not completely equal (in particular, in section 4).*

The rest of my comments mostly relate to more specific points and suggestions, which I believe could help improve the manuscript. In summary, I would support the acceptance of this contribution to Natural Hazards and Earth System Sciences, after major revisions have been made.

Receive my best regards,

Pablo Tierz

*Most of the specific comments were introduced in the new version of the manuscript, as described below.*

Specific comments

L16 – Change to "Densely inhabited cities are built on them and their surroundings"?

*Modified.*

L18-19 – Perhaps explain here that the novelty is on the 'thematic' representation of the vent-opening maps.

*Modified. We moved the sentence associated with the thematic nature of maps.*

L26-27 – Considering that PDC channelization does not appear to be crucial at the two volcanic systems presented (judging, preliminarily, by the topography in the DEMs shown in Figure 1), the manuscript should better justify the use of the branching-version of the energy cone, instead of the 'classical' one. If possible, performing an analysis with the classical energy cone at San Salvador Volcanic Complex, to compare the results from the two versions, would be an interesting addition to the manuscript, even if this analysis had to be included in the Supplementary Information (SI). Many thanks.

Please see general answer.

L26-28 – OK, but one could also argue that the aleatory uncertainty in PDC generation (i.e. source conditions, including vent location) could be explored using a non-thematic vent-opening map coupled with spatially-varying probability density functions (PDFs) for the energy cone model parameters. This could describe how 'external factors' (e.g. groundwater) might influence the eruptive style (e.g. phreatomagmatism) and, there-fore, the types of PDCs generated (modeled via different model parameter values, cf. Tierz et al., 2016a). While I understand that this is beyond the scope of this manuscript, I think a richer discussion on these aspects should be incorporated. In the end, the au-thors are proposing to calculate the probability of vent opening conditional on eruptive style (i.e. P(vent | style) while most of the previous approaches in the literature (e.g. event-tree models) have devised the opposite conditional probabilities: that is, prob-ability of eruptive style given vent location: P(style | vent). In my opinion, this aspect merits further discussion in the manuscript. Many thanks.

Thank you. In fact, the two approaches are linked through the application of the Bayes Theorem, but we believe that both alternatives can be useful for hazard assessment, and their suitability is controlled by the expected application. This discussion was in-cluded, as described in the general answer.

L30-31 – Please rephrase. Volcanic hazard assessment also includes the assessment of the probability of eruption or the probability of vent location. Many thanks.

We clarified the current formulation of the sentence by using "influenced" instead of "controlled" to emphasize that these are not the only parameters affecting volcanic hazard assessment.

L32-33 – Please add Newhall and Hoblitt (2002); Newhall and Pallister (2015); Sandri et al. (2014). Many thanks.

We added some of these citations.

[Figure]

L35-36 – I would diversify the references here to include other research groups and/or volcanic hazardous phenomena. You could include, for example: Bonadonna et al. (2005); Costa et al. (2009); Del Negro et al. (2013); Mead and Magill (2017); Strehlow et al. (2017); Gallant et al. (2018); Tierz et al. (2018). Many thanks.

We added Bonadonna et al. (2005), Del Negro et al. (2013), Mead and Magill (2017) and Strehlow et al. (2017).

L40 – Similarly to the previous comment, please expand/diversify the list of references. You could consider removing Alberico et al. (2002), as Bevilacqua et al. (2015) also analyze vent opening at Campi Flegrei. Additionally, you could include the following references: Connor et al. (2012); Thompson et al. (2015); Clarke et al. (2020). Many thanks.

We added Connor et al. (2012) and Thompson et al. (2015).

L47 – Please add Clarke et al. (2020) for Aluto volcano, in the Main Ethiopian Rift. Many thanks.

OK, added.

L50-51 – I agree with this statement, but I think it would be important to briefly comment here, and add a proper argumentation in the Discussion section, about the significant difficulties associated with interpreting and modeling the dependence of eruptive style on vent location (see e.g. Thompson et al., 2015; Tonini et al., 2015; Sandri et al., 2018; Tierz et al., 2020). Many thanks.

We added the following sentence to clarify : "However, there are significant difficulties associated with interpreting and modelling the dependence between eruption style and vent position (e.g. Thompson et al., 2015)."

L52 – Please use "hazard" instead of "risk" here. Many thanks.

Modified.

L57 – "is clearly influenced by their position (e.g. Tonini et al., 2015; Sandri et al., 2018)"

OK, we added (Tonini et al., 2015 ; Paris et al., 2019).

L66 – Please see my previous comments about the 'dichotomy' between quantifying P(vent | style) vs P(style | vent) (e.g. in event trees), and add some more details in the Discussion section. Many thanks.

Please see our general answer.

L70-71 – Is the sentence about the funding needed here within the body text of the manuscript? The statement is already included in the Acknowledgements section.

Yes, we believe it is was worth including it within the main text.

L88 – Maybe a brief description of how exactly the polygons were defined or calculated would be a good addition to the manuscript, even if it appears in the SI. Many thanks.

We think that this is a methodological consideration (already explained in section Methods, 3.1), while section in L88 is geological framework. We prefer to avoid redundance.

L90-91 – What is the source of this uncertainty? Poorly-preserved deposits? Buried crater structures? Other? Please briefly complement. Many thanks.

OK. We added this information within parenthesis.

L91-94 – Whether here and/or in the caption of Figure 1c-d, I think it would be important to add some explanation about the 'bifurcations' and the arrows in the 'stratigraphical/temporal trees' presented. Do the bifurcations represent events (units) identified on different stratigraphical sections? If they do: can (some of) the units be correlated across sections, beyond their stratigraphical position with respect to the regional markers? In the context of Stage III of volcanic activity at San Salvador, I think it could be beneficial to indicate which unit(s) is/are interpreted to represent the beginning of this most recent stage of activity. Many thanks.

In the caption, we indicate what bifurcarions mean. We also indicated the eruptions that separate the different stages of SSVC, including the most recent stage of activity. We remark that the spider-diagram mainly has a descriptive purpose and is not part of the vent opening model.

L100-101 – Are these 'probability levels' used as the 'weighting' to count a particular vent location as belonging to the datasets used to compute the different thematic maps? Many thanks for the clarification.

Yes, now it is explained.

L111 – "E-W"

Modified.

L112 – Change to something like: "is formed by a summit cone called Boqueron Volcano (BV), which is enclosed by the remnants of the volcanic edifice of San Salvador Volcano (SSV), which experienced a vertical collapse around 36 ka (Sofield, 1998). At least 25 monogenetic vents [...]"? Many thanks.

Modified.

L135-137 – Are you implying that this type of activity is unlikely to occur in the future?

Not necessarily, we are not saying so, for we do not have enough information to indicate that.

L150-151 – What is the recurrence period estimated for summit eruptions? Please report it, if it is available. Many thanks.

Of the order of one event every 1000 yr (clearly much higher than one event every 85 yr). However, the statistics is not robust so we prefer to avoid these numbers, which are poorly-constrained.

L154-155 – I think the merging of eruptive vent locations from the three stages of activity, considering the possibility of significant changes in the magmatic system after the end of each stage (as observed in changes in the eruptive dynamics and spatial location of vents), deserves an expanded justification. This choice implies the assumption of a common, underlying data-generation process for vent locations at SSVC over the last 70 ka, and it represents the basis for the statistical modeling adopted henceforth. In my opinion, it would be beneficial to present two different hypotheses/assumptions: one that considers all vent locations; another that considers only vent locations during the last stage of volcanic activity. You should clearly describe, for any future reader of the manuscript, which implications each of the hypotheses/assumptions has (e.g. stationarity of vent locations in time). Many thanks.

This is wrong. We are not considering the last 70 ka, but the last 36 ka for SSVC – our analysis starts with the end of Stage I, characterized by the collapse of SSV. In fact, we believe that this event significantly changed the volcanic structure. The last stage is only 3 kyrs long, not enough to be representative of the long-term eruption statistics. Indeed the recent activity did not leave the zones that were active in the second stage - the Boqueron volcano is currently active.

L156-158 – Please clarify how these probabilities of occurrence are exactly used in the thematic vent-location datasets, to run Model 2 shown in Figure 2. As specified in my general comments, moving Appendices A.1 and A.2 to the main text of the manuscript may be necessary. Many thanks.

Please see the general answer (in particular, in section 3).

L173-174 – Given the substantial differences in magma geochemistry, eruption sizes and styles, and even spatial distribution of eruptive vents between the Nejapa Miraflores fissure-vents system and the Apoyeque caldera, some extra explanation would be needed about why they are treated as having a common, underlying datageneration process in terms of vent locations. This should be carefully explained and justified in the manuscript. Many thanks.

[Figure]

Thank you, now it is discussed. Please see the general answer.

L186-187 – Please rephrase slightly: the difference between 28 past vent locations and 31 does not seem critical. Could you also expand on which vents were not included by Connor et al. (2019) and why you included them? Many thanks.

We deleted "solely" to not to suggest that the number is the most critical difference. Unfortunately the paper Connor et al., (2019) lacks of detailed volcanological information and does not include the datasets, so it is rather difficult to detail which vents they missed.

L187 – Uncertainty in vent locations: a few extra details on how you defined the polygons that define this source of uncertainty would be appreciated. Many thanks.

Please see section 3.

L192 – It is not clear how the current presence of water relates to the eruptive phenomenology at the time of each eruption. Please expand. Many thanks.

OK. We extended the sentence.

L195-199 – Please see my previous comment on how the justification of mixing the Nejapa-Miraflores and Apoyeque volcanic systems under a common underlying data-generation process deserves substantial further explanation and justification. Many thanks.

Please see general answer.

L202-205 – Please add a few references for previous similar approaches, from which the ones that are cited built upon: e.g. Connor and Hill (1995); Martin et al. (2004); Connor and Connor (2009); Bebbington and Cronin (2011); Connor et al. (2012); Cappello et al. (2012). I saw that many of this references are included in Appendices A.1 and A.2. This could be another reason to move these Appendices to the main text of the manuscript. Many thanks.

We added Martin et al., 2004; Jaquet et al., 2012 and Tadini et al. 2017b. Other explanaiton were preserved in the Appendix (see general answer).

L206-207 – If Model 1 considers the past vent locations: are these vent locations used twice (Model 1 and 2) in generating the final vent-opening-probability maps? Please clarify. Many thanks.

Yes, the vent locations are used in both models. This is not a methodological issue. The differences between the methods are not in the dataset, but in the mathematical processing of it. Please see Figure 2.

L210 – Please report the spatial resolution of the vent-opening computational grid. Many thanks.

100 x 100 m, now it is indicated.

L212-214 – As a key novelty, it would be very beneficial if Figure 2 could incorporate some explanation about the new methodology you present here for the thematic maps. Many thanks.

This information is already included in Tables 1 and 2. The weighting is linear and hence mathematically trivial. We think that the objective of Figure 2 is different, that is, explaining the nontrivial differences between the two models 1 and 2.

L214 – Change "leads" to "lead"

Modified.

L215-216 – Specify that the continuous PDF is bivariate (Easting-Northing)?

No, the kernel is not bivariate. We are not assuming the same preferential alignment direction is valid everywhere in the volcanic field. The vent alignments spatially changes, especially in SSVC. We remark that in Model 1 the orientation of faults breaks central symmetry around the past vents uncertainty areas. Please see Figure 2.

L216-218 – What is the area of each grid point in the computational grid? I wonder if providing the values of vent-opening probability by grid point (instead of per km2) could be easier to interpret by the readers. Also, could you specify whether the thematic ventopening probabilities integrate to 1, by hazardous phenomenon? That is: given, say, a lava flow (emission) to occur in one of the volcanic systems: is the (mean) probability of a vent opening, summed across the computational grid, considered equal to 1? I interpret from the Appendices that this is the case, but some clarification could be welcome around these lines. Many thanks.

We think that the best way to present this information is density distribution of probability, as adopted here. Computational grid resolution is 100 x 100 m, and hence the sum of the computational grid is 100 and not 1.

L218-219 – I think this is a very important point and perhaps it deserves a little extra explanation or justification here. Many thanks.

We extended the sentence.

L221 – Redundant text: "Jaquet et al."?

Modified.

L233 – How much the results could be expected to change if, say, p1 = Unif(0.6, 1)? Many thanks.

The boundary effects are unsignificant and the results do not significantly change under small modifications of the uncertainty ranges.

L236 – Would it be possible that the "unknown" fault structures in the region were not distributed uniformly, for instance, due to regional tectonics? Perhaps adding a brief comment about this aspect could be beneficial. Many thanks.

We used "assumed to be" to say expressly that this is an assumption. The not-uniform features due to mapped faults are well represented in the corresponding layer, while assigning new preferential directions based on regional tectonics not represented in past activity and mapped faults, would be arbitrary. We remark that Model 2 is symmetrical around past vent locations and so we are not excluding other possible alignments.

L238-240 – "[. . .] or likelihood based techniques (Bevilacqua et al., 2017a; Bevilacqua et al., 2018), and/or by carrying out further geophysical surveys in the area/region (e.g. Martin et al., 2004; Jaquet et al., 2012; Runge et al., 2016; Deng et al., 2017)".

Additional knowledge is always useful but the relationship between geophysical surveys and the definition of these parameters is not immediately evident.

L244 – Could you provide some more details on how many sub-regions and/or which ones? It is also interesting to see that the two volcanic systems, despite expected differences in terms of their local structural and tectonic conditions, have the same distribution for d1. Could you expand on this a bit? Many thanks.

Uncertanty affecting d1 is large – from 5 km to 10 km. It is the average distance between sub-parallel regional faults, which we found to be consistent between the two systems, but locally varying. We considered a small number of subregions, about 3-4 in each case, but their exact number is not significant, and 5 to 10 km is their envelope.

L253 – But f(x) is independent on fault structures, right? Please clarify. In addition, the fact that the past vent locations seem to be used in Model 1 should be made more evident. Many thanks.

We separated the dependence of f and g in two groups. The dependence of Model 1 on vent locations is clearly explained at the onset of section 3. We prefer not to be redundant.

L258-264 – Maybe it is better to present the non-thematic vent-opening maps first (as Figure 3) and then all the thematic maps for the two volcanic systems under study. If the ballistics thematic vent-opening map is equivalent to the non-thematic map, I would

suggest that the 5th- and 95th-percentile maps are included in the current Figure 5 and Figure S1 was not included in the manuscript. Many thanks.

Please see answer in Figure S1 (below in this document).

L266-267 – "N65W" and "N40W", right?

Modified.

L270-273 – It would be convenient to explain these calculations better: why 4 km? Where exactly is the "northern portion of fault A"? Which pixels are excluded (i.e. what "closer" means in this context)? Etc. Many thanks.

We do not have a reason to prefer 4 km rather than 5 km, it is only to fix a common criterion in the order of magnitude of the value used for d1 in our model. This parameter has only a descriptive purpose and does not require sensitivity analysis. The pixels excluded are indicated. Closer means that the distance to other faults is lower.

L273 – "Sequel"?

We used "in the analysis of the other thematic maps".

L275 – "N65W"?

Modified.

L279-280 – Why these two maps are quite similar if the datasets for ballistics and lava flows appear to be relatively different in the two volcanic systems (Tables 1, 2)? Is it because the fault structures (known or unknown) have a very strong influence on the final vent-opening probabilities? If that is the case, the discourse about the thematic vent-opening maps, based on past vent locations, loses a bit of strength, in my opinion.

Please see general answer.

L283-284 – At some point, one may start to wonder if the stress on the thematic maps in the manuscript should be put almost exclusively on the 'small-PDCs-conditioned'

vent-opening maps, given that the ballistics thematic map is mathematically equal to the non-thematic map, and both the lava-emission and tephra-fallout thematic maps seem similar to the former. In other words, if the data-generation process for the vent locations of (small-)PDC-forming eruptions is the only one that seems to be significantly different from the non-thematic vent-location variability (the former perhaps linked to local conditions, such as groundwater availability or fluid pathways, influencing the eruptive style?): why not to focus on these aspects more clearly in the manuscript? Many thanks.

In fact, we present in the main manuscript small scale PDCs and lava flows because they present the main differences in the resulting maps, so we are already paying special attention to small scale PDCs. We prefer not to focus in PDCs because our main message is that the methodology is applicable also to other eruption phenomena and limiting the paper to PDCs is not consistent with that. From the point of view of the methods, we want to remain as general as possible.

L288-289 – Figure S3 may not be needed, if it is equivalent to Figure 5b. Many thanks.

See answer in Figure S1.

L292 – Please see my previous comment.

See previous answer.

L295 – Is the Miraflores scoria cone identified in Figure 1b?

Yes, it is vent 5.

L303-310 – This part could be moved to the Discussion section. If the visual comparison of the results from Connor et al. (2019) and those presented in this manuscript is important, perhaps a figure could be added to the SI, where the two vent-opening probability maps are displayed. Many thanks.

Connor et al., 2019 did not include enough information to precisely reproduce their

map, which is not expressed in term of probability values, but in terms of "vent per km2". The main purpose of that paper was decribing general methods with a "how to" approach, including pieces of pseudo-code, rather than preparing a robust hazard assessment. Despite this, it remains the only map presently available for the area, and we think that it deserves to be cited. We believe that our method includes several improvements, first of all the use of faults, and the uncertainty quantification efforts, and the new thematic approach. We already included a first comparison in the main text - a further visual comparison is made possible by collecting the original reference.

L306-307 – Please rephrase. A vent-opening probability map is a PDF, even if it only captures the aleatory uncertainty (i.e. single value of probability at each grid point) in the eruptive vent location. Many thanks.

Connor et al., 2019 did not present a probability density function, but instead a spatial density function, that lacks of any normalization step. Thus, they did not provide probability values. We could obtain some probability values by dividing their density function values by their total number of vents, 28, but we could not do this processing for their entire map, as they did not include the data in their publication, only a Figure with colors and contour lines. Please see Connor et al., 2019.

L312-320 – If I understood correctly, you extracted 100 realizations from the array of vent-opening model parameters (p1, p2, d1, d2), which generated 100 different values of vent-opening probability at each of the vent locations (1024). Is this correct? Then, you used these sets of vent-opening probabilities to weight the hazard footprints generated from running the branching-energy-cone model (cf. Sandri et al., 2018; Clarke et al., 2020, for the 'classical' energy cone), with fixed parameters independently of vent location. Is this correct? If it is, I do not understand why the total number of simulations is: 100 x no. of vents (1024), and not just 1 x 1024, as I picture vent-opening probabilities to be independent from PDC propagation. If any of the above is not correct, please clarify here and in the main text of the manuscript as well. Many thanks.

We performed 100x1024 simulations in order to investigate how the uncertainty in vent opening maps is propagated in the derived uncertainty in PDC inundation probability. In fact, we performed 100 Latin Hypercube samplings of 1024 spatial points, thus repeating the PDC simulations from 1024 points of origin, different for each realization of the vent opening map. We added : "The reason for running different sets of simulations for each volcanic system is to investigate the propagation of the uncertainty associated with our vent opening maps in the resulting probability maps of PDC inundation."

L319 – Please state the Digital Elevation Model (DEM) resolution over which the branching energy cone was run. Many thanks.

See answer in L331-332.

L320 – Please express $\varphi$ in degrees, additionally. Many thanks.

Modified.

L321 – Please expand on the links between the model parameters and the volcanological observations. What are the runout distances mentioned? From how many eruptions were they calculated? Were the associated eruptions large or small? Many thanks.

Now we indicated the typical runout distance, and we think that we already indicate the scale and the type of these eruptions.

L322-323 – "We decided not to use variable input conditions for initial PDC characteristics (e.g. Tierz et al., 2016a, b; Sandri et al., 2018; Aravena et al., 2020a; Clarke et al., 2020)"

We do not understand the references, in all those works, variable input conditions were adopted.

L325 - "would require additional information to properly calibrate these input parameters (e.g. Cioni et al., 2020) and/or complementary data coming from analogue volcanoes (e.g. Sandri et al., 2012; Tierz et al., 2016a; Clarke et al., 2020)"

Modified.

L327-328 - "(e.g., Spiller et al., 2014; Ogburn et al., 2016; Ogburn and Calder, 2017; Tierz et al., 2016b, 2018; Rutarindwa et al., 2019; Patra et al., 2020)."

We think Tierz et al. 2016b does not use a dataset of dependent input parameters. The other citations are already included.

L331-332 – What is the spatial resolution of the hazard domain? Many thanks.

30 m, now it is indicated.

L335 – Consider removing "small-scale". Many thanks

Removed.

L337 – "Ciudad" or "Sitio"? Many thanks.

Modified.

L336-338 – In my view, it is not only the topographic barrier of Cerro El Picacho which is hindering PDC inundation over (the western areas of) San Salvador city. Very importantly, given that the highest thematic vent-opening probabilities (for small-scale PDCs) are located on the NW flank of the stratocone (outside the caldera rim of the SSV edifice) and beyond, onto the surrounding plain (Figure 4); PDCs generated from some of these vents would need to overcome the whole 14km-basal-width, 1.2-1.4km-height volcanic edifice! (NB. Morphology data from Grosse et al., 2014).

We agree. We are not saying that this is the only effect, but it is significant. We include a new phrase to include your comment: " and also by the low vent opening probability computed at the flank E of the volcano ".

L339-341 – I beg to disagree with this interpretation. In my opinion, the peak in thematic vent-opening probability for small-scale PDCs located 'between' Santa Tecla and San Salvador (and associated with vent 16, right? Figures 1 and 4) should be a more

important factor conditioning the probabilities of PDC invasion in this area. I wonder how the map displayed in Figure 8 would change if you conditioned the vent opening, spatially, to vent locations with a (mean) vent-opening probability of (say) 0.5

We agree, the maximum is not in Santa Tecla, we modified the description of results (also we modified the conclusions). We think that to conditionate vent opening to a specific threshold is highly arbitrary (we would not know how to justify such choice).

L339 – Consider removing "small-scale". Many thanks.

Removed.

L349 – Please change to "would be derived". Many thanks.

Modified.

L349-351 – I would rephrase the sentence as follows: "These results show clearly the relevance of using thematic vent opening maps in the assessment of hazard at volcanoes where eruptive style may significantly change with vent location". Many thanks.

Modified.

L356 – "at the NNE flank"?

Modified.

L357 – Change "parameter" to "probability"?

Modified.

L361 – Change "small" to "smaller"? Many thanks.

Modified.

L364 – Please remove "Poland and Anderson (2020)" from this list and use examples previously cited instead (e.g. L44-50), and/or some of the additional references

suggested below. Many thanks.

We believe that the original references are more appropriate as general reviews of PVHA and not just examples of vent opening probability maps.

L365 – Please change "fields" by "systems". Many thanks.

Modified.

L369 – This list of references should be more generic in terms of volcanic hazardous phenomena included, and wider in terms of research groups represented. Please add references like: Del Negro et al. (2013), Thompson et al. (2015), Sandri et al. (2014, 2018). Many thanks.

We added Del Negro et al. (2013) and Thompson et al. (2015).

L370 - "plugging-in"?

Modified.

L371-372 – Please add some references to previous relevant research in the topic: e.g. Marzocchi et al. (2008); Sandri et al. (2012); Selva et al. (2014); Tonini et al. (2016). Many thanks.

We added Selva et al. (2014).

L376-378 - "We remark that previous studies already produced vent opening maps devoted to specific types of eruptions (e.g. Plinian and sub-Plinian eruptions in Tadini et al., 2017b; or pumice-cone-forming eruptions in Clarke et al., 2020), to eruptions inside selected sub-regions (Bevilacqua et al., 2017b) or to a suite of pre-imposed eruptive scenarios (Ang et al., 2020)."

Modified.

L382 – "which has been reported for different volcanic systems (e.g. Andronico et al., 2005; Coppola et al., 2009; Sigmundsson et al., 2010; Sandri et al., 2012; Tierz et al.,

2020)"

We added some of these references.

L383 – "N65W-trending fault B" instead?

Yes, modified.

L384 – "N40W-trending fault A" instead?

Yes, modified.

L385-386 – This is interesting. Could you briefly expand on the possible reasons behind this lack of small-scale PDC activity from the central crater of Boqueron? Many thanks.

This would be extremely speculative. This is only based on the geological record.

L388 – I think the exact location of Managua Lake may not be indicated in Figure 1?

It is Xolotlan Lake. It has two names. Now it is indicated in the text.

L389-391 – This is mostly true for small-scale PDCs but it is more difficult to justify for any of the other three phenomena presented. Please try to expand your argumentation around this point here, and maybe in other parts of the manuscript. Many thanks.

We use "some" to clarify that differences are not evident always. For the reasons of the similarity in some of the maps, please see the general answer.

L394 – Consider changing as follows: "The design and implementation of mitigation strategies for volcanic risk are profoundly conditioned by the characteristics of the volcanic processes under consideration and the vent position."

Modified.

L409 - "sequel"?

We used " below ".

L410-414 – If I am not wrong, you are adopting this very same assumption by using the past vent locations across different eruptive stages, for instance, at SSVC. Is this correct?

We do not agree. Because the system is currently in a mostly monogenetic stage, if a past vent was polygenetic we are counting it once anyway. So we did not assume stationarity.

L414-416 – Please explain why did you not implement a similar approach in the presented study. And/or excluded pre-3ka vents from your analysis. Many thanks.

Because in this study there was not a migration of the main site of activity tens of km in the direction of a dike intrusion, but rather a change from a polygenetic stage in the central edifice, to a monogenetic stage in a volcanic field surrounding it. Please see response to L154-155.

L419-420 – One could also argue, then, that the spatial dataset used in the presented analysis may suffer from the same or very similar issues. How is the analysis influenced by 'buried vents', for instance? Many thanks.

We are saying that an event counting approach would be more significantly affected by underrecording than a vent counting approach. When an improved datased will be available, repeating the analysis could provide more accurate results in terms of event counting, but we believe that we are already capturing the vent opening spatial distribution well enough. In fact, we are not expecting to have many buried vents in zones far from the mapped vents.

L420-423 – I am not sure if I follow the reasoning presented here. If the stratocone has been the source of many eruptions, those may have produced many lava flows but also many PDC events, potentially. Moreover, monogenetic vents are interpreted to be the product of a single eruption, but that does not necessarily mean that they generated 1 lava flow or 1 PDC. In the end, all these considerations also depend on

the degree of (temporal) detail on which the eruptions, and their associated hazardous phenomena, are analyzed (e.g. Jenkins et al., 2007; Wolpert et al., 2018; Bebbington Jenkins, 2019).

We are interested in the next event and, because we are not currently during a sequence of activity at a specific location (either a monogenetic center or the stratovolcano), we assume that the number of repeated occurrences of PDCs and lava flows at the same vent, potentially during the same eruption, is not significant. An event counting approach, like the one we are showing as an example, would assume the opposite.

L426-434 – I wonder why not including the thematic vent-opening maps for small-scale PDCs in Figure 11, as the former are one of the main focuses of the manuscript. Many thanks.

We did not include the maps associated with small-scale PDCs because in this case the number of events of interest from the central event is limited and thus differences between the two approaches are small.

L431-434 – These sentences feel slightly disconnected and/or vague as they are now. Please improve the argumentation here. Many thanks.

We agree. We moved the previous sentence to remove this problem by linking " conversely " with the general argument.

L435-436 – Change "fields" to "systems"?

Modified.

L436-440 – Please note that what are presented as unique features of the presented volcanic systems could be said of many other volcanoes in the world. Consider rewording the sentence slightly. Many thanks.

We indicate expressly that this is a common feature in many volcanoes. Thank you.

L443 – Change "lavas" to "lava"

Modified.

L448 – "N65W-trending fault B" instead?

Modified.

L449 – "N40W-trending fault A" instead?

Modified.

L455-456 - "which implies major challenges in managing the volcanic risk associated with NCVC" may be better. Many thanks.

Modified.

L461-462 – You should introduce, and justify, more clearly early in the manuscript, why the use of the branching-energy-cone model is key in the context of the two volcanic systems analyzed, where PDC channelization does not appear to be a critical conditioning factor to PDC propagation (preliminarily judging by the DEMs of the areas). Many thanks.

We do not think that this is key in this context. This is an illustrative application of the branching energy cone model, with the novelty that this is the first systematic application. We could use the traditional energy cone model or depth-averaged models, but these tests are beyond the scope of this paper.

L461 – Please see my previous comment on clarifying where these 105 simulations exactly come from. Many thanks.

See answer in L312-320.

L470 – "Ciudad" or "Sitio"? Modified. It is Sitio.

L477 – "decreased"

Modified.

L482 – Change "hazards" to "risk"?

Modified.

L491 – Therefore, the past vent-location data are used twice in the final vent opening map explained in Figure 2: once in Model 1 and another in Model 2. Is this correct ? Could you briefly expand on the reasoning behind this choice? Many thanks.

Please see response to L206-207.

L496 – Please remove: "the outcrop of", and/or change it by "the surface projection of". Many thanks.

Modified.

L496-497 – How are these prior PDFs calculated? $z(x)$ in formula (3) is the prior probability of $\zeta$ or $\xi$? Symbols in the text and in formula (3) seem different. Are they referring to two different variables? Many thanks) at the vent location x = ($x_i$ , $y_i$). Is this correct? If it is, please explain better how this is calculated. Also: is z2 a Uniform PDF over the whole vent-opening spatial domain, right? Many thanks.

The prior pdfs are shortly described: "Namely, z=z1 is the distribution of mapped faults, and z=z2 is the distribution of unknown faults". In L244-245 they were already described as : "One layer, z1, is related to the mapped faults (i.e. fault outcrops), and the other layer, z2, is a uniform PDF representing the unknown (i.e. buried) faults.". We added a short sentence in the Appendix: "In particular, z1 is uniform inside a 100 m buffer along the fault outcrop, which is not sensibly affecting the results because it is an order of magnitude lower than d1."

The symbol $\xi$ is used as the integral variable for spatial location x. However, there was some ambiguity in the use of the symbols in formula (3), that we corrected it to improve clarity. Thanks for spotting this.

L500, 520 – Perhaps it would be useful to report the formula for the posterior PDFs: zj ($\zeta$|Di). It is not clear to me that it is shown. Many thanks.

OK. It was already shown, but after the symbol change it is more clear.

L505 – Given the key importance of the weights wi in the paper, I suggest that (parts of) Appendix A.1 are actually moved into the main text of the manuscript. Many thanks.

See general answer.

L514 - "assume that, on average, about 95

Modified.

L525-526 – Please rephrase slightly: e.g. "the expected distance between past and future eruptive vents". Many thanks.

Modified.

L530 – Similarly to my previous comment, the critical relevance of the weights wi in the results shown in the manuscript could justify moving (parts of) Appendix A.2 to the main text of the manuscript. Many thanks.

See general answer.

L531-533 – Could you please expand a bit on the meaning, reasoning or implications of your choice in the presented manuscript? Many thanks.

The bandwidth assume to estimate the bivariate distance so the Rayleigh distribution is more appropriate than a Gaussian. This is justa technical detail without the only consequence to be more mathematically rigorous.

Figures and Tables

Figure 1:

- It may be good to show where approximately within the 'timeline' of SSVC does the

time 3ka sit. Many thanks.

Ages are indicated in the figure and we prefer not to add a temporal indication because it would order eruptions whose stratigraphic relationships are not known (this is the reason of bifurcations). However, we added information about the three periods in the caption and about bifurcations meaning.

- I would consider changing, at least one of, the symbol colors (dark blue and black) as, at the moment, they are very difficult to differentiate, visually. Many thanks.

Thank you. We used yellow instead of blue.

- Please see my previous comments on further justifying the use of a common dataset to model vent-opening at the Nejapa-Miraflores fissure-vents system and the Apoyeque caldera. Many thanks.

See general answer.

Figure 2:

- I would recommend the inclusion here of some details on the approach to compute the 'thematic' vent-opening maps, not just the 'non-thematic' maps. Many thanks.

Please see the response to L212-214.

- If I understood correctly, the presence of the past vent in Figure 2b-Model 1 should modify the final vent-opening probability calculated using that model. Is this correct? If it is: would it be worthy to try show this more clearly with the 'example isolines' of vent-opening probability displayed in the figure? Many thanks.

We do not understand this suggestion. The example isolines are quite different.

Figures 8-9:

- Given the apparent low degree of PDC channelization that the topography of the area(s) suggests, combined with where the sectors of high vent-opening probability

are located (e.g. on plain areas), I would suggest a brief justification for the use of the branching energy cone, instead of the 'classical' version. If it was feasible, running the analysis using this 'classical' version (also available in the code presented by Aravena et al., 2020a, right?), and compare the results with those of Figure 8 could be an interesting addition to the manuscript (even if it was included in the SI only). Many thanks.

The analysis of the channelization properties of the model is not the scope of this paper, which was largelly discussed in Aravena el al. 2020. We think that this type of analysis is not meaninful in absence of calibration procedures to set input parameters and thus any conclusions would be strongly associated with the set of input parameters considered here. So we could say things that are only true in these specific conditions.

Figure 10:

- L850 – "that would be derived"?

Modified.

- L850 – I would change the last sentence of the caption to something like this: "in the assessment of volcanic hazard at volcanoes where eruptive style may significantly change with vent location". Many thanks.

Modified.

Table 2:

- I think there is an erratum in the title of the table (it should be "Table 2", right?). Many thanks.

Thank you, it was solved.

Figure S1:

- If this figure is mathematically equivalent to Figure 5a, I would move the 5th- and

95th-percentile maps to the main text of the manuscript and I would then not include Figure S1 in the manuscript. Many thanks.

We prefer not to follow this suggestion because it would increase the figure number (and we preferred to priorize other figures) and also it can produce confusion between thematic and non thematic maps, with a figure that is both of them at the same time. We prefer to present only thematic maps with the same format and the non-thematic map with other format, which is only used to compare.

Figure S3:

- If this figure is mathematically equivalent to Figure 5b, I would move the 5th- and 95th-percentile maps to the main text of the manuscript and I would then not include Figure S3 in the manuscript. Many thanks.

Please see previous answer.

Suggested additional references

Bebbington, M. S., Jenkins, S. F. (2019). IntraâAËŸ Reruption forecasting. ËĞ Bulletin of Volcanology, 81(6), 34.

Bonadonna, C., Connor, C. B., Houghton, B. F., Connor, L., Byrne, M., Laing, A., Hincks, T. K. (2005). Probabilistic modeling of tephra dispersal: Hazard assessment of a multiphase rhyolitic eruption at Tarawera, New Zealand. Journal of Geophysical Research: Solid Earth, 110(B3).

Clarke, B., Tierz, P., Calder, E., Yirgu, G. (2020). Probabilistic Volcanic Hazard Assessment for Pyroclastic Density Currents From Pumice Cone Eruptions at Aluto Volcano, Ethiopia. Frontiers in Earth Science, 8(348). http://dx.doi.org/10.3389/feart.2020.00348

Connor CB, Connor LJ (2009). Estimating spatial density with kernel methods. In:

Connor CB, Chapman NA, Connor LJ (eds) Volcanic and tectonic hazard assessment

for nuclear facilities. Cambridge University Press, Cambridge, UK, pp 346–368.

Costa, A., Dell'Erba, F., Di Vito, M. A., Isaia, R., Macedonio, G., Orsi, G., Pfeiffer, T. (2009). Tephra fallout hazard assessment at the Campi Flegrei caldera (Italy). Bulletin of Volcanology, 71(3), 259-273.

Del Negro, C., Cappello, A., Neri, M., Bilotta, G., Hérault, A., and Ganci, G. (2013). Lava flow hazards at Mount Etna: constraints imposed by eruptive history and numerical simulations. Sci. Rep. 3:3493.

Deng, F., Connor, C. B., Malservisi, R., Connor, L. J., White, J. T., Germa, A., et al. (2017). A geophysical model for the origin of volcano vent clusters in a colorado plateau volcanic field. J. Geophys. Res. Solid Earth 122, 8910–8924. doi: 10.1002/2017jb014434

Grosse, P., Euillades, P. A., Euillades, L. D., de Vries, B. V. W. (2014). A global database of composite volcano morphometry. Bulletin of Volcanology, 76(1), 1-16.

Jenkins, S. F., Magill, C. R., McAneney, K. J. (2007). Multi' Rstage volcanic events: ËĞ A statistical investigation. Journal of Volcanology and Geothermal Research, 161(4), 275–288.

Marzocchi, W., Sandri, L., Selva, J. (2008). BETEF: a probabilistic tool for long and short term eruption forecasting. Bulletin of Volcanology, 70(5), $623 - 632$.

Mead, S. R., and Magill, C. R. (2017). Probabilistic hazard modelling of rain-triggered lahars. J. Appl. Volcanol. 6:8.

Newhall, C., and Hoblitt, R. (2002). Constructing event trees for volcanic crises. Bull. Volcanol. 64, 3–20. doi: 10.1007/s004450100173

Newhall, C. G., and Pallister, J. S. (2015). Using multiple data sets to populate probabilistic volcanic event trees, in Volcanic Hazards, Risks, Disasters, eds P. Papale, J. C. Eichelberger, S. Nakada, S. Loughlin, and H. Yepes (Amsterdam: Elsevier), 203–232.

doi: 10.1016/b978-0-12-396453-3.00008-3

Runge, M. G., Bebbington, M. S., Cronin, S. J., Lindsay, J. M., and Moufti, M. R. (2016). Integrating geological and geophysical data to improve probabilistic hazard forecasting of Arabian Shield volcanism. J. Volcanol. Geotherm. Res. 311, 41–59. doi: 10.1016/j.jvolgeores.2016.01.007

Sandri, L., Jolly, G., Lindsay, J., Howe, T., and Marzocchi, W. (2012). Combining longand short-term probabilistic volcanic hazard assessment with cost-benefit analysis to support decision making in a volcanic crisis from the Auckland Volcanic Field, New Zealand. Bull. Volcanol. 74, 705–723. doi: 10.1007/s00445- 011-0556-y

Sandri, L., Thouret, J.-C., Constantinescu, R., Biass, S., and Tonini, R. (2014). Longterm multi-hazard assessment for El Misti volcano (Peru). Bull. Volcanol. 76, 1–26.

Sandri, L., Tierz, P., Costa, A., and Marzocchi, W. (2018). Probabilistic hazard from pyroclastic density currents in the neapolitan Area (Southern Italy). J. Geophys. Res. Solid Earth 123:890. doi: 10.1002/2017JB014890

Selva, J., Costa, A., Sandri, L., Macedonio, G., Marzocchi, W. (2014). Probabilistic short term volcanic hazard in phases of unrest: A case study for tephra fallout. ĔǦ Journal of Geophysical Research: Solid Earth 119(12), 8805-8826.

Sigmundsson, F., Hreinsdóttir, S., Hooper, A., Arnadóttir, T., Pedersen, R., Roberts, M. J., ... Feigl, K. L. (2010). Intrusion triggering of the 2010 Eyjafjallajökull explosive eruption. Nature, 468(7322), 426-430.

Strehlow, K., Sandri, L., Gottsmann, J. H., Kilgour, G., Rust, A. C., Tonini, R. (2017). Phreatic eruptions at crater lakes: occurrence statistics and probabilistic hazard forecast. Journal of Applied Volcanology, 6(1), 1-21.

Thompson, M. A., Lindsay, J. M., Sandri, L., Biass, S., Bonadonna, C., Jolly, G., et al. (2015). Exploring the influence of vent location and eruption style on tephra fall hazard

from the Okataina Volcanic Centre. New Zealand. Bull. Volcanol. 77:38.

Tierz, P., Sandri, L., Costa, A., Zaccarelli, L., Di Vito, M. A., Sulpizio, R., et al. (2016a). Suitability of energy cone for probabilistic volcanic hazard assessment: validation tests at Somma-Vesuvius and campi flegrei (Italy). Bull. Volcanol. 78:1073. doi: 10.1007/s00445-016-1073-9

Tierz, P., Sandri, L., Costa, A., Sulpizio, R., Zaccarelli, L., Vito, M. A. D., et al. (2016b). Uncertainty assessment of pyroclastic density currents at mount vesuvius (italy) simulated through the energy cone model, in Natural Hazard Uncertainty Assessment: Modeling and Decision Support, eds P. Webley, K. Riley, and M. P. Thompson (Hoboken: John Wiley Sons), 125–145. doi: 10.1002/9781119028116.ch9

Tierz, P., Stefanescu, E. R., Sandri, L., Sulpizio, R., Valentine, G. A., Marzocchi, W., et al. (2018). Towards quantitative volcanic risk of pyroclastic density currents: probabilistic hazard curves and maps around Somma-Vesuvius (Italy). J. Geophys. Res. Solid Earth 123:383. doi: 10.1029/2017JB015383

Tierz, P., Clarke, B., Calder, E. S., Dessalegn, F., Lewi, E., Yirgu, G., ... Loughlin, S. C. (2020). Event trees and epistemic uncertainty in long-term volcanic hazard assessment of rift volcanoes: The example of Aluto (Central Ethiopia). Geochemistry, Geophysics, Geosystems 21(10), e2020GC009219.

Tonini, R., Sandri, L., Costa, A., Selva, J. (2015). Brief Communication: The effect of submerged vents on probabilistic hazard assessment for tephra fallout. Natural Hazards and Earth System Sciences, 15(3), 409-415.

Tonini, R., Sandri, L., Rouwet, D., Caudron, C., Marzocchi, W., and Suparjan, (2016), A new Bayesian Event Tree tool to track and quantify volcanic unrest and its application to Kawah Ijen volcano, Geochem. Geophys. Geosyst. 17, 2539– 2555, doi:10.1002/2016GC006327

Wolpert, R. L., Spiller, E. T., Calder, E. S. (2018). Dynamic statistical models for pyroclastic density current generation at Soufrière Hills volcano. Frontiers in Earth Science 6, 55. https://doi.org/10.3389/feart.2018.00055

Please see the inidividual answer, where we indicate the included references in the new version of the manuscript.

---

## Author Comment (AC4) · 25 Feb 2021

Please note that we added the file with changes marked as a supplementary file in a new comment.

---

## Author Comment (AC5) · 25 Feb 2021

Please note that we added the file with changes marked as a supplementary file in a new comment.

---

## Referee Comment (RC3) · Pablo Tierz (Referee) · 23 Mar 2021

**General comments**

The manuscript presents a collection of 'thematic' maps for the probability of vent opening, given eruption, at San Salvador (El Salvador) and Nejapa-Chiltepe (Nicaragua) volcanic complexes. By 'thematic', the authors interpret that vent-opening-probability

maps are built using data points (or areas, including epistemic uncertainty) of past vent locations, which are partitioned according to the occurrence of different hazardous phenomena during the associated eruptions. Additionally, the authors present maps of probability of invasion from pyroclastic density currents (PDCs), computed adopting the thematic description of the vent-opening probabilities, and compare them with those computed using 'non-thematic' (i.e. independent of hazardous phenomena) vent-opening models. They use the novel tree-branching energy cone model, recently developed and presented by some of the authors, to simulate PDC invasion.

I honestly think that the manuscript is a valuable contribution, as it proposes an interesting approach to explore spatial patterns in eruptive style, and could be complementary to previous and future studies tackling this complex problem, which is highly relevant to volcanic hazard assessment. Moreover, the initial probabilistic volcanic hazard assessment (PVHA), carried out at two volcanic systems with such high density of population on and around them, should represent vital information to manage volcanic risk in the area.

I would like to sincerely thank the authors for the effort put on their first round of reviews. I am generally satisfied with most of their responses and modifications. However, I have a few additional comments that I think are important before the article can be accepted for publication. These are the following:

1. I think it is extremely important that the authors provide any future reader of the manuscript with a list of references that is as comprehensive and unbiased as possible. This list must reflect, and acknowledge, the research work developed by diverse groups of colleagues on the different topics covered by the article. In particular, I find that the omission of references to previous research on the first attempts to apply the (classical) energy cone model to PVHA of PDCs (i.e. Tierz et al., 2016a, b; Sandri et al., 2018), prevents the authors as well as any future readers of the manuscript from recognising that previous work, which I think is quite relevant in the presented manuscript. Many thanks.

2. I also think that it is necessary that the authors clearly state and discuss, not only the advantages of the methods, models and results presented, but also their potential limitations. This will help the future readers of the manuscript, especially those who come from slightly different backgrounds, to obtain a better picture of the problems presented, and partially tackled, in the submitted manuscript. Many thanks.

3. Finally, I keep thinking that including Appendices A.1 and A.2 in the main text of the manuscript (leaving A.3 as the only Appendix) would improve its readability. They are only 1.5-pages long and, in my opinion, they include the main novelty of the work, which is incorporating the eruptive style (hazardous phenomena) into the calculation of vent-opening probabilities. Many thanks.

The rest of my comments mostly relate to more specific points and suggestions, which I believe could help improve the manuscript. Hereinafter, responses to authors' comments in the file (https://doi.org/10.5194/nhess-2020-382-AC2) are given in italic, red text, and introduced as, e.g. 'C1', for reference to the pdf page in the aforementioned file. 'LR1' refers to the Line 1 in the revised manuscript (https://nhess.copernicus.org/preprints/nhess-2020-382/nhess-2020-382-AC3-supplement.pdf). If there was anything unclear, please do not hesitate to let me know. Many thanks.

In summary, I would support the acceptance of this contribution to Natural Hazards and Earth System Sciences, after minor revisions have been implemented on its first reviewed version.

Receive my best regards,
Pablo Tierz

**Specific comments**

C1 - Thank for your useful comments and suggestions. We followed most of your comments, as described in detail in the following pages (blue texts). We preferred not to

follow two of your suggestions because we do not agree with them. Nevertheless, we thoroughly explained the reason behind that. In particular, we prefer to maintain the current structure of the manuscript with Appendixes rather than moving Appendixes to the main text, as the main novelties of this work are not there. We also think that presenting a comparison of the results we obtained with the branching energy cone model respect to those that could be derived from the traditional formulation is completely beyond the scope of this paper, where this model is used just to illustrate the importance of adopting thematic maps. A comparison of these results has been already presented in detail in Aravena et al. 2020, clearly referenced in the text.

Moreover, you suggested to include thirty-one additional papers in the Bibliography, many of them repeated multiple times in the body text. We appreciate the effort to improve a literature review on PVHA, that however was not the purpose of this study. We included those that we found to be really significant additions in the context of the present paper.

We remark that all the modifications described here are already implemented in the text and figures (we did not include the file because it is not asked by the journal at this stage).

*Many thanks for your explanations. I agree with the point of not including a comparison with the results obtained using the classical energy cone. This was more a suggestion or curiosity, but I understand that it may be beyond the purpose of the manuscript. Nevertheless, I would like to stress that further comparisons between the branching and classical versions of the energy cone model would be desirable in the future, as PDC inundation drastically changes from one volcanic system to another, both in terms of PDC source conditions as well as due to the topographic surface. Hence, I would suggest that the results in Aravena et al. (2020) were a useful starting point, but not the final and conclusive point as regards comparisons between the branching and classical energy cone models. Many thanks.*

*Concerning the references, I thought that it was convenient that the authors acknowledged, more clearly, the research work done by a variety of groups of scientific colleagues, in the different topics covered in the manuscript, e.g. vent-opening probability models, PVHA of different hazardous phenomena, and PVHA of PDCs. I would say that it is not a matter of the number of references, but their relevance in the context of the manuscript. In my opinion, the most important omissions are the references to previous work on PVHA of PDCs using the energy cone model (Tierz et al., 2016a, b; Sandri et al., 2018), which at the time were the first attempts to use the energy cone model for PVHA of PDCs. I would also like to note that Sandri et al. (2018), additionally, presented the application of an approach to model the effect of partially-submerged vents. In this respect, I find extremely surprising the deliberate omission of this reference in LR60-61 ("This is evident for example in partially submerged calderas, in which the style of activity of vents in the submerged zones is clearly influenced by their position"), as I had already suggested its inclusion during my first round of reviews. I sincerely think that the future readers of the manuscript should be aware of this previous research work. Many thanks.*

*Finally, in terms of Appendices A.1, A.2, please see my general comment above. I really think it would not be detrimental in terms of the length of the presented manuscript (they only represent 1.5 pages, in peer-review format); and it will give the future readers a more immediate hold to the main methodological novelty of the presented manuscript (i.e. including the hazardous-phenomena data in the vent-opening model, e.g. L533-541). Many thanks.*

C3 - We believe that moving the Appendix into the main text would decrease the paper readability. In fact, the other reviewer appreciates the current organization of the paper, that we decided after a thoughtful discussion between all the coauthors. Furthermore, despite the methods described in the Appendix are important to the description of our approach, the main novelties of this work are not related to that part of the paper. The

methods are already fully described in several previous papers and the Appendix is mostly a summary, and highlights a few minor changes in some technicalities. However, we slightly expanded the explanation of the methods in the main text, also including more citations.

*I will let the Associate Editor to decide on this aspect, but I hardly see how including Appendices A.1, A.2 inside Section 3.1 of the manuscript would decrease its readability. Those appendices are referred to twice inside the text of that section, which seems to suggest that this is important information for the reader to know at that point inside the manuscript. Personally, as one of the first 'external readers' of the manuscript, I would have obtained a much better picture of the methodology presented if those Appendices had been included in Section 3.1. Many thanks.*

C4 - Regarding to the similarity of lava emission maps and ballistics/non-thematic maps, we have that the peaks of vent opening probability are located in the same zones, but the maps are not completely equivalent. The maximum vent opening probability is naturally located in the zones of high density of past vents (northern portion of fault A and the southern portion of NML, respectively). In the specific case of SSVC, this zone coincides with the zones where lava flows are concentrated, so the peaks coincide but the lava flows thematic maps present higher values in this peak. In the case of NCVC, a similar situation is observed, with most of the lava flows along NML, at south of Xolotlan Lake.

Accordingly, peaks are located in the same zone but maximum values differ. In other words, we agree, they are similar, but this is a consequence of the specific characteristics of these volcanic systems and it is not related to methodological, extrapolable considerations. We also want to repeat that the main message of the paper is related to the possibility and the importance to build "thematic maps of vent openinig probability" aimed at improving our capability of mapping any specific volcanic hazard in complex volcanic fields. Keeping this in mind, it is clear that the examples given in

the manuscript are also used to show the potential of the method, despite the specific results of a single.

*Many thanks. I may have overlooked the differences in the case of SSVC. In the case of NCVC, I still see the maps as strikingly similar. I wonder if adding a few more isolines to the maps (perhaps at the cost of reducing the text font size for the isoline values?) could help the reader appreciate the differences between the non-thematic/ballistics and the lava-flows maps more clearly.*

*There is also the case of the low-intensity tephra-fallout maps, which to me, look hard to differentiate from the non-thematic/ballistics thematic maps. I suppose that my point was that, out of the four thematic maps presented (ballistics, lava flows, tephra fallout and PDCs), one is mathematically equivalent to the non-thematic map (ballistics) and another two (lava flows and tephra fallout) are very similar to the former. Accordingly, I think it would be informative for any future reader that these observations were clearly stated in the main text of the manuscript. Of course, this result is volcano-specific and may be completely different at another volcanic system under study. However, in the presented cases, I think it is important to stress the high relevance of considering thematic vent-opening maps for PDCs (i.e. an advantage of the method in the presented cases), but also to acknowledge the limited impact of considering the thematic vent-opening maps for lava flows and tephra fallout (i.e. somewhat a 'limitation' of the method for the presented cases). Many thanks.*

C5 - Thank you. This is an interesting point, similar to what we already discussed in the section about how to weight the sequences of lava flows that constructed the Boqueron volcano in SSVC. We believe that this should not be considered a methodological limitation in our approach, and we better clarified that. However, since we do not use an event-counting based approach, the effects of including the Apoyeque volcano are extremely restricted, as observed in the resulting vent opening maps (i.e. Apoyeque volcano does not present high vent opening probabilities in the different thematic maps).

In fact, this is only 1 of 31 vents and the weight assigned is low (or zero) in 3 of 4 thematic maps. We added the following text: "We choose to consider the volcanic activity in Apoyeque volcano and in other zones of this volcanic system in a common framework to assess volcanic hazard. It is worth noting that the influence of this assumption in the analysis of small-scale events is limited because of the restricted influence of a single vent within the entire volcanic system, and because the weight assigned to the different volcanic phenomena at Apoyeque caldera is null or small in most of the cases (for three of the four considered volcanic phenomena).". From a volcanological point of view, although volcanoes of the Nejapa-Miraflores lineament and those of the Chiltepe peninsula have different style amd magma composition, it us undoubtful that their activity was strongly intefingered in the recent past (Kutterolf et al., JVGR 2007), and that the tectonic structures controlling these volcanoes are strictly interrelated (the Xiloa maar being placed just at the tip of the two structures). Dealing with a map aimed at defining probability of vent opening for volcanoes in the area of Managua, to be used for volcanic hazard assessment in that area, we are confident that the approach used is the most efficient. This is particularly true in the light of our suggestion of tracing thematic maps of vent opening probability for different hazards and/or styles of activity.

*Many thanks. I was not suggesting that this was a methodological limitation of the 'thematic-vent-opening' approach but an interesting point about the volcanic hazard assessment presented in the manuscript. I still think it deserves a sentence or two in the main text of the manuscript, probably citing the relevant Kutterolf et al. (2007) reference. In my view, even if the mapped tectonic structures are closely related in space, the evident differences in terms of magma chemistry and eruption size/style would point to differences in the magmatic plumbing systems. One could expect that this has implications for vent opening and, therefore, it should be briefly discussed in the manuscript. Hence, in the text introduced in the reviewed version of the manuscript, a volcanological explanation is still lacking:*

*"We choose to consider the volcanic activity in Apoyeque volcano and in other*

*zones of this volcanic system in a common framework to assess volcanic hazard.* **[Volcanological explanation about the implications of this choice, including the possible limitations, e.g. decoupling in terms of magma geochemistry and eruption sizes/styles]. Nonetheless,** *it is worth noting that the influence of this assumption in the analysis of small-scale events is limited because of the restricted influence of a single vent within the entire volcanic system, and because the weight assigned to the different volcanic phenomena at Apoyeque caldera is null or small in most of the cases (for three of the four considered volcanic phenomena)."*

C6-C7 - Although the topography may imply limited channelization processes, that is not a point against the use of the branching energy cone. In fact, in absence of channelization processes, the branching formulations tends to coincide with the traditional formulation and thus this is not a relevant element to discuss its applicability. Moreover, the focus of this study is not on the analysis of the properties of the branching energy cone model versus those of the traditional formulation. This is widely discussed already in Aravena et al. (2020) and another comparison would be a repetition of those results. The main target of this study is the illustration of the use of thematic maps, which is the reason for using a model to describe an example of hazard maps. To address this point we clarify that channelization is not expected to be large and that under these conditions, the branching formulation tends to be similar to the traditional model, but not completely equal (in particular, in section 4).

*Please see my previous comments about how it is important that the presented manuscript cites previous research on PVHA of PDCs using the classical energy cone model, particularly if PDC channelization at the volcanoes under study is expected to be low and, hence, the behaviour of the branching energy cone model is expected to approximate that of the classical energy cone.*

*As I said in a previous comment, I understand that comparing the two formulations of the model is not the goal of the presented study. However, as I also said in a*

*previous comment, I find the assertion: "This is widely discussed already in Aravena et al. (2020) and another comparison would be a repetition of those results", very difficult to justify. Considering the strong variability in PDC source conditions and topographical surface among different volcanic systems, one may strongly argue that future comparisons between the branching and classical energy cone models will be anything but the repetition of the results presented in Aravena et al. (2020). Many thanks.*

C9 - We added the following sentence to clarify: "However, there are significant diffi-culties associated with interpreting and modelling the dependence between eruption style and vent position (e.g. Thompson et al., 2015)."

*LR55-56 - I still think that, in this context, Tierz et al. (2020) would be an informative reference for the future readers of your manuscript (please see Section 5.3 and Supplementary Material of that article, https://doi.org/10.1029/2020GC009219). Many thanks.*

C10 - OK, we added (Tonini et al., 2015 ; Paris et al., 2019).

*LR60-61 - I genuinely do not understand the deliberate omission of Sandri et al. (2018) in these lines.*

C12 - This is wrong. We are not considering the last 70 ka, but the last 36 ka for SSVC – our analysis starts with the end of Stage I, characterized by the collapse of SSV. In fact, we believe that this event significantly changed the volcanic structure. The last stage is only 3 kyrs long, not enough to be representative of the long-term eruption statistics. Indeed the recent activity did not leave the zones that were active in the second stage - the Boqueron volcano is currently active.

*LR158-167 – OK. In that case, please clearly specify this temporal (and spatial) constraint on the analysis in lines LR158-167. In my view, this is not straightforward to understand upon the first read of the main text of the manuscript alone. Many thanks.*

C14 - Yes, the vent locations are used in both models. This is not a methodological issue. The differences between the methods are not in the dataset, but in the mathematical processing of it. Please see Figure 2.

*My comment here wanted to address the fact that the vent positions have 'double' influence on your final vent-opening probability maps, as computed following the schematic shown in Figure 2. In other words, the dataset of vent locations is used 'twice' (in Models 1 and 2), while the dataset of fault locations is used only once (in Model 1). Is this correct? If it is, perhaps some comments on this choice could be added to the main text of the manuscript. Many thanks.*

C16 - Additional knowledge is always useful but the relationship between geophysical surveys and the definition of these parameters is not immediately evident.

*LR250-253 - Still, I think the future readers of your manuscript would benefit from knowing about approaches that try to link geophysical information with vent-opening probabilities in this point of the main text. Otherwise, it appears as if the future solutions to this complex problem could only be provided by the approaches adopted by the authors of the presented manuscript. Many thanks.*

C16 - We separated the dependence of f and g in two groups. The dependence of Model 1 on vent locations is clearly explained at the onset of section 3. We prefer not to be redundant.

*OK. But some clarification was still needed in LR265-266, which I think now reads*

*more clearly. Many thanks.*

C17 - We do not have a reason to prefer 4 km rather than 5 km, it is only to fix a common criterion in the order of magnitude of the value used for d1 in our model. This parameter has only a descriptive purpose and does not require sensitivity analysis. The pixels excluded are indicated. Closer means that the distance to other faults is lower.

*LR282-290 – Perhaps you just need to clarify the latest point in the main text, stating: "excluding the pixels that are closer to fault B **than to fault A**". This may apply to other occurrences of "closer", "near". Many thanks.*

C18 - In fact, we present in the main manuscript small scale PDCs and lava flows because they present the main differences in the resulting maps, so we are already paying special attention to small scale PDCs. We prefer not to focus in PDCs because our main message is that the methodology is applicable also to other eruption phenomena and limiting the paper to PDCs is not consistent with that. From the point of view of the methods, we want to remain as general as possible.

*Please see my previous comment about how it is important to stress the main advantages of the presented methods, and results, but also the potential limitations. The fact that 3 out of 4 thematic vent-opening maps presented are either identical or quite similar to the non-thematic vent-opening maps does not necessarily have to stand as a limitation of the methodology presented but, at least, it has to be discussed as a potential limitation of the presented results. Many thanks.*

C20 - We performed 100x1024 simulations in order to investigate how the uncertainty in vent opening maps is propagated in the derived uncertainty in PDC inundation probability. In fact, we performed 100 Latin Hypercube samplings of 1024 spatial points,

thus repeating the PDC simulations from 1024 points of origin, different for each re-alization of the vent opening map. We added : "The reason for running different sets of simulations for each volcanic system is to investigate the propagation of the uncer-tainty associated with our vent opening maps in the resulting probability maps of PDC inundation."

*OK. Many thanks for the clarification. Thus, if I understood correctly, your set of 1024 initiation points (i.e. vent locations) for the energy cone model change for each of the 100 LHS designs explored. As a consequence, the number of initiation points per grid cell across the vent-opening spatial domain may change from one design to another, and some grid cells may lack initiation points for a given LHS design. Is this correct? The approach presented here is more similar to that of, e.g., Rutarindwa et al. (2019) [for TITAN2D] and different from, e.g., Sandri et al. (2018) or Clarke et al. (2020), where initiation points for the energy cone model were fixed at the center of each grid cell across the vent-opening spatial domain. I think you should add a brief comment on this. Many thanks.*

C20 - Now we indicated the typical runout distance, and we think that we already indicate the scale and the type of these eruptions.

*Please note how phreatomagmatic PDC events with average runout distances of 2 km may not be optimal examples to model PDC channelization. I think the future readers of the manuscript should be clearly informed about these aspects. Many thanks.*

C20 - We do not understand the references, in all those works, variable input conditions were adopted.

*LR340-344 - Maybe I did not express myself properly in this comment. What I meant was acknowledging previous studies that had used variable input conditions for the energy cone model: "We decided not to use variable input conditions for initial PDC*

*characteristics, so our hazard assessment is only valid in this specific scenario of PDC size and friction angle **(see Tierz et al., 2016a, b; Sandri et al., 2018; Aravena et al., 2020a; Clarke et al., 2020; for hazard assessments that incorporated this variability into the energy cone modeling)**". Many thanks.*

C21 - Modified

*LR342-344 - Here, the authors deliberately choose to omit Tierz et al. (2016a) and Clarke et al. (2020), both of which use analogue-volcanoes data to parameterize the model used in the presented study (the energy cone), to perform "more complete PDC hazard assessment considering variable size and friction properties". On the contrary, they choose to cite Sandri et al. (2012), which I agree represents an interesting use of analogue volcanoes for PDC hazard assessment, but it does not use the model presented in the current manuscript. Please also note that Sandri et al. (2012) is omitted later on in LR390-392, in a context where I believe it is a very relevant reference to cite. I struggle to understand all this reasoning on the choice of references.*

C21 - We think Tierz et al. 2016b does not use a dataset of dependent input parameters. The other citations are already included.

*LR344-347 - Tierz et al. (2016b) did use a dataset of dependent input parameters to explore the importance of theoretical uncertainty (a source of epistemic uncertainty) on the outputs of the energy cone model (please see, for instance, Sections 9.2.6, 9.3.4 and Figures 9.4, 9.8 of the article,* https:// doi.org/ 10.1002/ 9781119028116.ch9*). Additionally, Tierz et al. (2018) adopted an inter-eruption-size dependence between the PDC volume and the bed friction angle at Somma-Vesuvius (Italy), and discussed the role of theoretical uncertainty for TITAN2D (please see Section 4.2 of the article,* https:// doi.org/ 10.1029/ 2017JB015383*). Many thanks.*

C21 - We agree. We are not saying that this is the only effect, but it is significant. We include a new phrase to include your comment: " and also by the low vent opening probability computed at the flank E of the volcano ".

*LR356-358 - Many thanks. Still, I would say that something like the following could be more appropriate:*

*"Modelled invasion probability at the SE flank of San Salvador volcano and in the city of San Salvador tends to be low, where a significant effect is exerted by the topographic barrier of Cerro El Picacho, **as well as of the entire volcanic edifice of SSVC** (Fig. 8), **given that vent-opening probabilities are highest on the NW flank of the volcano and on the surrounding plain to the N-NW of the edifice; and low on the E** flank of the volcano **(Fig. 8)**."*

C22 - We agree, the maximum is not in Santa Tecla, we modified the description of results (also we modified the conclusions). We think that to conditionate vent opening to a specific threshold is highly arbitrary (we would not know how to justify such choice).

*L358-361 – It is not clear to me how the description of the results has been modified around these lines. My suggestion of conditioning the vent-opening to the summit of SSVC, its N-NW flanks, and the surrounding plain in these directions was to address the plausibility of this assertion: "This is favoured by the high slope of the main edifice and the absence of significant topographic barriers in this direction".*

*If the high values of P(PDC) between Santa Tecla and San Salvador shown in Figure 8 decreased significantly when vent-opening was conditioned to the aforementioned areas, then the assertion would be difficult to support.*

*I think that, at least, you should reword the sentence in L358-361 following something similar to the text below. Many thanks.*

*"The highest PDC invasion probability calculated at the metropolitan area of San*

*Salvador, 2-3%, is located in its western sector, i.e. between the western sector of San Salvador and Santa Tecla. **These values might be explained by the peak in vent-opening probability shown in Figure 4 and/or by** the high slope of the main edifice and the absence of significant topographic barriers **on the southern sector of the SSVC edifice"***

C23 - We believe that the original references are more appropriate as general reviews of PVHA and not just examples of vent opening probability maps.

*LR383-384 - In my personal view, I still struggle with Poland and Anderson (2020) being the most appropriate reference to accompany the sentence: "Several probabilistic assessments of vent opening at different volcanoes have been presented during the last decade". As I suggested in my first review, I think there are other references which would fit better in this sentence. Many thanks.*

C23 - We added Del Negro et al. (2013) and Thompson et al. (2015).

*LR388-390 - In my opinion, omitting Sandri et al. (2014) from this list is unfortunate, as that article presents a variety of examples of "models aimed at describing the dispersal of volcanic products". Many thanks.*

C23 - We added Selva et al. (2014).

*LR390-392 - In my opinion, omitting Sandri et al. (2012) here is also quite unfortunate. The article is cited earlier in the text but it is in these lines where the reference is highly relevant. The article by Sandri et al. (2012) shows how vent-opening probabilities, and as a result PDC invasion probabilities, change in time as a result of changes in the recorded volcano monitoring data (please see Figures 3, and 6-8 of the article, https:// doi.org/ 10.1007/ s00445-011-0556-y). I think this is strongly relevant for the*

*content of the presented manuscript. Moreover, I think adding one of the other two references (either Marzocchi et al., 2008 or Tonini et al., 2016) could still be beneficial for the future readers of your manuscript. Many thanks.*

C25 - We are saying that an event counting approach would be more significantly affected by underrecording than a vent counting approach. When an improved datased will be available, repeating the analysis could provide more accurate results in terms of event counting, but we believe that we are already capturing the vent opening spatial distribution well enough. In fact, we are not expecting to have many buried vents in zones far from the mapped vents.

*OK. Many thanks for the clarification.*

C26 - We do not think that this is key in this context. This is an illustrative application of the branching energy cone model, with the novelty that this is the first systematic application. We could use the traditional energy cone model or depth-averaged models, but these tests are beyond the scope of this paper.

*OK. I understand that the thematic vent-opening approach could be implemented using different PDC models (and models for other volcanic hazardous phenomena as well). Nevertheless, given that you present a first systematic application of the branching energy cone model for PVHA, I sincerely believe that it is important that any future reader of your manuscript can be referred (earlier in the main text of the presented manuscript) to previous research that investigated the first-ever applications of the (classical) energy cone to PVHA of PDCs (Tierz et al., 2016a, b; Sandri et al., 2018). This was an interesting matter of scientific debate back in those years and, judging by the recent developments in the field, it will continue to be relevant over the years to come. Many thanks.*

C30 - We do not understand this suggestion. The example isolines are quite different.

*What I meant is that the example for Model 1 in Figure 2b seems to show well the influence of the presence of the fault on the isolines of vent-opening. However, it does not seem to show a clear influence of the presence of the past vent, which should also modify the isolines of vent-opening (compared to the case if the past vent was not there). Is this correct? I was just wondering if the isolines shown for Model 1 in Figure 2b could demonstrate the influence of **both** the fault line and the past vent on the vent-opening probabilities computed from Model 1. Many thanks.*

**Suggested additional references** *(please see the list given here, https://doi.org/10.5194/nhess-2020-382-RC2, for a few relevant references that are still missing from the presented manuscript. Many thanks)*

---

## Author Comment (AC6) · 26 Mar 2021

Here we answer each comment individually. In red and black we indicate the text present in the reviewer comments file, while in blue we present our answers. The marked manuscript is attached. In green we present the modifications of the first revision while in red we present the modifications associated with the second revision.

Thank you.

**Pablo Tierz, review 2 (Referee 3)**

**General comments**

The manuscript presents a collection of 'thematic' maps for the probability of vent open­ing, given eruption, at San Salvador (El Salvador) and Nejapa-Chiltepe (Nicaragua) volcanic complexes. By 'thematic', the authors interpret that vent-opening-probability maps are built using data points (or areas, including epistemic uncertainty) of past vent locations, which are partitioned according to the occurrence of different haz­ardous phenomena during the associated eruptions. Additionally, the authors present maps of probability of invasion from pyroclastic density currents (PDCs), computed adopting the thematic description of the vent-opening probabilities, and compare them with those computed using 'non-thematic' (i.e. independent of hazardous phenomena) vent-opening models. They use the novel tree-branching energy cone model, recently developed and presented by some of the authors, to simulate PDC invasion.

I honestly think that the manuscript is a valuable contribution, as it proposes an interest­ing approach to explore spatial patterns in eruptive style, and could be complementary to previous and future studies tackling this complex problem, which is highly relevant to volcanic hazard assessment. Moreover, the initial probabilistic volcanic hazard assess­ment (PVHA), carried out at two volcanic systems with such high density of population on and around them, should represent vital information to manage volcanic risk in the area.

This preamble repeats the first revision. Thanks again.

I would like to sincerely thank the authors for the effort put on their first round of reviews. I am generally satisfied with most of their responses and modifications. However, I have a few additional comments that I think are important before the article can be accepted for publication.

We appreciate the additional effort. We are carefully considering them.

These are the following:

1. I think it is extremely important that the authors provide any future reader of the manuscript with a list of references that is as comprehensive and unbiased as possible. This list must reflect, and acknowledge, the research work developed by diverse groups of colleagues on the different topics covered by the article. In particular, I find that the omission of references to previous research on the first attempts to apply the (classical) energy cone model to PVHA of PDCs (i.e. Tierz et al., 2016a, b; Sandri et al., 2018), prevents the authors as well as any future readers of the manuscript from recognising that previous work, which I think is quite relevant in the presented manuscript. Many thanks.

We added several new sentences including the references that you find that were missing (please see below). We remark that these papers are not the first attempts to apply the energy cone to PVHA of PDCs. Please see for example Sheridan and Macías 1995 (JVGR66, varying parameters probabilistically), Alberico et al. 2002 (JVGR116, using a vent opening map for the PDC source).

2. I also think that it is necessary that the authors clearly state and discuss, not only the advantages of the methods, models and results presented, but also their potential limitations. This will help the future readers of the manuscript, especially those who come from slightly different backgrounds, to obtain a better picture of the problems presented, and partially tackled, in the submitted manuscript. Many thanks.

We carefully evaluated your new specific comments below, and addressed them in most of the cases. Please see below.

3. Finally, I keep thinking that including Appendices A.1 and A.2 in the main text of the manuscript (leaving A.3 as the only Appendix) would improve its readability. They are only 1.5-pages long and, in my opinion, they include the main novelty of the work,

which is incorporating the eruptive style (hazardous phenomena) into the calculation of vent-opening probabilities. Many thanks.

As we already said in the previous review, " *we prefer to maintain the current structure of the manuscript with Appendixes rather than moving Appendixes to the main text, as the main novelties of this work are not there [. . .] We believe that moving the Appendix into the main text would decrease the paper readability. In fact, the other reviewer appreciates the current organization of the paper, that we decided after a thoughtful discussion between all the coauthors. Furthermore, despite the methods described in the Appendix are important to the description of our approach, the main novelties of this work are not related to that part of the paper. The methods are already fully described in several previous papers and the Appendix is mostly a summary, and high- lightes a few minor changes in some technicalities. However, we slightly expanded the explanation of the methods in the main text, also including more citations* ".

The rest of my comments mostly relate to more specific points and suggestions, which I believe could help improve the manuscript. Hereinafter, responses to au- thors' comments in the file (https://doi.org/10.5194/nhess-2020-382-AC2) are given in italic, red text, and introduced as, e.g. 'C1', for reference to the pdf page in the aforementioned file. 'LR1' refers to the Line 1 in the revised manuscript (https://nhess.copernicus.org/preprints/nhess-2020-382/nhess-2020-382-AC3 supple- ment.pdf). If there was anything unclear, please do not hesitate to let me know. Many thanks.

In summary, I would support the acceptance of this contribution to Natural Hazards and Earth System Sciences, after minor revisions have been implemented on its first reviewed version. Receive my best regards, Pablo Tierz

**Specific comments 1 (General Comments in the previous review)**

C1 - Thank for your useful comments and suggestions. We followed most of your com- ments, as described in detail in the following pages (blue texts). We preferred not to

follow two of your suggestions because we do not agree with them. Nevertheless, we thoroughly explained the reason behind that. In particular, we prefer to maintain the current structure of the manuscript with Appendixes rather than moving Appendixes to the main text, as the main novelties of this work are not there. We also think that presenting a comparison of the results we obtained with the branching energy cone model respect to those that could be derived from the traditional formulation is completely beyond the scope of this paper, where this model is used just to illustrate the importance of adopting thematic maps. A comparison of these results has been already presented in detail in Aravena et al. 2020, clearly referenced in the text.

Moreover, you suggested to include thirty-one additional papers in the Bibliography, many of them repeated multiple times in the body text. We appreciate the effort to improve a literature review on PVHA, that however was not the purpose of this study. We included those that we found to be really significant additions in the context of the present paper.

We remark that all the modifications described here are already implemented in the text and figures (we did not include the file because it is not asked by the journal at this stage).

*Many thanks for your explanations. I agree with the point of not including a comparison with the results obtained using the classical energy cone. This was more a suggestion or curiosity, but I understand that it may be beyond the purpose of the manuscript. Nevertheless, I would like to stress that further comparisons between the branching and classical versions of the energy cone model would be desirable in the future, as PDC inundation drastically changes from one volcanic system to another, both in terms of PDC source conditions as well as due to the topographic surface. Hence, I would suggest that the results in Aravena et al. (2020) were a useful starting point, but not the final and conclusive point as regards comparisons between the branching and classical energy cone models. Many thanks.*

Thank you.

*Concerning the references, I thought that it was convenient that the authors acknowledged, more clearly, the research work done by a variety of groups of scientific colleagues, in the different topics covered in the manuscript, e.g. vent-opening probability models, PVHA of different hazardous phenomena, and PVHA of PDCs. I would say that it is not a matter of the number of references, but their relevance in the context of the manuscript. In my opinion, the most important omissions are the references to previous work on PVHA of PDCs using the energy cone model (Tierz et al., 2016a, b; Sandri et al., 2018), which at the time were the first attempts to use the energy cone model for PVHA of PDCs.*

We added those three references, see below.

*I would also like to note that Sandri et al. (2018), additionally, presented the application of an approach to model the effect of partially submerged vents. In this respect, I find extremely surprising the deliberate omission of this reference in LR60-61 ("This is evident for example in partially submerged calderas, in which the style of activity of vents in the submerged zones is clearly influenced by their position"), as I had already suggested its inclusion during my first round of reviews. I sincerely think that the future readers of the manuscript should be aware of this previous research work. Many thanks.*

Differently from Tonini et al. 2015 (pyroclastic fallout) and Paris et al. 2019 (tsunami), Sandri et al., 2018 did not show any hazard map with the effect of submerged vents, although they tested that case.

They state " . . . *whatever the explosive size class,we first assume that PDCs are not generated if the vent opens at a depth larger than 10 m. [. . .] We then test the sensitivity of our results to such choice by adopting the empirical approach by Tonini, Sandri, Thompson (2015), that is, by assuming a linear decay in the probability of generating explosive activity, and thus PDCs, as the water depth increases. [. . .] We observe that*

*the changes in the final probability maps are negligible.* "

In particular, they did not assume any effect of magma-water interaction on the style of activity, but implemented a linear decrease in the probability of occurrence of PDCs with (current) water depth. We cited the paper as you suggest but also added a sentence to clarify that water can also increase explosivity (please see below).

*Finally, in terms of Appendices A.1, A.2, please see my general comment above. I really think it would not be detrimental in terms of the length of the presented manuscript (they only represent 1.5 pages, in peer-review format); and it will give the future readers a more immediate hold to the main methodological novelty of the presented manuscript (i.e. including the hazardous-phenomena data in the vent-opening model, e.g. L533-541). Many thanks.*

Please see our response above.

C3 - We believe that moving the Appendix into the main text would decrease the paper readability. In fact, the other reviewer appreciates the current organization of the paper, that we decided after a thoughtful discussion between all the coauthors. Furthermore, despite the methods described in the Appendix are important to the description of our approach, the main novelties of this work are not related to that part of the paper. The methods are already fully described in several previous papers and the Appendix is mostly a summary, and highlights a few minor changes in some technicalities. However, we slightly expanded the explanation of the methods in the main text, also including more citations.

*I will let the Associate Editor to decide on this aspect, but I hardly see how including Appendices A.1, A.2 inside Section 3.1 of the manuscript would decrease its readability. Those appendices are referred to twice inside the text of that section, which seems to suggest that this is important information for the reader to know at that point inside the manuscript. Personally, as one of the first 'external readers' of the manuscript, I would have obtained a much better picture of the methodology presented if those Appendices*

*had been included in Section 3.1. Many thanks.*

Please see our response above.

C4 - Regarding to the similarity of lava emission maps and ballistics/non-thematic maps, we have that the peaks of vent opening probability are located in the same zones, but the maps are not completely equivalent. The maximum vent opening probability is naturally located in the zones of high density of past vents (northern portion of fault A and the southern portion of NML, respectively). In the specific case of SSVC, this zone coincides with the zones where lava flows are concentrated, so the peaks coincide but the lava flows thematic maps present higher values in this peak. In the case of NCVC, a similar situation is observed, with most of the lava flows along NML, at south of Xolotlan Lake.

Accordingly, peaks are located in the same zone but maximum values differ. In other words, we agree, they are similar, but this is a consequence of the specific characteristics of these volcanic systems and it is not related to methodological, extrapolable considerations. We also want to repeat that the main message of the paper is related to the possibility and the importance to build "thematic maps of vent openinig probability" aimed at improving our capability of mapping any specific volcanic hazard in complex volcanic fields. Keeping this in mind, it is clear that the examples given in the manuscript are also used to show the potential of the method, despite the specific results of a single.

*Many thanks. I may have overlooked the differences in the case of SSVC. In the case of NCVC, I still see the maps as strikingly similar. I wonder if adding a few more isolines to the maps (perhaps at the cost of reducing the text font size for the isoline values?) could help the reader appreciate the differences between the non-thematic/ballistics and the lava-flows maps more clearly.*

We adopted the same algorithm to set the isolines in all maps, and in the case of NCVC some of the maps are similar. We believe that highliting too small differences would not

be meaningful, because we are speaking about maps affected by uncertainty. Please see next comment.

*There is also the case of the low-intensity tephra-fallout maps, which to me, look hard to differentiate from the non-thematic/ballistics thematic maps. I suppose that my point was that, out of the four thematic maps presented (ballistics, lava flows, tephra fallout and PDCs), one is mathematically equivalent to the non-thematic map (ballistics) and another two (lava flows and tephra fallout) are very similar to the former. Accordingly, I think it would be informative for any future reader that these observations were clearly stated in the main text of the manuscript. Of course, this result is volcano-specific and may be completely different at another volcanic system under study. However, in the presented cases, I think it is important to stress the high relevance of considering thematic vent-opening maps for PDCs (i.e. an advantage of the method in the presented cases), but also to acknowledge the limited impact of considering the thematic vent-opening maps for lava flows and tephra fallout (i.e. somewhat a 'limitation' of the method for the presented cases). Many thanks.*

OK. It's not helpful to use thematic maps if almost all past eruptions were characterized by a particular volcanic hazard. We added a sentence saying that more clearly (please see below).

C5 - Thank you. This is an interesting point, similar to what we already discussed in the section about how to weight the sequences of lava flows that constructed the Boqueron volcano in SSVC. We believe that this should not be considered a methodological limitation in our approach, and we better clarified that. However, since we do not use an event-counting based approach, the effects of including the Apoyeque volcano are extremely restricted, as observed in the resulting vent opening maps (i.e. Apoyeque volcano does not present high vent opening probabilities in the different thematic maps). In fact, this is only 1 of 31 vents and the weight assigned is low (or zero) in 3 of 4 thematic maps. We added the following text: "We choose to consider the volcanic activity in Apoyeque volcano and in other zones of this volcanic system in a common framework to assess volcanic hazard. It is worth noting that the influence of this assumption in the analysis of small-scale events is limited because of the restricted influence of a single vent within the entire volcanic system, and because the weight assigned to the different volcanic phenomena at Apoyeque caldera is null or small in most of the cases (for three of the four considered volcanic phenomena).". From a volcanological point of view, although volcanoes of the Nejapa-Miraflores lineament and those of the Chiltepe peninsula have different style and magma composition, it is undoubtful that their activity was strongly intefingered in the recent past (Kutterolf et al., JVGR 2007), and that the tectonic structures controlling these volcanoes are strictly interrelated (the Xiloa maar being placed just at the tip of the two structures). Dealing with a map aimed at defining probability of vent opening for volcanoes in the area of Managua, to be used for volcanic hazard assessment in that area, we are confident that the approach used is the most efficient. This is particularly true in the light of our suggestion of tracing thematic maps of vent opening probability for different hazards and/or styles of activity.

*Many thanks. I was not suggesting that this was a methodological limitation of the 'thematic-vent-opening' approach but an interesting point about the volcanic hazard assessment presented in the manuscript. I still think it deserves a sentence or two in the main text of the manuscript, probably citing the relevant Kutterolf et al. (2007) reference.*

*In my view, even if the mapped tectonic structures are closely related in space, the evident differences in terms of magma chemistry and eruption size/style would point to differences in the magmatic plumbing systems. One could expect that this has implications for vent opening and, therefore, it should be briefly discussed in the manuscript. Hence, in the text introduced in the reviewed version of the manuscript, a volcanological explanation is still lacking:*

*"We choose to consider the volcanic activity in Apoyeque volcano and in other zones of this volcanic system in a common framework to assess volcanic hazard. [Volcanological explanation about the implications of this choice, including the possible limitations,*

*e.g. decoupling in terms of magma geochemistry and eruption sizes/styles]. Nonetheless, it is worth noting that the influence of this assumption in the analysis of small-scale events is limited because of the restricted influence of a single vent within the entire volcanic system, and because the weight assigned to the different volcanic phenomena at Apoyeque caldera is null or small in most of the cases (for three of the four considered volcanic phenomena)."*

OK. We added the following additional sentence, including the missing reference, in the main text: " *Although volcanoes of the Nejapa-Miraflores lineament and those of the Chiltepe peninsula have different style and magma composition, it is undoubtful that their activity was strongly intefingered in the recent past (Kutterolf et al., 2007), and that the tectonic structures controlling these volcanoes are strictly interrelated.* "

C6-C7 - Although the topography may imply limited channelization processes, that is not a point against the use of the branching energy cone. In fact, in absence of channelization processes, the branching formulations tends to coincide with the traditional formulation and thus this is not a relevant element to discuss its applicability. Moreover, the focus of this study is not on the analysis of the properties of the branching energy cone model versus those of the traditional formulation. This is widely discussed already in Aravena et al. (2020) and another comparison would be a repetition of those results. The main target of this study is the illustration of the use of thematic maps, which is the reason for using a model to describe an example of hazard maps. To address this point we clarify that channelization is not expected to be large and that under these conditions, the branching formulation tends to be similar to the traditional model, but not completely equal (in particular, in section 4).

*Please see my previous comments about how it is important that the presented manuscript cites previous research on PVHA of PDCs using the classical energy cone model, particularly if PDC channelization at the volcanoes under study is expected to be low and, hence, the behaviour of the branching energy cone model is expected to approximate that of the classical energy cone.*

OK. We added those references (please see below).

*As I said in a previous comment, I understand that comparing the two formulations of the model is not the goal of the presented study. However, as I also said in a previous comment, I find the assertion: "This is widely discussed already in Aravena et al. (2020) and another comparison would be a repetition of those results", very difficult to justify. Considering the strong variability in PDC source conditions and topographical surface among different volcanic systems, one may strongly argue that future comparisons between the branching and classical energy cone models will be anything but the repetition of the results presented in Aravena et al. (2020). Many thanks.*

We believe that if the interaction with topography is absent/minor, then the PDC may be inertia-dominated and the energy cone would not physically appropriate. A box model approach would be more appropriate in that case. We're glad that you agree that an additional model comparison was not the goal of this study.

**Specific comments 2 (Specific Comments also in the previous review)**

C9 - We added the following sentence to clarify: "However, there are significant difficulties associated with interpreting and modelling the dependence between eruption style and vent position (e.g. Thompson et al., 2015)."

*LR55-56 - I still think that, in this context, Tierz et al. (2020) would be an informative reference for the future readers of your manuscript (please see Section 5.3 and Supplementary Material of that article, https:// doi.org/ 10.1029/ 2020GC009219). Many thanks.*

OK, we added your reference.

C10 - OK, we added (Tonini et al., 2015 ; Paris et al., 2019).

*LR60-61 - I genuinely do not understand the deliberate omission of Sandri et al. (2018) in these lines.*

We added the following sentence, including that reference: " *...by their position either increasing the explosivity by magma-water interaction, or reducing explosivity in deep water conditions (Tonini et al., 2015; Sandri et al., 2018; Paris et al., 2019)* ". Please see the previous comments for an explanation of the previous omission.

C12 - This is wrong. We are not considering the last 70 ka, but the last 36 ka for SSVC – our analysis starts with the end of Stage I, characterized by the collapse of SSV. In fact, we believe that this event significantly changed the volcanic structure. The last stage is only 3 kyrs long, not enough to be representative of the long-term eruption statistics. Indeed the recent activity did not leave the zones that were active in the second stage - the Boqueron volcano is currently active.

*LR158-167 – OK. In that case, please clearly specify this temporal (and spatial) constraint on the analysis in lines LR158-167. In my view, this is not straightforward to understand upon the first read of the main text of the manuscript alone. Many thanks.*

OK. We added the following sentence: " *In this study, we focused on the last 36 ka of volcanic activity at SSVC (i.e. our analysis starts with the end of Stage I, when the collapse of SSV occurred). In particular...* ".

C14 - Yes, the vent locations are used in both models. This is not a methodological issue. The differences between the methods are not in the dataset, but in the mathematical processing of it. Please see Figure 2.

*My comment here wanted to address the fact that the vent positions have 'double' influence on your final vent-opening probability maps, as computed following the schematic shown in Figure 2. In other words, the dataset of vent locations is used 'twice' (in Models 1 and 2), while the dataset of fault locations is used only once (in Model 1). Is this correct? If it is, perhaps some comments on this choice could be added to the main text of the manuscript. Many thanks.*

Both Models are using past vent locations. Only Model 1 uses the fault locations. The

current version of the manuscript already clarifies this fact. Please see the following text from the beginning of section 3: " . . . *Model 1 considers the faults and the position of past vents following a Bayesian approach, [. . .] On the other hand, Model 2 [. . .] does not consider structures but only a measure of the expected distance between past and future vent positions. We remark that in both the models past vents do not comprise simple points, but areas of uncertainty of different extent.* "

C16 - Additional knowledge is always useful but the relationship between geophysical surveys and the definition of these parameters is not immediately evident.

*LR250-253 - Still, I think the future readers of your manuscript would benefit from knowing about approaches that try to link geophysical information with vent-opening probabilities in this point of the main text. Otherwise, it appears as if the future solutions to this complex problem could only be provided by the approaches adopted by the authors of the presented manuscript. Many thanks.*

OK. However, this study is about long-term analyses and should not be mixed with short-term monitoring signals/geophysical surveys. We added the following sentence: " . . . *and/or through other geophysical models able to address regional volcanism (e.g. Martin et al., 2004; Jaquet et al., 2012; Runge et al., 2016; Deng et al., 2017)* ".

C16 - We separated the dependence of f and g in two groups. The dependence of Model 1 on vent locations is clearly explained at the onset of section 3. We prefer not to be redundant.

*OK. But some clarification was still needed in LR265-266, which I think now reads more clearly. Many thanks.*

You're welcome.

C17 - We do not have a reason to prefer 4 km rather than 5 km, it is only to fix a common criterion in the order of magnitude of the value used for d1 in our model. This parameter has only a descriptive purpose and does not require sensitivity analysis.

The pixels excluded are indicated. Closer means that the distance to other faults is lower.

*LR282-290 – Perhaps you just need to clarify the latest point in the main text, stating: "excluding the pixels that are closer to fault B than to fault A". This may apply to other occurrences of "closer", "near". Many thanks.*

OK. We've done that.

C18 - In fact, we present in the main manuscript small scale PDCs and lava flows because they present the main differences in the resulting maps, so we are already paying special attention to small scale PDCs. We prefer not to focus in PDCs because our main message is that the methodology is applicable also to other eruption phenomena and limiting the paper to PDCs is not consistent with that. From the point of view of the methods, we want to remain as general as possible.

*Please see my previous comment about how it is important to stress the main advantages of the presented methods, and results, but also the potential limitations. The fact that 3 out of 4 thematic vent-opening maps presented are either identical or quite similar to the non-thematic vent-opening maps does not necessarily have to stand as a limitation of the methodology presented but, at least, it has to be discussed as a potential limitation of the presented results. Many thanks.*

OK, we added the following sentence clarifying that some maps are indeed similar, because data are not highlighting otherwise: " *In general, we remark that if almost all past eruptions are characterized by a common volcanic hazardous phenomenon (e.g. the small scale fallout), the corresponding thematic map will be very similar to the non-thematic map (Fig. S2).*"

C20 - We performed 100x1024 simulations in order to investigate how the uncertainty in vent opening maps is propagated in the derived uncertainty in PDC inundation probability. In fact, we performed 100 Latin Hypercube samplings of 1024 spatial points,

thus repeating the PDC simulations from 1024 points of origin, different for each re-alization of the vent opening map. We added : "The reason for running different sets of simulations for each volcanic system is to investigate the propagation of the uncer-tainty associated with our vent opening maps in the resulting probability maps of PDC inundation."

*OK. Many thanks for the clarification. Thus, if I understood correctly, your set of 1024 initiation points (i.e. vent locations) for the energy cone model change for each of the 100 LHS designs explored. As a consequence, the number of initiation points per grid cell across the vent-opening spatial domain may change from one design to another, and some grid cells may lack initiation points for a given LHS design. Is this correct? The approach presented here is more similar to that of, e.g., Rutarindwa et al. (2019) [for TITAN2D] and different from, e.g., Sandri et al. (2018) or Clarke et al. (2020), where initiation points for the energy cone model were fixed at the center of each grid cell across the vent-opening spatial domain. I think you should add a brief comment on this. Many thanks.*

OK. Yes, our set of 1024 initiation points (i.e. vent locations) for the energy cone model changes for each of the 100 LHS designs explored, like in Rutarindwa et al. (2019). Our approach does not rely on a number of fixed 'grid cells' as initiation points, and the LHS design values are given in a continuous space. Our samples are denser where vent opening probability is higher, but their total number ( 105) is great enough to not leave any region free of our testing. We added the following sentence about that: " *In particular, our set of PDC initiation points changes for each of the 100 LHS designs explored (Rutarindwa et al., 2019), and we did not rely on a number of fixed grid cells as initiation points (e.g. Sandri et al., 2018; Clarke et al., 2020). The samples are denser where vent opening probability is higher, and their total number is great enough not to leave any region free of testing.* "

C20 - Now we indicated the typical runout distance, and we think that we already indicate the scale and the type of these eruptions.

*Please note how phreatomagmatic PDC events with average runout distances of 2 km may not be optimal examples to model PDC channelization. I think the future readers of the manuscript should be clearly informed about these aspects. Many thanks.*

OK. We added the following sentence on the possible appropriateness of other approaches in case the PDC are inertia-dominated pyroclastic currents: " *In this study we focus on gravity driven PDCs, but we remark that small scale phreatomagmatic eruptions may also produce dominantly inertial dilute fully-turbulent density currents, whose dynamics is better replicated by the so-called box model approach (Huppert and Simpson 1980).* "

C20 -We do not understand the references, in all those works, variable input conditions were adopted.

*LR340-344 - Maybe I did not express myself properly in this comment. What I meant was acknowledging previous studies that had used variable input conditions for the energy cone model: "We decided not to use variable input conditions for initial PDC characteristics, so our hazard assessment is only valid in this specific scenario of PDC size and friction angle (see Tierz et al., 2016a, b; Sandri et al., 2018; Aravena et al., 2020a; Clarke et al., 2020; for hazard assessments that incorporated this variability into the energy cone modeling)". Many thanks.*

OK. We followed your suggestion, adding the following sentence: " *(for examples of friction angle variability in studies based on the energy cone model, see Hayashi and Self, 1992; Sheridan and Macias, 1995; Tierz et al., 2016b; Sandri et al., 2018)* " We did not cite Aravena et al., 2020, Tierz et al., 2016a and Clarke et al. 2020 because they are already cited in the same paragraph.

C21 - Modified

*LR342-344 - Here, the authors deliberately choose to omit Tierz et al. (2016a) and Clarke et al. (2020), both of which use analogue-volcanoes data to parameterize the*

*model used in the presented study (the energy cone), to perform "more complete PDC hazard assessment considering variable size and friction properties". On the contrary, they choose to cite Sandri et al. (2012), which I agree represents an interesting use of analogue volcanoes for PDC hazard assessment, but it does not use the model presented in the current manuscript. Please also note that Sandri et al. (2012) is omitted later on in LR390-392, in a context where I believe it is a very relevant reference to cite. I struggle to understand all this reasoning on the choice of references.*

OK. We moved Sandri et al., 2012 later and put the other two in this sentence. Please see our previous comments about the eccessive requests of additional references, to explain our previous omission.

C21 - We think Tierz et al. 2016b does not use a dataset of dependent input parameters. The other citations are already included.

*LR344-347 - Tierz et al. (2016b) did use a dataset of dependent input parameters to explore the importance of theoretical uncertainty (a source of epistemic uncertainty) on the outputs of the energy cone model (please see, for instance, Sections 9.2.6, 9.3.4 and Figures 9.4, 9.8 of the article, https:// doi.org/ 10.1002/ 9781119028116.ch9). Additionally, Tierz et al. (2018) adopted an inter-eruption-size dependence between the PDC volume and the bed friction angle at Somma-Vesuvius (Italy), and discussed the role of theoretical uncertainty for TITAN2D (please see Section 4.2 of the article, https:// doi.org/ 10.1029/ 2017JB015383). Many thanks.*

OK. In my mind I mismatched the reference with Tierz et al. 2016a and I did not realize that you did test the correlations in 2016. We added the suggested reference in the same paragraph. We also added Tierz et al 2018 in the list, as you suggested.

C21 - We agree. We are not saying that this is the only effect, but it is significant. We include a new phrase to include your comment: " and also by the low vent opening probability computed at the flank E of the volcano ".

*LR356-358 - Many thanks. Still, I would say that something like the following could be more appropriate:*

*"Modelled invasion probability at the SE flank of San Salvador volcano and in the city of San Salvador tends to be low, where a significant effect is exerted by the topographic barrier of Cerro El Picacho, as well as of the entire volcanic edifice of SSVC (Fig. 8), given that vent-opening probabilities are highest on the NW flank of the volcano and on the surrounding plain to the N-NW of the edifice; and low on the E flank of the volcano (Fig. 8)."*

OK, thank you, we followed your suggestion.

C22 - We agree, the maximum is not in Santa Tecla, we modified the description of results (also we modified the conclusions). We think that to conditionate vent opening to a specific threshold is highly arbitrary (we would not know how to justify such choice).

*L358-361 – It is not clear to me how the description of the results has been modified around these lines. My suggestion of conditioning the vent-opening to the summit of SSVC, its N-NW flanks, and the surrounding plain in these directions was to address the plausibility of this assertion: "This is favoured by the high slope of the main edifice and the absence of significant topographic barriers in this direction".*

*If the high values of P(PDC) between Santa Tecla and San Salvador shown in Figure 8 decreased significantly when vent-opening was conditioned to the aforementioned areas, then the assertion would be difficult to support.*

*I think that, at least, you should reword the sentence in L358-361 following something similar to the text below. Many thanks.*

*"The highest PDC invasion probability calculated at the metropolitan area of San Salvador, 2-3%, is located in its western sector, i.e. between the western sector of San Salvador and Santa Tecla. These values might be explained by the peak in vent-opening probability shown in Figure 4 and/or by the high slope of the main edifice and*

*the absence of significant topographic barriers on the southern sector of the SSVC edifice"*

OK, thanks for the additional explanation. We followed your suggestion.

C23 - We believe that the original references are more appropriate as general reviews of PVHA and not just examples of vent opening probability maps.

*LR383-384 - In my personal view, I still struggle with Poland and Anderson (2020) being the most appropriate reference to accompany the sentence: "Several probabilistic assessments of vent opening at different volcanoes have been presented during the last decade". As I suggested in my first review, I think there are other references which would fit better in this sentence. Many thanks.*

As we said in the previous review: " We believe that the original references are more appropriate as general reviews of PVHA and not just examples of vent opening probability maps ".

C23 - We added Del Negro et al. (2013) and Thompson et al. (2015).

*LR388-390 - In my opinion, omitting Sandri et al. (2014) from this list is unfortunate, as that article presents a variety of examples of "models aimed at describing the dispersal of volcanic products". Many thanks.*

Thompson et al., 2015 already references Sandri et al. 2014 in its introduction. We added Tierz et al., 2018 instead.

C23 - We added Selva et al. (2014).

*LR390-392 - In my opinion, omitting Sandri et al. (2012) here is also quite unfortunate. The article is cited earlier in the text but it is in these lines where the reference is highly relevant. The article by Sandri et al. (2012) shows how vent-opening probabilities, and as a result PDC invasion probabilities, change in time as a result of changes in the recorded volcano monitoring data (please see Figures 3, and 6-8 of the article, https://*

*doi.org/ 10.1007/ s00445-011-0556-y). I think this is strongly relevant for the content of the presented manuscript. Moreover, I think adding one of the other two references (either Marzocchi et al., 2008 or Tonini et al., 2016) could still be beneficial for the future readers of your manuscript. Many thanks.*

OK. We moved Sandri et al. 2012 to this line. We also added Chaussard and Amelung 2012.

C25 - We are saying that an event counting approach would be more significantly affected by underrecording than a vent counting approach. When an improved datased will be available, repeating the analysis could provide more accurate results in terms of event counting, but we believe that we are already capturing the vent opening spatial distribution well enough. In fact, we are not expecting to have many buried vents in zones far from the mapped vents.

*OK. Many thanks for the clarification.*

You're welcome.

C26 - We do not think that this is key in this context. This is an illustrative application of the branching energy cone model, with the novelty that this is the first systematic application. We could use the traditional energy cone model or depth-averaged models, but these tests are beyond the scope of this paper.

*OK. I understand that the thematic vent-opening approach could be implemented using different PDC models (and models for other volcanic hazardous phenomena as well). Nevertheless, given that you present a first systematic application of the branching energy cone model for PVHA, I sincerely believe that it is important that any future reader of your manuscript can be referred (earlier in the main text of the presented manuscript) to previous research that investigated the first-ever applications of the (classical) energy cone to PVHA of PDCs (Tierz et al., 2016a, b; Sandri et al., 2018). This was an interesting matter of scientific debate back in those years and, judging by the recent*

*developments in the field, it will continue to be relevant over the years to come. Many thanks.*

OK. We referenced those three papers. Please see above comments.

C30 - We do not understand this suggestion. The example isolines are quite different.

*What I meant is that the example for Model 1 in Figure 2b seems to show well the influence of the presence of the fault on the isolines of vent-opening. However, it does not seem to show a clear influence of the presence of the past vent, which should also modify the isolines of vent-opening (compared to the case if the past vent was not there). Is this correct? I was just wondering if the isolines shown for Model 1 in Figure 2b could demonstrate the influence of both the fault line and the past vent on the vent-opening probabilities computed from Model 1. Many thanks.*

No, that is not correct. If the vent was not there, then the probability would be uniform. Thus the isolines are already showing the influence of both the data, according to Model 1. Please note that the isolines are not enclosing the entire fault, but only the portion closer to the vent.

**Suggested additional references**

(please see the list given here, https:// doi.org/ 10. 5194/ nhess-2020-382-RC2, for a few relevant references that are still missing from the presented manuscript. Many thanks)

Please also note the supplement to this comment:
https://nhess.copernicus.org/preprints/nhess-2020-382/nhess-2020-382-AC6-supplement.pdf

**Supplement:**

[revised manuscript text omitted]

---

## Author Response (AR1)

The answers point-by-point to the reviewers' comments are fully described in the interactive discussion. Modifications of the manuscript are presented in red and green in the marked version of the manuscript. Green texts are associated with modifications performed in the first revision iteration and red texts are related to the modifications performed in the second revision iteration.